# Validation of the coupled physical-biogeochemical ocean model NEMO-SCOBI for the North Sea-Baltic Sea system

Itzel Ruvalcaba Baroni[1], Elin Almroth-Rosell[1], Lars Axell[1], Sam T. Fredriksson[1], Jenny Hieronymus[1], Magnus Hieronymus[1], Sandra-Esther Brunnabend[1], Matthias Gröger[1,2], Ivan Kuznetsov[1,3], Filippa Fransner[1,4], Robinson Hordoir[1,5], Saeed Falahat[1], and Lars Arneborg[1]

[1]Department of Research and Development, Swedish Meteorological and Hydrological Institute, Norrköping, Sweden
[2]Department of Physical Oceanography and Instrumentation, Leibniz Institute for Baltic Sea Research Warnemünde, Rostock, Germany
[3]Alfred Wegener Institute, Helmholtz Centre for Polar and Marine Research, Bremerhaven, Germany
[4]Geophysical Institute, University of Bergen, and Bjerknes Centre for Climate Research, Bergen, Norway
[5]Institute of Marine Research, and Bjerknes Centre for Climate Research, Bergen, Norway

**Correspondence:** Itzel Ruvalcaba Baroni (itzel.ruvalcaba@smhi.se)

**Abstract.** The North Sea and the Baltic Sea still experience eutrophication and deoxygenation in spite of large international efforts to mitigate such environmental problems. Due to the highly different oceanographic frameworks of the two seas, modelling efforts so far mainly focused either on one or the other sea making it difficult to study inter-basin exchange of mass and energy. Here, we present an ocean model (NEMO-Nordic) coupled to the Swedish Coastal and Ocean Biogeochemical model (SCOBI), which covers the North Sea, the Skagerrak-Kattegat transition zone and the Baltic Sea. We address its validity to further investigate biogeochemical changes in the North Sea-Baltic Sea system. The model reproduces the long-term temporal trends, the temporal variability, the yearly averages and the general spatial distribution of all assessed biogeochemical parameters. It is particularly suitable to be used in future multi-stressor studies such as to evaluate combined climate and nutrient forcing scenarios. In particular, the model performance is best for oxygen and phosphate concentrations. However, there are important differences for chlorophyll-a and nitrate between model results and observations in coastal areas of the southeastern North Sea, the Skagerrak-Kattegat transition zone, the Gulf of Riga, the Gulf of Finland and the Gulf of Bothnia. These are partially linked to different local processes and biogeochemical forcing that lead to a general overestimation of nitrate. Our model results are validated for individual areas that are in agreement with policy management assessment areas, which gives an added value to better contribute to international programs aiming to reduce eutrophication in the Baltic Sea-North Sea system.

## 1 Introduction

The North Sea and the Baltic Sea share similar ecological problems, such as eutrophication and deoxygenation (e.g. Peeters et al., 1995; Rönnberg and Bonsdorff, 2004; Greenwood et al., 2010; Gustafsson et al., 2012; Große et al., 2016; Andersen et al., 2017), despite being two substantially different basins. They differ from each other in bathymetry, geometry and forcing conditions which control their ocean dynamics that respectively lead to two fundamentally different turnover time scales. The

area between the two seas, hereafter referred to as the Skagerrak-Kattegat transition zone, includes several sub-basins (Fig. 1) and is the only connection between the Baltic Sea and Atlantic waters. This zone is also one of the most heavily human impacted areas of the North Sea-Baltic Sea system (e.g. Korpinen et al., 2013; Kenny et al., 2017) and therefore, relevant to include in ecological assessment studies for both seas. Previous biogeochemical studies in the Skagerrak-Kattegat transition zone, both model and observational based, have focused on nutrient fluxes, eutrophication, summer algal blooms and primary production, but mainly in the Kattegat for the period between 1950 and 2000. These studies concluded that there was a decline in phosphorus and bottom oxygen concentrations after ∼1980 (Andersson, 1996; Rasmussen and Gustafsson, 2003), that primary productivity increased until at least the year 1980 after which the trends are less clear (Carstensen and Conley, 2004; Rydberg et al., 2006), and that important nutrient gradients exist within the Kattegat (Danielsson et al., 2004).

The Baltic Sea is a landlocked sea with several sub-basins separated by sills. It is shallow with an average depth of about only 53 m, however, encompassing the Gotland deep (∼249 m) and the Landsort Deep (∼459 m) in the Eastern and Western Gotland basins, respectively (e.g. Jakobsson et al., 2019). It has brackish water with both a north to south salinity gradient and a strong perennial stratification in all deep basins. In the shallow areas, winds are capable of mixing the entire water column down to the sea floor, while the deeper basins have a permanent strong halocline. In the central Baltic Sea, horizontal advection of saline water below the permanent halocline can occur (Reissmann et al., 2009). The stratification is due to large freshwater input from rivers at the surface and advection of dense salty and oxygenated waters to deeper layers from the North Sea entering through the Danish Straits and spreading across the Baltic Sea basins (e.g. Stigebrandt, 1987; Döös et al., 2004; Leppäranta and Myrberg, 2009). However, this stratification also inhibits the supply of oxygen to the deep waters through vertical mixing. Thus, oxygen transport to the deep water occurs mainly through intermittent inflows of saline water through the Danish Straits, primarily during the winter at irregular (yearly to multiyearly) intervals (e.g. Gustafsson, 1997; Omstedt et al., 2004; Lass and Matthäus, 1996; Feistel et al., 2008; Hordoir et al., 2015). This strong stratification also promotes a homogeneous distribution of it's biogeochemical properties and a long residence time of the water masses (ca. 35 years) (Döös et al., 2004; Wulff et al., 2001; Meier and Kauker, 2003; Feistel et al., 2008). Primary productivity in the Baltic Sea is mainly limited by nitrate, which favours the growth of nitrogen-fixing cyanobacteria (e.g. Granéli et al., 1990; Janssen et al., 2004; Eilola et al., 2009; Kuliński et al., 2022).

Contrary to the Baltic Sea, the North Sea is much more dynamic, with a residence time of only a few years (Otto et al., 1990; Hordoir et al., 2019), and is heavily influenced by tides. Consequently, it is generally well-mixed and well-oxygenated, but its deeper areas are periodically stratified. Deoxygenation can occur in coastal and stratified areas of the Eastern North Sea, primarily when influenced by riverine input (Devlin et al., 2022; van Leeuwen et al., 2023). Tidal mixing fronts occur between deep stratified and tidally mixed shallow waters (e.g. Ikeda et al., 1989; McGlade, 2002; Ducrotoy et al., 2000; Sündermann and Pohlmann, 2011). About 53% of the North Sea is permanently, seasonally or intermittently stratified (van Leeuwen et al., 2015). While nutrients have been identified as the main factor controlling primary productivity in the North Sea, other factors such as temperature, stratification and light penetration depth, have a regional impact (e.g. Holt et al., 2012; Ly et al., 2014; Burson et al., 2016). Its ocean dynamics and biogeochemistry are also greatly influenced by the adjacent open Atlantic Ocean (Winther and Johannessen, 2006; Gröger et al., 2013; Mathis et al., 2019; Huthnance et al., 2022, e.g.).

Diatoms and flagellates, which are adapted to a wide range of salinity conditions, can dominate the primary production in both seas. More specifically, in the Baltic Sea, diatoms have been found to dominate stations with the highest salinities, while flagellates and other autotrophs were most abundant at low saline stations (Olofsson et al., 2020a). In the North Sea, tidal mixing fronts favour the growth of diatoms (e.g. Ikeda et al., 1989; McGlade, 2002; Ducrotoy et al., 2000; Sündermann and Pohlmann, 2011), while stratified waters favour the growth of flagellates (e.g. van Leeuwen et al., 2015). However, their spatial distribution and total biomass in both seas may vary significantly from year to year and from subbasin to subbasin (e.g. Henriksen, 2009; Reid et al., 1990; Ford et al., 2017; Olofsson et al., 2020a). In contrast, filamentous cyanobacteria (hereafter referred to cyanobacteria) do not tolerate high salinity conditions (e.g. Mazur-Marzec et al., 2005; Olofsson et al., 2020b) and therefore, do not grow in the North Sea. However, they are key in the brackish Baltic Sea, where they can dominate the (late−)summer primary production (e.g. Finni et al., 2001; Janssen et al., 2004; Olofsson et al., 2020b).

The entire Baltic Sea-North Sea system has experienced increased anthropogenic nutrient loads from rivers, the atmosphere and point sources (i.e. any single identifiable source of pollution from which pollutants or nutrients are discharged, such as sewage) since the early 1900s and especially after the 1950s (e.g. Savchuk et al., 2008; Vermaat et al., 2008; Gustafsson et al., 2012; Holt et al., 2012). This has led to an acceleration of algal growth (i.e. eutrophication), especially in coastal waters, and oxygen deficiency in bottom waters. In particular, the (late-)summer cyanobacteria blooms occur in the Baltic proper, which have been found to be closely linked to the variability of phosphorus supply to surface waters and stratification and to changing redox conditions in the water column (Kahru et al., 2000; Janssen et al., 2004; Eilola et al., 2009). Because of the different renewal time scales, the nutrients are recycled much faster in the North Sea than in the Baltic Sea. However, neither the North Sea nor the Baltic Sea have yet fully recovered despite large efforts to reduce nutrient loads since the 1980-1990s. In the North Sea, the persistent eutrophication has been linked to a stronger reduction in phosphorus versus nitrogen loads, which have created nutrient imbalances that affect the growth and species composition of marine phytoplankton communities (Burson et al., 2016; Ly et al., 2014). In the Baltic Sea, the slow recovery is linked to the fact that the water column nutrient inventory is tightly coupled to that of the sediments due to the long and frequent exposure to low-oxygen conditions. This accelerates the recycling of phosphorus from sediments under deoxygenated bottom waters (e.g. Koop et al., 1990; Mort et al., 2010; Jilbert and Slomp, 2013).

Studies on the ecological and geopolitical status of the North Sea and the Baltic Sea highlight the need for more long-term monitoring data and improved models to support the Marine Strategy Directives and international programs aiming to improve the ecological conditions in the Baltic Sea-North Sea system (Ducrotoy and Elliott, 2008; Mee et al., 2008; Koho et al., 2021). Indeed, several EU management programs and directives are now well established (e.g. OSPAR, 2003; Borja, 2006; HELCOM, 2006) and can in a challenging but possible joint effort cover the whole of the North Sea and the Baltic Sea (Koho et al., 2021). In studies by Almroth and Skogen (2010) and Eilola et al. (2011b), respectively, the eutrophication status of the North Sea-Baltic Sea system for the year 2005 and the years 2001-2006 was assessed based on observations and ensemble model results. However, none of the models used in these studies include both seas and therefore, they instead combined results from several models to analyze the entire North Sea-Baltic Sea system. Covering both seas in one single model is a big advantage as this avoids the need to formulate reasonable lateral boundary conditions, often based on a limited number of observations that

may oversimplify the actual dynamics. This can greatly influence the model results, which is particularly true in the Kattegat-Skagerrak transition zone. To our knowledge, only two other 3D ocean models with fully coupled biogeochemistry cover both the North Sea and the Baltic Sea (Daewel and Schrum 2013 and Maar et al. 2011). Their results show generally good agreement with observations in time and space, but contain biases for different biogeochemical parameters. Thus, large model uncertainties still exist for the Baltic Sea-North Sea system, linked to differences in model set-ups and process descriptions. Having a variety of independent models for similar domains is important to assess such uncertainties (Eilola et al., 2011a).

In the present study, we use the ocean model NEMO-Nordic (Hordoir et al., 2019) with a model domain covering both the North Sea and the Baltic Sea, for the first time coupled to the Swedish Coastal and Ocean Biogeochemical model (SCOBI) that earlier has been used in many applications for the Baltic Sea (e.g. Almroth-Rosell et al., 2015; Eilola et al., 2009, 2012; Meier et al., 2012) and the Swedish coast (Edman et al. 2018). We present model results and model skills compared to observational-based estimations. We also link our analysis to the latest policy management areas agreed in the 4[th] application of the Comprehensive Procedure (COMP4) within the Oslo-Paris Commission (OSPAR) for North Atlantic waters and the North Sea and the Helsinki Commission (HELCOM) for the Baltic Sea (Fig. 1) to identify regional model performances and to, in the future, better contribute to European initiatives on de-eutrophication of both the North Sea and the Baltic Sea.

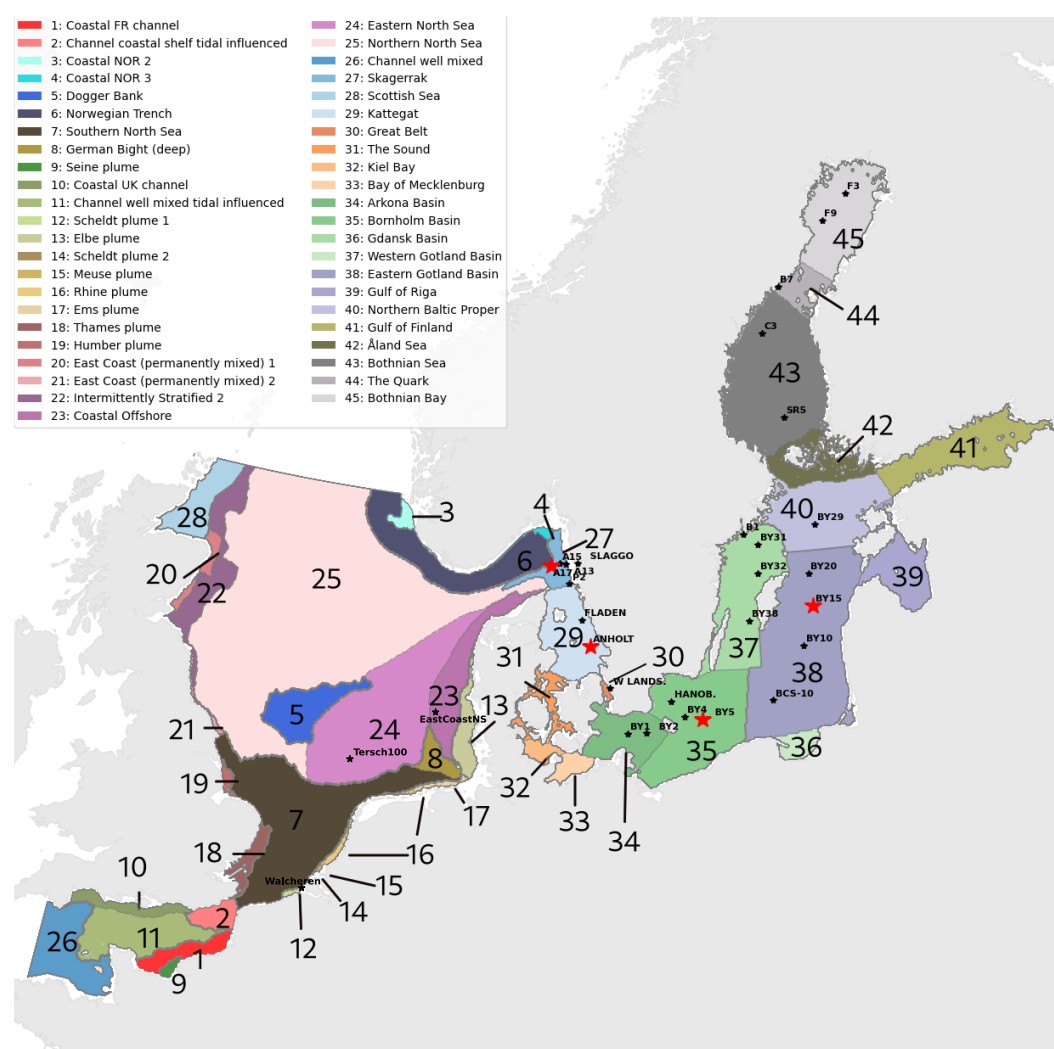

**Figure 1.** Combined division of assessment areas from COMP4 OSPAR 2021 (1 to 28) and Baltic Sea assessment units from HELCOM 2021 (29 to 45) adapted to our model domain. Note the HELCOM division is shown for the Kattegat, while the OSPAR is shown for the Skagerrak. Stars represent the analysed stations. Highlighted in red are the stations shown in this study. The bathymetry of the domain is shown in Hordoir et al. (2019).

## 2 Material and methods

### 2.1 Model Description

We use the coupled physical-biogeochemical ocean model NEMO-SCOBI, in which the ocean component is based on the Nucleus for European Modelling of the Ocean (NEMO) framework (Madec et al., 2017), version 3.6. This three dimensional model has been specifically configured for the Baltic Sea and the North Sea (NEMO-Nordic; Hordoir et al. 2015, 2019) by SMHI and covers an area from 4.15278 $^o$W to 30.1802 $^o$E and 48.4917 $^o$N to 65.8914 $^o$N (Fig 1). The biogeochemistry is simulated by SCOBI also developed at SMHI (Marmefelt et al. 1999; Eilola et al. 2009; Meier et al. 2012; Eilola et al. 2012; Almroth-Rosell et al. 2015). The SCOBI model has been successfully coupled to different ocean and coastal models (e.g. Almroth-Rosell et al., 2011; Edman et al., 2018). However, previous model domains only cover either the Baltic Sea including the Kattegat or the Swedish coastal waters. Here, the SCOBI model is coupled for the first time to NEMO-Nordic and therefore, computes changes of key biogeochemical properties in the water and sediments for the entire NEMO-Nordic domain. The ocean model coupled to the biogeochemistry is referred to as NEMO-SCOBI.

In the present study, the model was integrated from 1961 to 2017 and validated for recent years (2001-2017) when biogeochemical observations are the most abundant. The period before 1975 is regarded as a spin-up. For the physics, we use all settings as in Hordoir et al. (2019), but with an updated physical forcing and representation of fast ice, allowing it to form only in shallow areas when attached to the shore (Siiriä et al., 2022). We also use daily river forcing instead of monthly. The physical and the biogeochemical settings are described in section 2.1.3 and section 2.1.4, respectively.

#### 2.1.1 Ocean hydrodynamic model: NEMO-Nordic

NEMO-Nordic has a regular grid with 56 vertical levels with a resolution of 3 m close to the surface, decreasing to 22 m at the bottom of the deepest part of the domain (Norwegian trench) and a horizontal resolution of approximately 2 nautical miles ($\sim$3.7 km). NEMO-Nordic has two open boundaries: a meridional one located in the western English Channel between Brittany and Cornwall, and a zonal one located between Scotland and Norway (Fig. 1). For further details on ice and ocean dynamics see Pemberton et al. (2017) and Hordoir et al. (2019), respectively.

#### 2.1.2 Biogeochemical model: SCOBI

The SCOBI model (first described by Marmefelt et al. 1999) is a process-oriented nutrient, phytoplankton, zooplankton, and detritus (NPZD) model that traditionally simulated three major marine biogeochemical cycles (nitrogen, phosphorous and oxygen) in both the water column and sediments (Eilola et al., 2009; Almroth-Rosell et al., 2011, 2015). Now coupled to NEMO-Nordic, SCOBI includes also the marine silicon cycle. Currently, the model has 17 biogeochemical state variables, of which 13 are pelagic and four are benthic. The state variables are summarized in Table 1 and a diagram summarizing the biochemical cycling in SCOBI is shown in Figure 2. The inorganic forms in the water column are represented by six state variables: dissolved oxygen (O2), nitrate (NO3), ammonia (NH4), phosphate (PO4), mineral-bound inorganic phosphorus (WIP)

and dissolved silicate (DSi). Dead particulate organic material in the water column is separated in three variables as detritus: nitrogen detritus (DETN), phosphorus detritus (DETP) and amorphous biogenic silica (OPAL). Nutrients are assimilated by three phytoplankton functional groups defined as diatoms (PHY1), flagellates and others (PHY2), and cyanobacteria (PHY3), which are all grazed by bulk zooplankton (ZOO). In SCOBI, hydrogen sulfide concentrations are represented by 'negative oxygen' equivalents so that $-[O_2] = \frac{1}{2} \cdot [H_2S]$ in $\mathrm{ml\,l^{-1}}$ (Fonselius, 1962). The model accounts for one sedimentary layer containing the benthic reservoirs of nitrogen (BN), silicon (BSi), organic phosphorus (BOP) and inorganic phosphorus (BIP), where BIP represents a benthic pool of phosphate adsorbed to mineral particles (e.g. iron-oxides) (Almroth-Rosell et al., 2015).

The main processes included in the water column are primary production, $N_2$-fixation, grazing and sloppy feeding, remineralisation of organic matter and its resulting oxygen consumption, sinking of particles, nitrification, denitrification and organic matter deposition to the sediments. Within the sediments, dissolved nutrients can be released back to the water column due to remineralisation of organic matter, and deposited organic material can be resuspended back to the water column due to currents and wave bottom friction. Under oxic conditions a fraction of the phosphate from remineralized benthic organic phosphorus adds to the benthic pool of inorganic phosphorus while the other fraction is released directly to the water column. The sizes of the fractions are oxygen dependent. Also, scavenging of phosphorus from the water column takes place adding to the benthic inorganic phosphorus pool. During anoxic conditions, all the remineralized phosphorus is directly released to the water column, as well as a fraction of the benthic pool of inorganic phosphorus. For the benthic nitrogen, a fraction of the remineralized nitrogen is removed by benthic denitrification ($BDEN$). The fraction depends on the available oxygen concentrations in the bottom waters with a medium rate during oxic conditions and a maximum rate under low-oxygen conditions that decreases rapidly to a null rate during anoxic conditions. However, benthic denitrification can continue when the bottom water is anoxic if nitrate is available in bottom waters following equation A26 (Appendix A2). The parametrization of the other processes above is described in Eilola et al. (2009) and for the latest modifications to the benthic phosphorus in Almroth-Rosell et al. (2015). In the SCOBI version coupled to NEMO, rates and dependencies for phytoplankton growth were modified with respect to previous SCOBI versions in order to account for silica limitation of diatoms (not included in earlier versions), to improve the occurrence of dominant groups in both the North Sea and the Baltic Sea and to limit cyanobacteria growth in the Skagerrak-Kattegat transition zone and stop their growth in the North Sea. To include benthic processes in the North Sea, new rates of burial and nutrient release from sediments and resuspension of benthic organic nutrients due to wave and current friction were introduced. In addition, the parametrization of oxygen penetration depth was replaced by the oxygen concentration in bottom waters in benthic-redox dependent processes for phosphorus as NEMO-SCOBI does not include oxygen in the sediment layer. For clarity, we detail the current SCOBI formulations for phytoplankton growth and all relevant sedimentary processes in Appendix A, sections A1 and A2.

**Table 1.** SCOBI state variables

| Variable | Description | Units |
|----------|-------------|-------|
| | **Water column** | |
| PHY1 | Diatoms | $\mathrm{mg\,Chla\,m^{-3}}$ |
| PHY2 | Flagellates and others | $\mathrm{mg\,Chla\,m^{-3}}$ |
| PHY3 | Cyanobacteria | $\mathrm{mg\,Chla\,m^{-3}}$ |
| ZOO | Zooplankton | $\mathrm{mg\,C\,m^{-3}}$ |
| PO4 | Phosphate | $\mathrm{mmol\,P\,m^{-3}}$ |
| WIP | Mineral-bound inorganic phosphate | $\mathrm{mmol\,P\,m^{-3}}$ |
| NO3 | Nitrate | $\mathrm{mmol\,N\,m^{-3}}$ |
| NH4 | Ammonium | $\mathrm{mmol\,N\,m^{-3}}$ |
| Si | Silica | $\mathrm{mmol\,Si\,m^{-3}}$ |
| DETN | Nitrogen detritus | $\mathrm{mg\,C\,m^{-3}}$ |
| DETP | Phosphorous detritus | $\mathrm{mg\,C\,m^{-3}}$ |
| OPAL | Biogenic siliceous material | $\mathrm{mmol\,Si\,m^{-3}}$ |
| O2 | Dissolved oxygen | $\mathrm{ml\,O_2\,l^{-1}}$ |
| | **Sediments** | |
| BOP | Benthic organic phosphorous | $\mathrm{mmol\,P\,m^{-2}}$ |
| BIP | Benthic inorganic phosphorus | $\mathrm{mmol\,P\,m^{-2}}$ |
| BN | Benthic nitrogen | $\mathrm{mmol\,N\,m^{-2}}$ |
| BSi | Benthic silicon | $\mathrm{mmol\,Si\,m^{-2}}$ |

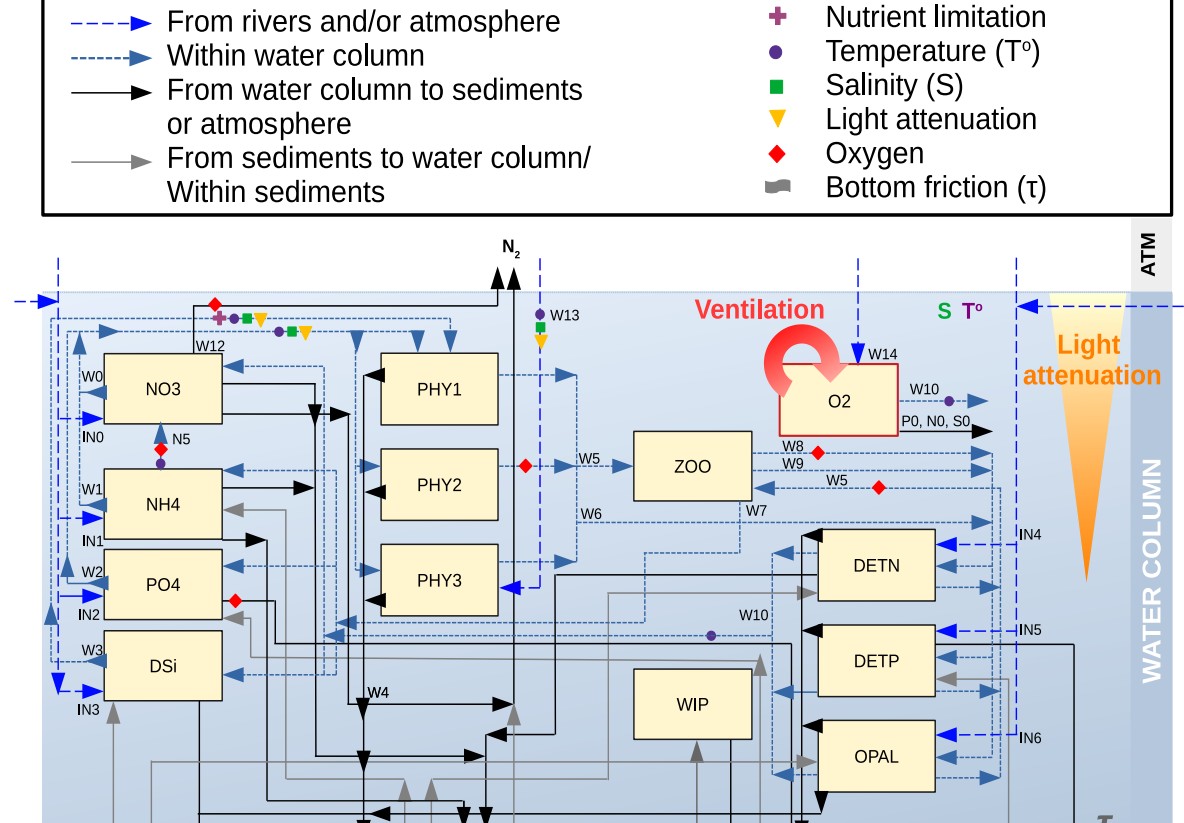

**Figure 2.** Schematics of the biogeochemical processes included in NEMO-SCOBI, where main variables determining a process (as independent variable or as a threshold) are indicated with symbols. Processes for plankton, detritus and oxygen are: W0 = Nitrate uptake for phytoplankton growth, W1 = Ammonium uptake for phytoplankton growth, W2 = Phosphate uptake for phytoplankton growth, W3 = Silicate uptake for diatoms growth, W4 = Sinking and sedimentation of phytoplankton, W5 = Grazing, W6 = Mortality, W7 = Excretion/Sloppy feeding, W8 = Predation, W9 = Zooplankton faeces, W10 = remineralisation of detritus and oxygen consumption, W11 = Sinking and sedimentation of detritus, W12 = Water column denitrification, W13 = $N_2$-fixation by cyanobacteria, W14 = Oxygen exchange between atmosphere and surface water. Processes for phosphate are:

**Figure 2.** ... caption continuation ... P0 = Phosphate release from decomposition of organic matter in sediments (P0' = Fraction of the mineralized benthic organic phosphorus that is directly released to the overlying water and P0" = Fraction that is transferred to the benthic inorganic phosphorus), P1 = Resuspension of inorganic phosphorus, P2 = Deposition of inorganic phosphorus, P3 = Phosphate release from benthic inorganic phosphorus, P4 = Scavenging of phosphorus into sediments under oxic conditions, P5 = Resuspension of organic phosphorus, P6 = Deposition of organic phosphorus, P7 = Burial of inorganic phosphorus, P8 = Burial of organic phosphorus. Processes for nitrogen are: N0 = Nitrogen release from decomposition of organic matter in sediments (N0' = Fraction released as nitrate and N0" = Fraction released as ammonium), N1 = Resuspension of organic nitrogen, N2 = Ammonium adsorption to particles, N3 = Deposition of organic nitrogen, N4 = Total benthic denitrification (which consists of 2 pathways: denitrification of pelagic nitrate and denitrification of benthic nitrogen) N5 = Water column nitrification, N6 = Burial of organic and inorganic nitrogen. Processes for silica are: S0 = Silicon release of benthic silicon, S1 = Resuspention of benthic silicon, S2 = Deposition of silicon, S3 = Burial of organic and inorganic silicon. Input fluxes from rivers are for nitrate, ammonium, phosphate, silica and detritus for both phosphorus and nitrogen (IN0 to IN5). Atmospheric inputs are for all nutrients except silica, which is only supplied by rivers.

### 2.1.3 Physical setting

The meteorological forcing is taken from the reanalysis data set Uncertainties in Ensembles of Regional ReAnalyses (UERRA; e.g. Dahlgren et al., 2016, available at www.uerra.eu) with a spatial resolution of 11 km and a time resolution of 1 hour for wind, air pressure, air temperature, humidity, and solar and long-wave downward radiation, and 12 hours for precipitation (i.e.
rain and snow).

The open boundary forcing consists of barotropic currents, sea level and nine tidal constituents as well as monthly salinity and temperature data. The barotropic currents and sea level are calculated using the two dimensional storm-surge model North Atlantic Model (NOAMOD; She et al., 2007) for 1979-2017. These were corrected for baroclinic effects using monthly sea level data from ORAS4 to improve the ocean circulation in the North Sea. To extend it back in time, we applied a neural
network based regression technique, following Hieronymus et al. (2019). The salinity and temperature profiles are monthly mean values interpolated from an ORAS4 configuration (Balmaseda et al., 2013).

Daily values of runoff for the period 1961 to 2019 were provided by a dedicated simulation with the Hydrological Predictions for the Environment model with the European application v.3.1.8 (E-HYPE; Donnelly et al. 2016). These were here reduced by a factor of 0.9 by recommendation of the E-HYPE developers, due to an overestimation in the precipitation in this E-HYPE
run, especially over the Baltic Sea. For more details on the applied river runoff see Ruvalcaba Baroni et al. (XXXX). For the physical initial conditions we used restart files from the simulation in Hordoir et al. (2019) that were the closest to the observations for physical properties at the start of the simulation.

### 2.1.4 Biogeochemical settings

For the initialization, the biogeochemical initial values are derived from a combination of typical North Sea and Baltic Sea
profiles and spin-up values from previous sensitivity tests performed with NEMO-SCOBI. Thus, they already represent the physical-biogeochemical conditions, in opposition to using a homogeneous 3D field, as imprint in the initialization that is then

followed by the actual spin-up. The forcing for the open boundary conditions for the biogeochemical model was created based on the ICES data base (Beszczynska-Möller et al., 2009) and interpolated as seasonal cycle climatology to the model grid.

The atmospheric nutrient forcing, consisting of bioavailable nitrate, ammonium and phosphate, and nitrogen detritus, was interpolated as seasonal cycle climatology from yearly averages of total atmospheric loads of nitrogen and phosphorus in the Baltic Sea and period averages reported per basin for 1994 to 2006 in Savchuk et al. (2012). Ammonium, nitrate and detritus are here assumed to be 40, 50 and 10 % of the total atmospheric nitrogen load, respectively. A constant value per nutrient per basin and per month is then given to individual model grid cells in $\mathrm{mmol\,m^{-2}\,s^{-1}}$. While both nitrogen and phosphorus atmospheric input increase from 1975 to the 1980s, only nitrogen clearly decreases after the 1980s. The resulting atmosphere input to the Baltic Sea of both total phosphorus and nitrogen are comparable to those reconstructed by Gustafsson et al. (2012) and reported for recent years by HELCOM (Gauss et al., 2022). However, historic nitrogen atmospheric inputs are uncertain, and our method resulted in higher loads ($\sim$100 $\mathrm{kton\,N\,y^{-1}}$) than those in Gustafsson et al. (2012) for years before 1995. In the North Sea, historic atmospheric loads are also uncertain, especially those for phosphorus. Here, we use the total atmospheric nutrient load for the Baltic proper to reconstruct the specific loads in the North Sea, resulting in nitrogen loads that are comparable to those reported by OSPAR for years 1994 to 2014 (Bartnicki et al., 2019) but fairly constant values for phosphorus loads after the year 1970.

The daily riverine nutrient loads are based on the dedicated E-HYPE run, which includes a more realistic number of river outlets than those reported in the observational data sets. While the E-HYPE data set captures well the interannual variability, it does not account for the increase of nutrients due to increased fertilizers in the 1960s and the consequent reduction due to nutrient regulation policy in the 1980s. Hence, we do not use the river forcing as originally provided. Instead, the riverine nutrient loads were corrected for each year and each basin to the level of compiled observational data for nutrient loads combining two data sets, one for the North Sea (the ICG-EMO database of European rivers; Lenhart et al. 2010) and one for the Baltic Sea (Gustafsson et al., 2012). The result is an E-HYPE forcing, with river points adapted to the NEMO-Nordic grid (Fig. 3), with slightly reduced runoff and modified nutrient loads that include the increase and the following decrease seen in observations (Fig. 4). The latter nutrient forcing is available, described and compared to the former datasets in Ruvalcaba Baroni et al. (XXXX).

In addition, the detritus loads for nitrogen and phosphorus in the SCOBI model are reduced by a factor of 0.3 for nitrogen and 0.75 for phosphorus once they reach the coastal waters (i.e. the river loads of detritus enter the detritus pool in the SCOBI model as 0.3 x river-DETN and 0.75 x river-DETP, respectively). This is to account for only the bioavailable fraction of the organic matter coming from rivers and coastal retention. Because the response to nutrient removal of different coastal types is poorly quantified, especially in the North Sea, these factors are taken from previous studies in the Baltic Sea (Eilola et al., 2011a; Edman and Anderson, 2014; Asmala et al., 2017). Because E-HYPE does not include the silica cycle, we use a compilation of observations for both the North Sea and the Baltic Sea adapted to the NEMO-Nordic river points to reconstruct the silica river loads. Two main data sets for silica loads (the Baltic Nest Institute - Stockholm University; personal communication with Bo Gustafsson and the ICG-EMO database of European rivers; Lenhart et al. 2010) were combined to include as many observation points as possible for both the North Sea and Baltic Sea (not shown).

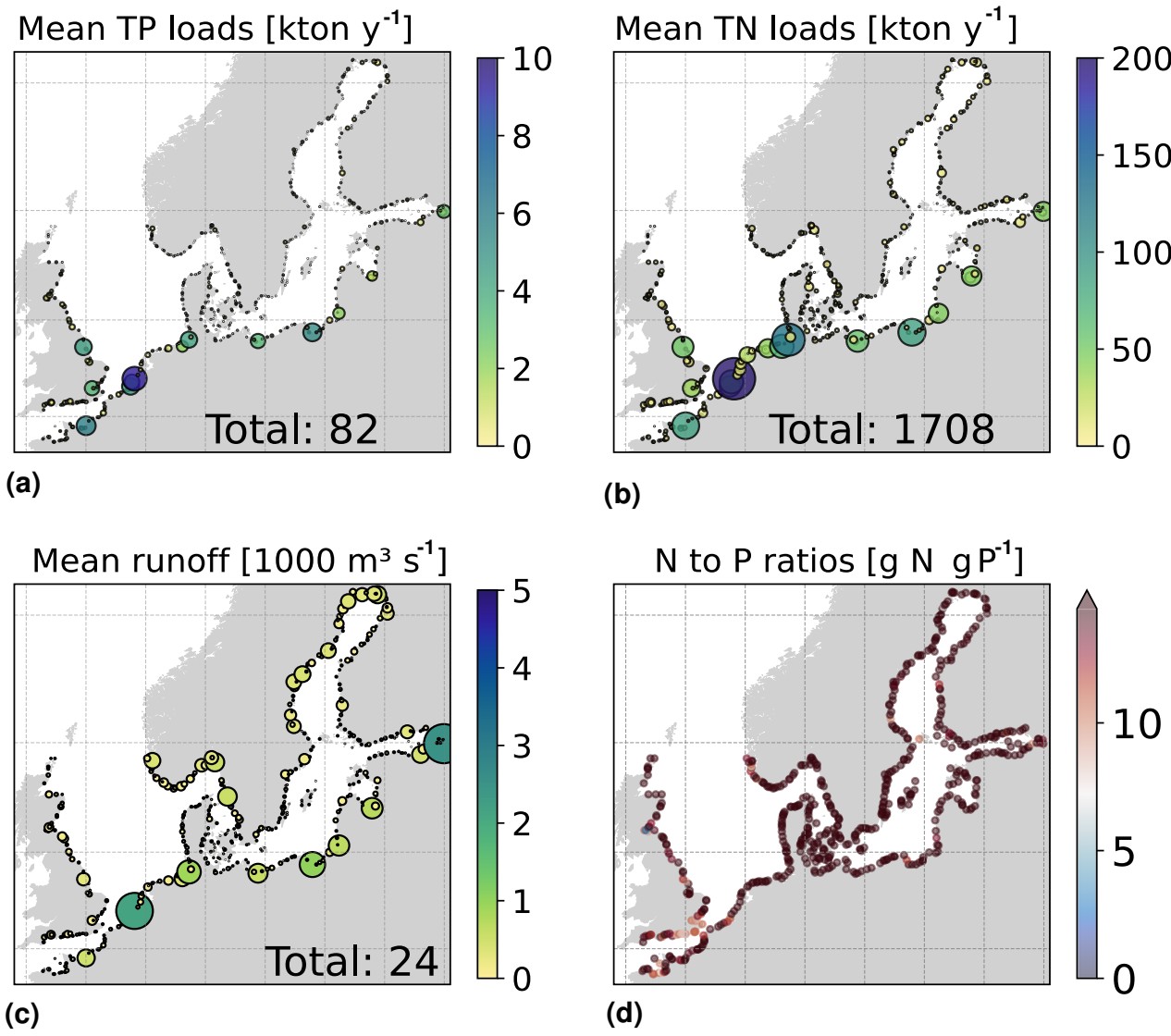

**Figure 3.** Averaged riverine loads of a) total phosphorus ($TP = PO4 + DENP$), b) total nitrogen ($TN = NO3 + NH4 + DETN$) and c) runoff for the period 2001 to 2017 in the river forcing applied to the model domain. Circle sizes illustrate the relative load contribution in the area and the total period average for the domain is also shown. d) The nitrogen and phosphorus (N to P) ratios for each river point, where the Redfield ratio is $\sim 7\,\mathrm{g\,N\,(g\,P)^{-1}}$.

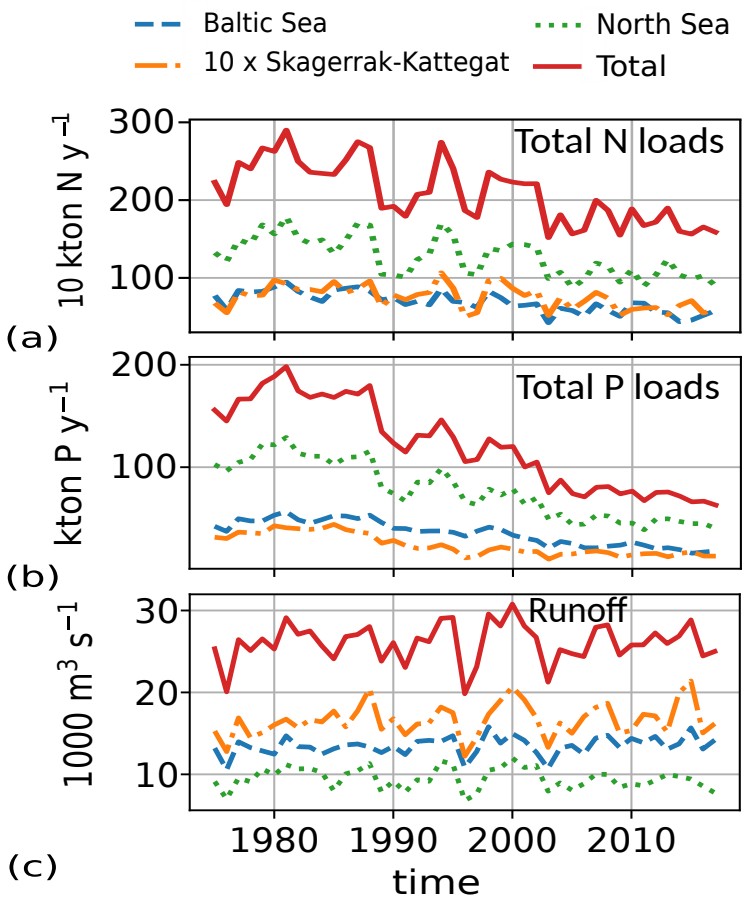

**Figure 4.** Riverine loads of a) total nitrogen ($TN = NH4 + NO3 + DETN$), b) total phosphorus ($TP = PO4 + DETP$) and c) total runoff from 1975 to 2017 in the model domain and its sub-basins: the North Sea, the Skagerrak-Kattegat transition zone and the Baltic Sea. For plotting reasons, numbers for the Skagerrak-Kattegat are here multiplied by ten.

## 2.2 Observations and validation method

In order to compare model results to observations, nitrate, nitrite, ammonium, phosphate, particulate organic nitrogen, particulate organic phosphorus, chlorophyll-a, oxygen and hydrogen sulfide, as well as sea water temperature and salinity were downloaded from the open SHARK database (The Swedish national archive for oceanographic data; https://sharkweb.smhi.se) and the ICES database (The International Council for the Exploration of the Sea; www.ices.dk). Observations measured more than once a day were averaged to obtain one daily value. However, both datasets are not homogeneously distributed in time, space and depth. Except at fixed stations, which have a temporal resolution of a maximum of twice a month, most positions in the ICES database are rarely measured more than twice a year.

### 2.2.1 SHARK database

The observations from the SHARK database were used to analyze long-term model results and their interannual variability at 27 selected stations in the Skagerrak-Kattegat transition zone and in the Baltic Sea (Fig. 1). These stations were selected based on their distribution in relevant sub-basins of the Skagerrak-Kattegat area and the Baltic Sea (Fig. 1). All stations are well documented and broadly used in monitoring studies. However, observations for chlorophyll-a at discrete depths are lacking at stations B7, C3, F3 and F9. At C3, only observations for phosphate in the water column and for oxygen in bottom waters are available. At B7, nitrate, nitrite and ammonium observations are lacking. The number of observations of nitrate and chlorophyll-a available in Skagerrak is also scarce before the year 2000.

Euxinia (i.e. waters with no oxygen but free hydrogen sulfide; $H_2S$) is converted from $H_2S$ to "negative oxygen" equivalents in the same way as in SCOBI (see section 2.1.2). If both oxygen and hydrogen sulfides are present and oxygen concentrations are above the detection limit, then we use the oxygen concentration. On the other hand, if oxygen concentrations are below the detection limit, then we use the hydrogen sulfide conversion.

Daily time series at selected stations in the model were plotted against observations of phosphate, nitrate plus nitrite, oxygen, discrete chlorophyll-a, salinity and temperature for the period 1975 to 2017. These are compared to model results at several discrete depths at all selected stations. For surface waters, both model and observations were averaged over the first ∼10 m for comparison. For bottom waters, observations within the bottom layer of the model were averaged. The daily time series are also used to evaluate the model skill at selected stations for the main biogeochemical parameters (section 2.2.3).

The salinity, temperature and current fields in the North Sea-Baltic Sea system show large variability on decadal time scales that are superimposed on multi-decadal long-term trends, particularly in the North Sea (Daewel and Schrum, 2017). This variability is not necessarily in phase in all regions. In order to evaluate the model interannual variability, we analyzed averages for a 17 year period (from 2001 to 2017), so that at least one decadal cycle is included as well as the years with the most observations. Note that within this chosen period several medium to strong inflows to the Baltic Sea occurred, e.g. in 2003 and 2014 (Mohrholz et al., 2015; Mohrholz, 2018). In addition, a simple linear regression analysis is performed for the periods 1975-1996 and 1996-2017 for both observations and model results to detect differences in long-term trends. The year 1996 has been chosen as a reference year when nutrients are high in most of the model domain and observations, but not necessarily

when these are at their maximum. Therefore, the regression analysis here cannot be used to detect the exact timing of potential

changes in trends. If the number of observations is less than 40 within the regression time period, no regression line is plotted.

The p-values are calculated and evidenced against the null hypothesis, considering a significant trend when p-value $\leq 0.05$.

Monthly (m), seasonal (s) and period (p) averages together with their corresponding standard deviations with time are

calculated at all depths. Note that seasonal averages for the winter months are here considered to be from December to February,

the spring months are from March to April, the summer months are from May to August and the autumn months are from

September to November. The number of observations for each averaged profile (nobs$_{\text{m,s,p}}$) was calculated in percentage based

on the corresponding total number of days (ndays$_{\text{m,s,p}}$) within the period so that the coverage in % = nobs$_{\text{m,s,y}}$ · 100/ndays$_{\text{m,s,y}}$.

We take a minimum nobs$_{\text{m,s,y}}$ of 3 to calculate the averages and no interpolation in depth or time was performed. Monthly

values and their standard deviation were also averaged over the first $\sim 10$ m. We evaluate the model skill at selected stations as

described in section 2.2.3.

### 2.2.2   ICES database

To analyze the spatial variability in surface waters, we use the ICES observations for nitrate plus nitrite, phosphate and

chlorophyll-a within our model domain. These were seasonally averaged from 2001 to 2017 and over the first $\sim 10$ m. The dif-

ference between each observation data point and its corresponding model point in surface waters is calculated as $M_i - O_i$. We

also use the ICES data base to evaluate the spatio–temporal model skill for the main biogeochemical parameters (section 2.2.3)

following the management areas agreed in OSPAR for the North Sea and in HELCOM for the Baltic Sea. These area divisions

were combined but, as the area definition from OSPAR and HELCOM in the Kattegat overlap and differ from each other, we

use the HELCOM definition for the Kattegat (Fig. 1; HELCOM, 2006; OSPAR, 2022). Oxygen is only evaluated below surface

waters, because in surface waters the model is mainly controlled by the atmospheric conditions, resulting in surface oxygen

values close to saturation and therefore always in good agreement with observations. The number of observations used per area

in this study is illustrated in Fig. 5. Areas with less than 100 observations for all four variables within the period 2001 to 2017

are not shown. These are areas 3, 10, 14, 19, 20 and 21. In addition, the number of observations of oxygen below surface is

also less than 100 in areas 1, 2, 9, 11, 12, 15, 16, 18 and 26. Based on the highest observational coverage, three stations in the

southern North Sea (Fig. 1) have also been selected from this dataset, but not included in the statistical analysis per station (see

Appendix B): Walcheren (51.55º N, 3.41º E), Tersch100 (54.15º N, 4.3419º E) and EastCoastNS (55.25º N, 7º E).

### 2.2.3   Model skill evaluation

To evaluate the model skill two dimensionless parameters, the Pearson correlation coefficient ($r$) and a cost function ($CF$) are

used. Following Eilola et al. (2009), $CF$ is defined as:

$$CF = \frac{\sum_{i=1}^{n} \left| \frac{M_i - O_i}{std(O)} \right|}{n},$$

(1)

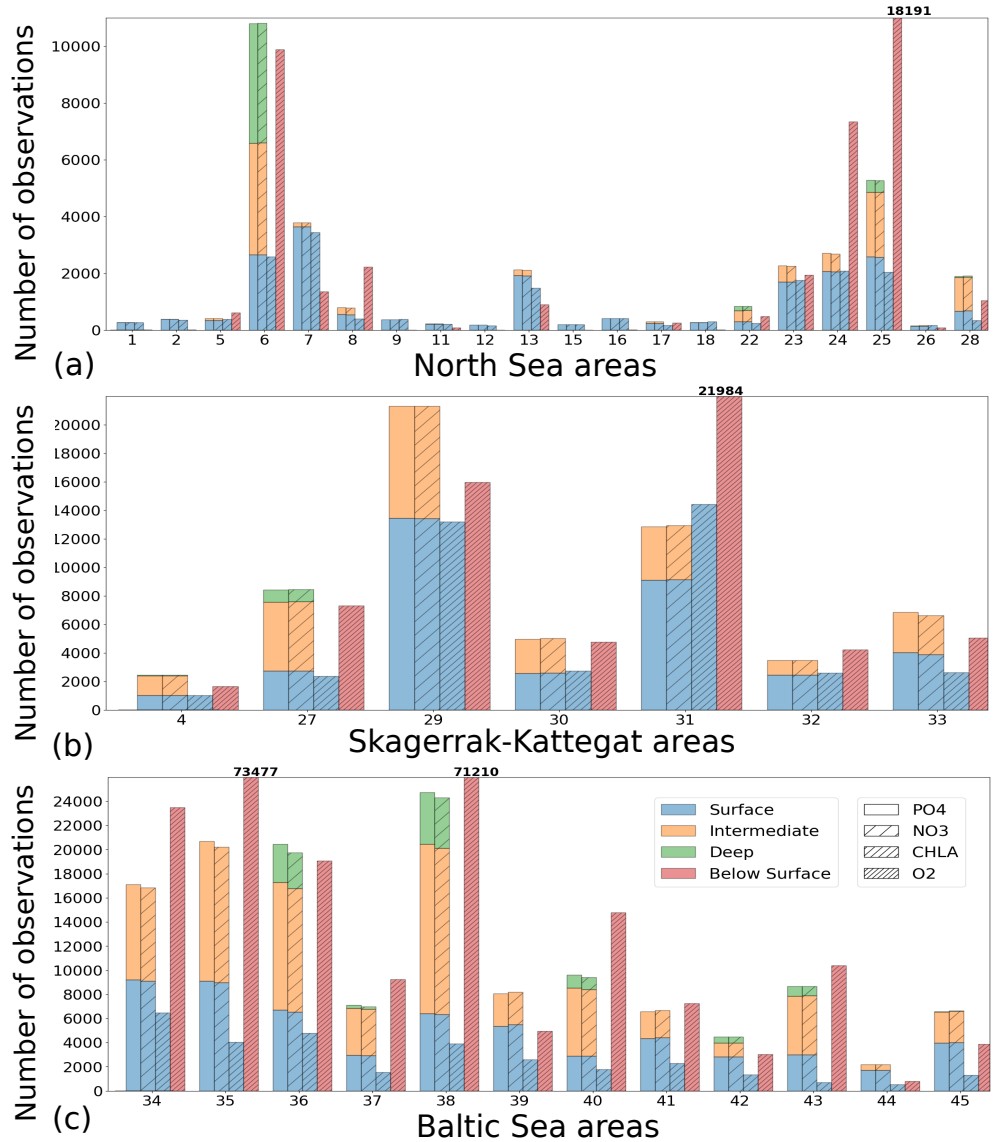

**Figure 5.** Number of observations per area according to Fig. 1 in a) the North Sea, b) the Skagerrak-Kattegat transition zone and c) the Baltic Sea. Observations are for phosphate (PO4), nitrate (NO3), chlorophyll-a (CHLA) and oxygen (O2) for the period 2001 to 2017 in surface (above 10 m), intermediate (in between 10 m and 100 m) and deep waters (below 100 m). For oxygen only observations below surface waters are considered (below 10 m). Areas with less than 100 number of observations for all four variables are not shown (i.e. areas 3, 10, 14, 19, 20 and 21 in the North Sea). Note that the y-axis scale in a), b) and c) differ.

where $i$ denotes the point in depth and/or time, $n$ is the number of data points and $M$ and $O$ are the model results and observations, respectively. The Pearson correlation coefficient provides information about how well variability in the observations is represented by the variability in the model. The cost function gives the proximity of the model to observations by normalizing the bias with the standard deviation of the observations. While a positive $r$ will always fall between 0 and 1, with 1 being a perfect fit, $CF$ will be below 1 only when model results fall within the standard deviation of the observations. Similarly to other

studies (e.g. Edman et al., 2018; Edman and Anderson, 2014), we combine both skill metrics as $1 - r$ vs $CF$. Thus, the closer values are to origin, the better the model performance is. More specifically, when values fall within an inner quarter circle with axis $(1 - r, CF) = (0.33, 1)$, the model performance is considered to be good. Model values are considered acceptable if they fall outside this inner quarter circle but within an outer quarter circle with axis $(1 - r, CF) = (0.66, 2)$. Large model biases are found when values fall outside the outer quarter circle.

This analysis is performed per station for the period 2001 to 2017 considering all days and depths. We select the stations with good salinity and temperature model skill to best evaluate the SCOBI performance and therefore, avoid biases from the ocean model. We also discarded stations with less than 500 observations for this analysis. The period mean and the seasonal model skill for phosphate, nitrate, chlorophyll-a and oxygen are then evaluated.

To get an overview of the spatio-temporal performance of NEMO-SCOBI, the model skill for the same biogeochemical

parameters is also evaluated for the entire domain as a whole and its sub-basins: The Baltic Sea, the Skagerrak-Kattegat transition zone and the North Sea. In order to have an overview on a finer regional scale, a model skill analysis per area was evaluated according to the latest assessment area definitions from HELCOM and OSPAR (Fig. 1). This evaluation is done for the entire water column and also for surface, intermediate and bottom waters. Areas with too little observations ($< 100$) for each variable during the period 2001 to 2017 are not evaluated.

## 3 Results and discussions

### 3.1 Validation per stations

In general, the long-time trends, the period means and to a lesser extent the seasonal cycle of all biogeochemical state variables are well captured by the model at all 27 stations. In addition, the interannual variability in the model is in good agreement with observations at most stations. Each station has local specificities both in observations and in model performance. However, the

310 model performance between stations shows large similarities depending on their proximity to one another. In this section, we present results of time series for surface and bottom waters as well as averaged-profile analysis from two stations: ANHOLT (Fig. 6 and Fig. 7) and BY15 (Fig. 8 and Fig. 9). These are here considered to represent the Skagerrak-Kattegat transition zone and the Baltic proper, respectively. This is because the model response at these two stations is very similar, both in magnitude and trend, to that at stations within their corresponding regions (i.e. the model response at ANHOLT is similar to

315 that at Å13, Å15, Å17, P2, SLÄGGÖ, FLADEN and W LANDSKRONA in the Skagerrak-Kattegat transition zone, and the model response at BY15 is similar to that at BY1, BY2, BY4, BY5, HANÖBUKTEN, BCSIII-10, BY10, BY20, BY29, BY31, BY32 and BY38 in the Baltic proper). In Appendix B, section B, we also show results from Å17 (Fig. B1) in the Skagerrak

and BY5 (Fig. B2) in the Bornholm Basin (Fig. 1) as further examples. Note that the model response at F9 is similar to that at F3 and C3, representing the Gulf of Bothnia. However, results from this area are not shown due to the lack of observations in this region. Together with the averaged-profiles, we show the corresponding observation coverage at all depths, which is never larger than 8% at all analyzed stations (e.g. Fig. 7 and Fig. 9) and highlights the low temporal resolution of observations (of a maximum of twice a month). Consequently, the probability of missing the monthly maximum (or minimum) in this data set is high and therefore, may not show the full variability range. Higher temporal resolution data sets exist (e.g. Rantajärvi et al., 1998; Greenwood et al., 2010), however, they only cover a few recent years and are not available for all biogeochemical parameters. Therefore, they cannot be used to analyze historic model results. When accounting for long-time series, the used observational data set becomes more reliable and more likely to be representative of the system. A summary of the regression analysis at ANHOLT and BY15 is listed in Table 2 and plotted together with the time series (Fig. 6 and Fig. 8).

**Table 2.** Summary of both model and observation trends for the period 1975 to 1995 and 1976 to 2017 at ANHOLT and BY15 for phosphate, nitrate, chlorophyll-a and oxygen, where the slope ($s$) and the number of observations (nobs) for surface and bottom waters are shown. Symbols "<", ">" and "−" denote negative, positive and no trend, respectively. No trend is here assumed when $s \leq \pm 0.001$. Significant trends - i.e. when p-values ($p$) are $\leq 0.05$ - are indicated in enlarged bold.

| Station | Variable | Period | $s$ Obs., model | $p$ Obs. | Model | nobs | $s$ Obs., Model | $p$ Obs. | Model | nobs |
|---|---|---|---|---|---|---|---|---|---|---|
| | | | Surface | | | | Bottom | | | |
| ANHOLT | PO4 | 1975-1995 | -0.005,0.005 | < | > | 171 | -0.006,0.001 | < | − | 122 |
| | PO4 | 1996-2017 | 0.001,-0.004 | − | < | 498 | -0.003,-0.01 | < | < | 473 |
| | NO3 | 1975-1995 | ,0.02 | | > | 50 | ,0.08 | | > | 26 |
| | NO3 | 1996-2017 | -0.01, -0.1 | < | < | 484 | -0.1,-0.1 | < | < | 459 |
| | CHLA | 1975-1995 | -0.2,0.04 | < | > | 83 | | | | |
| | CHLA | 1996-2017 | -0.02,-0.03 | < | < | 496 | | | | |
| | O2 | 1975-1995 | | | | | -0.03,0.006 | < | > | 167 |
| | O2 | 1996-2017 | | | | | 0.01,0.003 | > | > | 495 |
| BY15 | PO4 | 1975-1995 | 0.0001,0.02 | − | > | 104 | -0.1,-0.01 | < | < | 106 |
| | PO4 | 1996-2017 | 0.005,0.004 | > | > | 256 | -0.01,-0.1 | < | < | 254 |
| | NO3 | 1975-1995 | ,0.04 | | > | 32 | ,0.2 | | | 26 |
| | NO3 | 1996-2017 | -0.02,-0.05 | < | < | 250 | | | | 92 |
| | CHLA | 1975-1995 | -0.04,0.03 | < | > | 57 | | | | |
| | CHLA | 1996-2017 | -0.02,-0.05 | < | < | 254 | | | | |
| | O2 | 1975-1995 | | | | | 0.03,-0.03 | < | > | 103 |
| | O2 | 1996-2017 | | | | | 0.003,0.03 | > | > | 247 |

At ANHOLT, temperature, salinity and oxygen are very well captured by the model both in time and depth (Fig. 6a, Fig. 6b
and Fig. 6f). Phosphate also shows good agreement with observations at all depths, especially in surface waters, where it is
yearly depleted in both observations and the model (Fig. 6c). However, nitrate and chlorophyll-a are higher in the model than in
the observations (Fig. 6d and Fig. 6e). As in observations, the modeled nitrate at ANHOLT is depleted in surface waters, but the
yearly maxima are too high compared to observations. This results in a period-averaged positive bias of about 5 $\mathrm{mmol\,N\,m^{-3}}$
(Fig. 7f). In bottom waters, all model values for nitrate at ANHOLT fall outside the observation range. Note that the nitrate
bias is smaller at all stations in the Skagerrak (e.g. at Å17; Fig. B1, Appendix B) than those at ANHOLT and FLADEN (not
shown) in the Kattegat. Despite this bias, a decreasing nitrate trend after 1995 is seen in bottom waters in both observations
and model results (Fig. 6d and Table 2), suggesting that the long-time trend is still captured by the model. The model results
show an overall reduction of phosphate in the Skagerrak-Kattegat transition zone for the entire period (i.e. all stations in this
region show a negative trend with p-values smaller than 0.05 when evaluated for 1975 to 2017; not shown), in agreement
with that reported by Rasmussen and Gustafsson (2003) and Wulff and Stigebrandt (1989) after the 1980s in the Kattegat.
However, our model results suggest a significant increasing trend from 1975 until the 1990s for surface and bottom nitrate and
for surface phosphate at ANHOLT (Fig. 6c and Fig. 6d, Table 2). At ANHOLT the number of observations before 1996 is too
low, especially for nitrate, to be able to validate this trend. However, at the stations in the Skagerrak (Å13 and SLÄGGÖ),
where the observation coverage is better, such trends are not observed. In the model results, there is a significant decreasing
trend (p-value < 0.05) in both nutrients at stations in the Skagerrak after 1996 in both surface and bottom waters, but the trend
in surface waters is not statistically significant for observations (p-value > 0.05). The model trends in the Skagerrak-Kattegat
transition zone are strongly linked to the applied river forcing in this region which shows an increase in nutrients from the start
of the period followed by a decrease after the 1990s (Fig. 4). The poor observational coverage, especially before 1996, makes
it difficult to analyze trends in this region only based on observations.

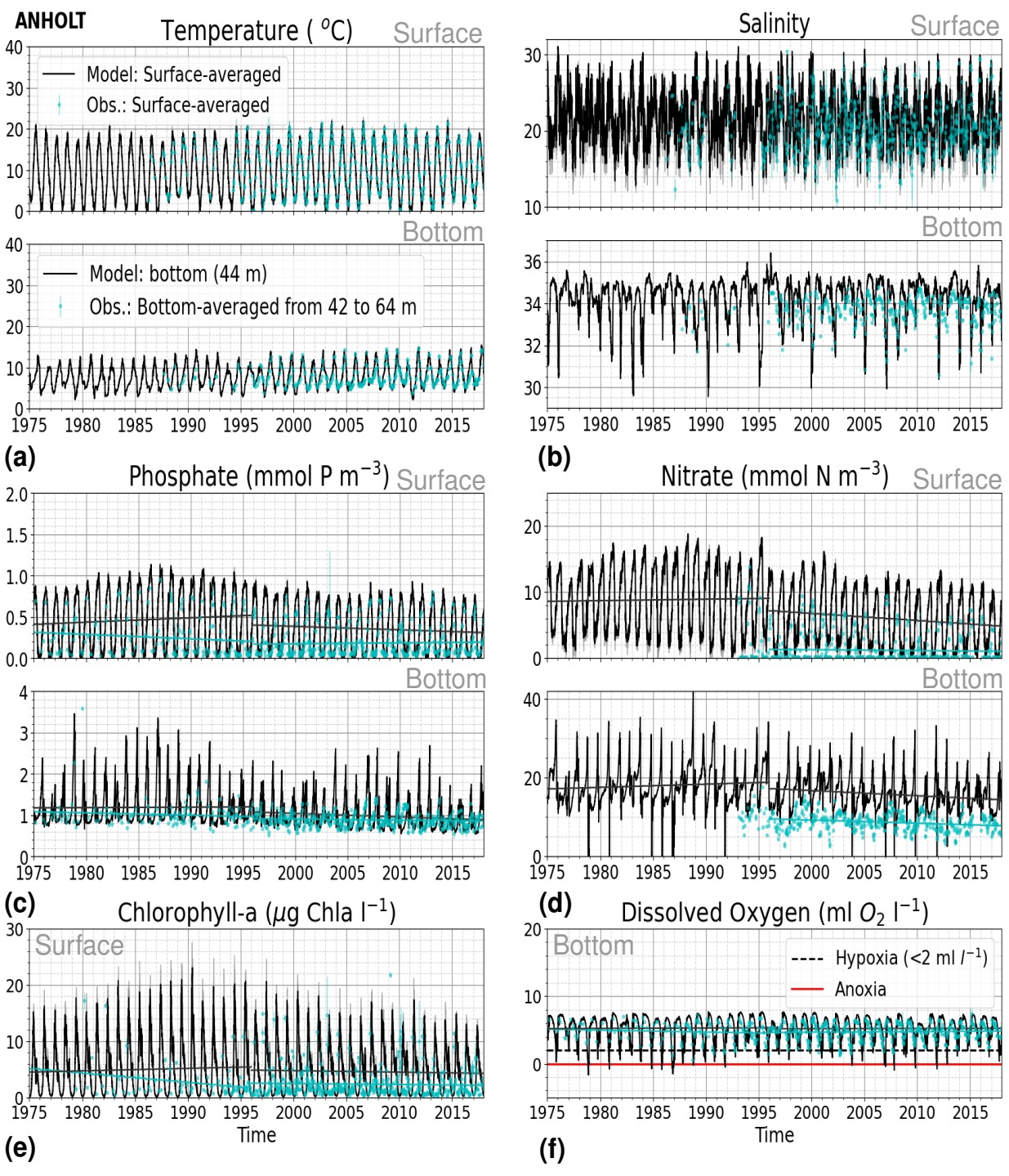

**Figure 6.** Time series of a) temperature, b) salinity, c) phosphate, d) nitrate, e) chlorophyll-a and f) bottom oxygen for model results versus

**Figure 6.** ... caption continuation ... observations in surface (averaged over the first $\sim 10$ m) and bottom waters for the period 1975 to 2017 at ANHOLT. A simple linear regression is shown for the periods 1975-1996 and 1996-2017 for both model results (black) and observations (blue) for phosphate, nitrate, chlorophyll-a and oxygen. The standard deviation for surface averages is shown for both model (gray) and observations (blue).

Similarly to findings by Carstensen and Conley (2004), the model shows a small, but significant increasing and decreasing trend in chlorophyll-a from 1975 to 1996 and towards 2017, respectively. These are not shown by the chlorophyll-a observations, which do not show a statistically significant slope (p-value > 0.5) from 1975 to 1996 and towards 2017 (Fig. 6e and Table 2). Observations of chlorophyll-a are lacking, especially before the year 1996. In addition, higher maxima of chlorophyll-a are more frequently displayed by the model than in the observations. Consequently, the model monthly averages (seasonal

cycle climatology) over the period 2001 to 2017 show higher values than observations (Fig. 7a). Besides the general low temporal resolution in observations, the chlorophyll-a distribution is usually patchy in the Baltic Sea (e.g. Pavelson et al. 1999; Janssen et al. 2004), thus difficult to measure *in situ*. Hence, these observations represent only a snapshot of nature and there are no guarantees that the measurements actually capture the chlorophyll peaks and may therefore not represent the full amplitude of interannual variability. The monthly averages at ANHOLT show a consistent peak in the observed chlorophyll-a in

late-winter/early-spring (February and March) with a later peak in Autumn (November), while the model only peaks in May, slowly decreases towards December. Because of this model delay during spring, the seasonal profiles for chlorophyll-a, nitrate and phosphate are less well represented by the model at this station, especially for summer (Fig. 7b, Fig. 7e and Fig. 7h). The period-mean nitrate and chlorophyll-a profiles show a consistent positive bias. However, the shape of the seasonal and mean period profiles are well captured by the model, especially for phosphate.

Even though there are biases in nitrate and chlorophyll-a, the monthly, the seasonal and the mean oxygen profiles are in good agreement with observations (Fig. 7j, Fig. 7k and Fig. 7l). Note that the overestimation of the modelled chlorophyll-a above the pycnocline during the summer months is mainly due to the model-delay in the phytoplankton growth. A positive oxygen bias of maximum $\sim 2 \, \mathrm{ml} \, \mathrm{O}_2 \, \mathrm{l}^{-1}$ is however observed at ANHOLT in bottom waters, especially for the summer months. This bias is much smaller at all other stations in the Skagerrak-Kattegat transition zone (e.g. Appendix B, Fig. B1j, Fig. B1k and Fig. B1l). Oxygen concentrations in bottom waters have been suggested to have declined in most basins in the Skagerrak-

Kattegat transition zone from 1971 to 1990 (Andersson, 1996). Model results for oxygen concentrations in bottom waters show no clear trends (with small regression slopes) at any Skagerrak-Kattegat transition zone stations (e.g. Fig. 6f and Table 2). In the Skagerrak-Kattegat, there is a tendency in both model and observations for an increase in oxygen with time after the 1990s, however, the trend is not statistically significant (p-value > 0.05). Thus, a long-term decline in oxygen followed by a recovery

is possible in the entire water column and may affect the overall oxygenation in the entire transition zone. In the Skagerrak-Kattegat region (e.g., at ANHOLT, Fig. 6) observational data is largely lacking before the 1990s, including that for oxygen in bottom waters and therefore model trends may be more representative of the system for historic values.

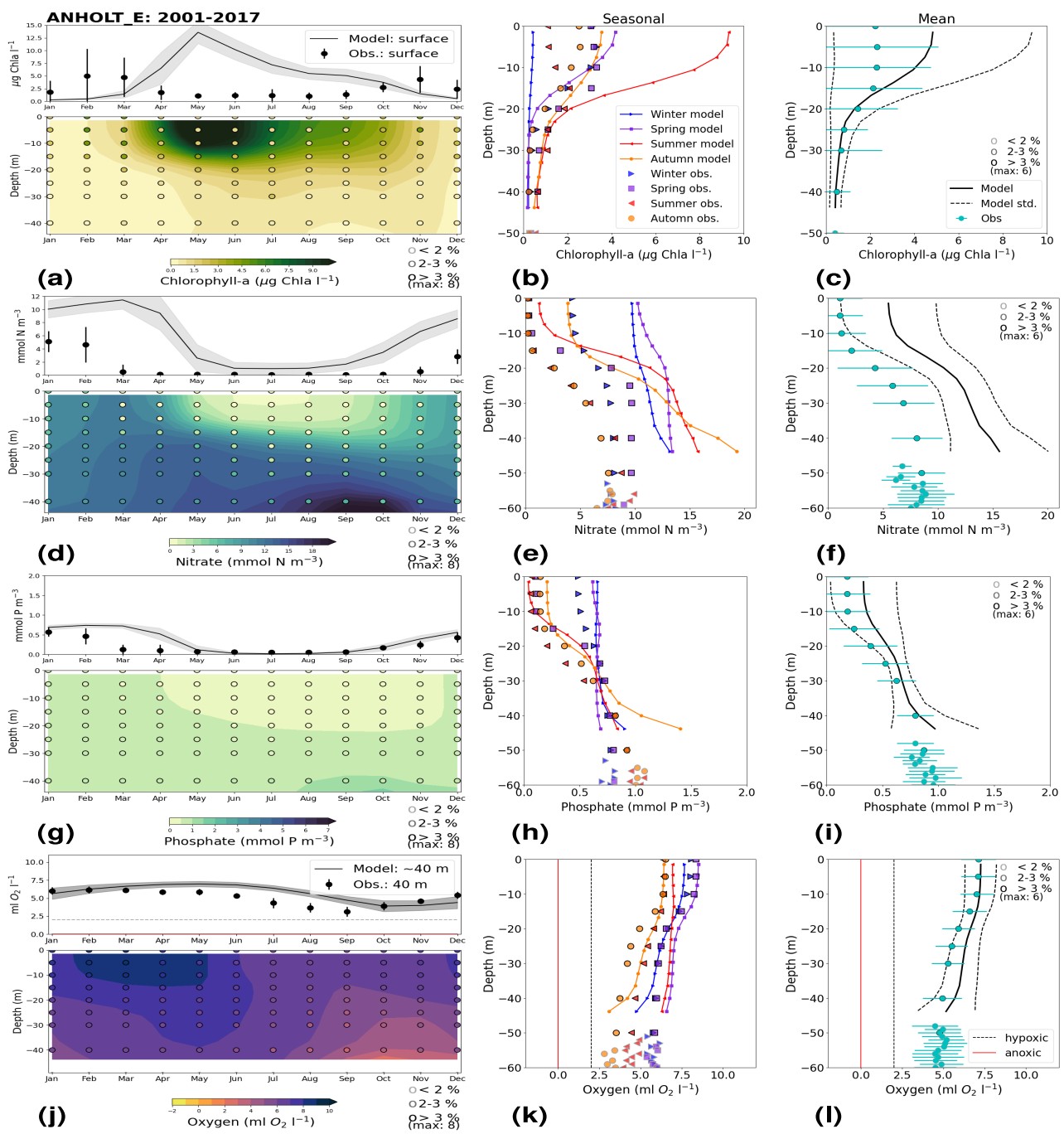

**Figure 7.** Monthly-, seasonal- and period-averages of the main biogeochemical variables at ANHOLT for 2001-2017. Variables are a-c) chlorophyll-a, d-f) nitrate, g-i) phosphate and j-l) dissolved oxygen for both model and observations. Monthly averages (a, d, g, and j) are shown over the entire

**Figure 7.** ... caption continuation ... water column (colors) and a close up for surface waters for all variables, except for dissolved oxygen where a close up of near bottom waters is shown instead. Near bottom is here considered to be the depth within the last model depth that has the most observations. The standard deviation in time for each averaged monthly value is shown for the model as a gray shaded area and as bars for the observations. The standard deviation of the period means (c, f, i and l) are also displayed for both model (dashed lines) and observations (cyan crosses). The observation coverage in all plots is shown as open symbols with shades of grays as indicated in the legend.

BY15: Baltic proper

At BY15, the temperature and surface salinity are well captured by the model (Fig. 8a and Fig. 8b). Importantly, the timing of
the inflows from the Skagerrak-Kattegat transition zone to the Baltic proper (reflected by the observed temperature and salinity peaks in bottom waters) are in good agreement with observations and previous findings (e.g. Gustafsson, 1997; Omstedt et al., 2004; Lass and Matthäus, 1996; Feistel et al., 2008; Hordoir et al., 2015). Modelled surface and bottom nitrate, as well as bottom phosphate follow well the observations at BY15, both in the long-time trends (Fig. 8c, Fig. 8d and Table 2) and in the interannual variability range (Fig. 9c and Fig. 9d). However, phosphate in surface waters is consistently overestimated by the
model (Fig. 8c), especially in late-spring and summer (with a positive bias of $\sim$0.5 $\mathrm{mmol\,P\,m^{-3}}$; Fig. 9g), and does not get yearly depleted due to an imbalance in the surface N to P ratios.

The model results suggest a small, but significant increasing trend in chlorophyll-a concentrations from 1975 to 1996 (Fig. 8e and Table 2). After this, the trend slowly decreases until the end of the period in both the model and observations. This period trend is visible at other stations in the Baltic proper with similar or better observation coverage (i.e. BY1, BY5 and
BY31), but p-values generally higher than 0.05. The increased trend in chlorophyll-a is in good agreement with a long-term increase in primary production for $\sim$1980 to 2004 found by Daewel and Schrum (2013). However, the model does not show equally high maxima in chlorophyll-a as those observed around 1995-2000 at BY15 (Fig. 8e). Such high chlorophyll-a values ($>$15 $\mathrm{\mu g\,chla\,l^{-1}}$) are rarely captured by *in situ* measurements, but are not uncommon when derived from satellite sensors which, for example, have shown summer patches with concentrations higher than 60 $\mathrm{\mu g\,chla\,l^{-1}}$ near the entrance of the
Gulf of Finland and the Southern Baltic proper (Reinart and Kutser, 2006; Dabuleviciene et al., 2020). This suggests that the model does not capture high growth rates of phytoplankton bloom in the Baltic proper, which may cause imbalances in the model N to P ratios. Due to such uncertainties, chlorophyll-a observations are here used more as an indication of how well the general patterns are reproduced by the model rather than as a quantitative parameter. In bottom waters, oxygen follows the salinity in bottom waters, where oxygen maxima coincide with salinity peaks. This indicates that the timing and intensity of
the oxygenation events at BY15 are well reproduced by the model (Fig. 8f). These inflows from the Kattegat bring oxygenated waters with high nitrate, but low phosphate concentrations that are also captured by the model (Fig. 8c and Fig. 8d).

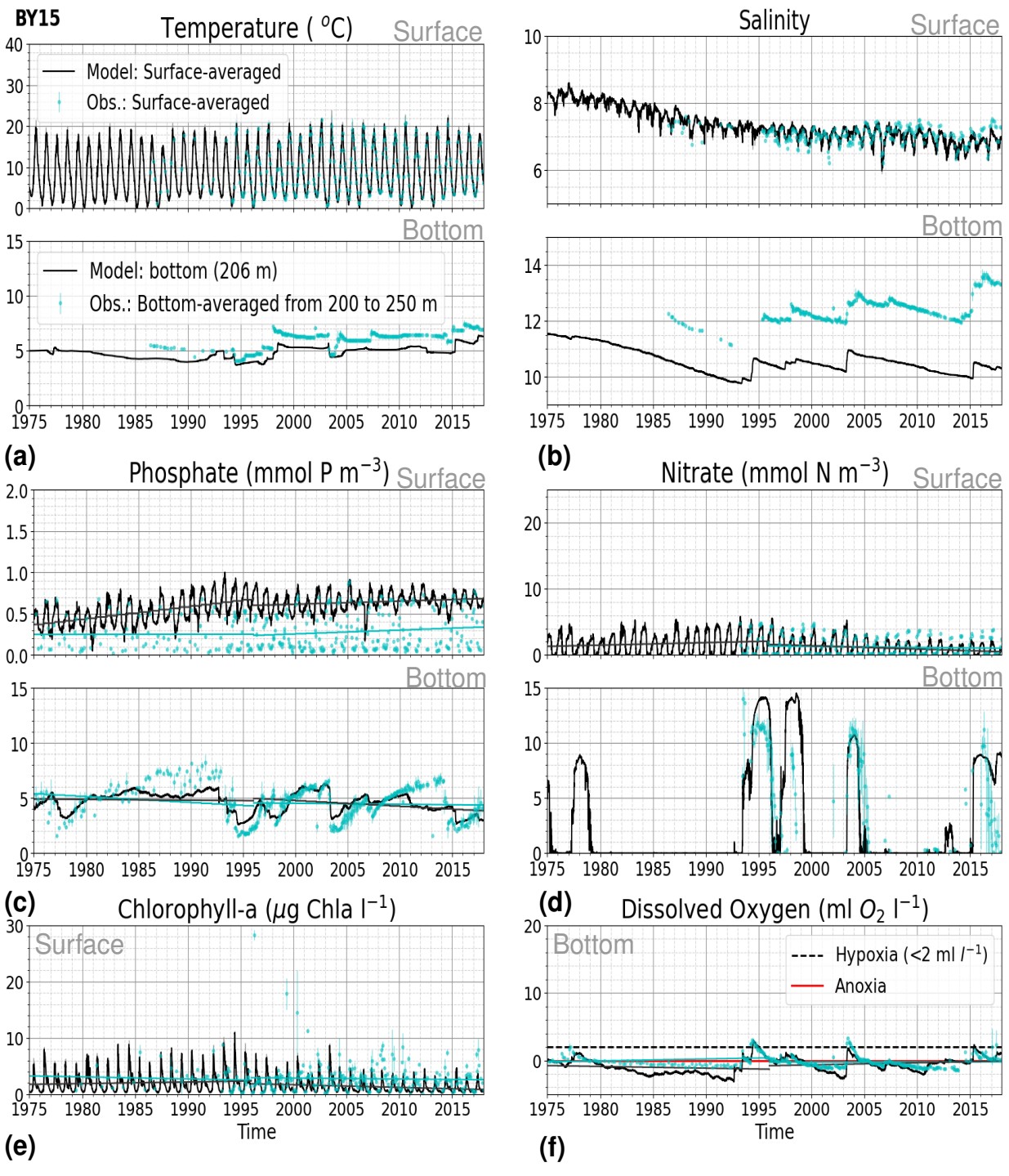

**Figure 8.** Time series of temperature, salinity and main biogeochemical parameters at BY15. Detailed description is as in Fig. 6

The monthly-, seasonally- and period-averaged chlorophyll-a and nitrate are well represented by the model at BY15 (Fig. 9a-c and Fig. 9d-f). The monthly-, seasonally- and period-averaged profiles of the modelled oxygen above the halocline and in bottom waters are also in good agreement with observations. However, the model shows an increasing bias with depth and time around the halocline ($\sim$75 m) in oxygen and nitrate. Applying river forcing, which includes daily instead of monthly runoff, significantly improved the surface salinity results in the Baltic Sea when compared to results in Hordoir et al. (2019), but increased the existing negative bias in intermediate and deep waters of the Baltic proper. As a result of this negative bias, the stratification in the Baltic proper is weaker in the model than in observations, with less saline, more nitrate-enriched and more oxygenated intermediate waters (Fig. 9d-f and Fig. 9j-l). The positive oxygen bias (of max. $3 \, \mathrm{ml} \, O_2 \, l^{-1}$) in intermediate waters is only found at the deeper stations (e.g. BY10, BY15, BY20). This bias also decreases with depth and it is linked to the salinity bias that brings less saline and more oxygenated inflows to the Baltic proper. The more oxygenated intermediate waters lead to less denitrification and therefore, more nitrate, explaining the overestimation of nitrate at these depths ($\sim$75-150 m).

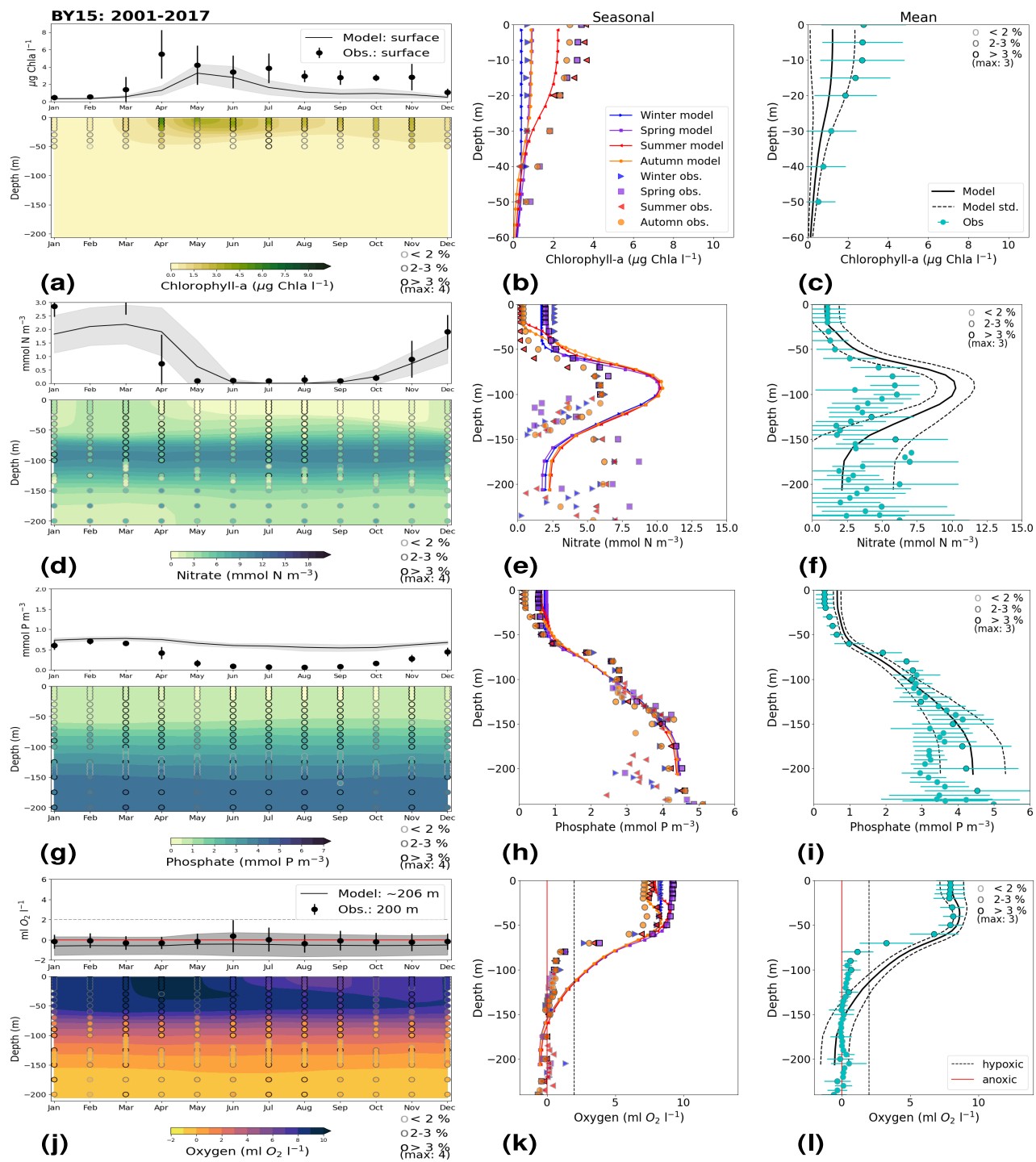

**Figure 9.** Monthly-, seasonal- and period-averages of the main biogeochemical variables at BY15. Detailed description is as in Fig. 7

## 3.2 Model skill at selected stations

The model skill for phosphate, nitrate, chlorophyll-a and oxygen for the period 2001-2017, as well as the seasonal model
skill, is analyzed at 14 stations distributed in the Skagerrak-Kattegat transition zone, the Bornholm Basin, the Western Gotland
Basin, the Eastern Gotland Basin and the Northern Baltic Proper. All these stations show good model skill for both the period
and the seasons when evaluated for temperature and salinity (Fig. 10a and Fig. 10b) and therefore $CF$ and $1 - r$ values for the
biogeochemical variables are mainly representative of the SCOBI model performance.

The phosphate model skill for the entire period at all evaluated stations has low enough $CF$ and $1 - r$ values to be placed
within the outer quarter circle in the model skill figure, with most stations falling within the inner circle (Fig. 10c, black
markers). More specifically, all stations located in the Baltic proper show good phosphate model skill as well as most stations
in the Skagerrak-Kattegat transition zone, except SLÄGGÖ, Å13, FLADEN and W LANDSKRONA, which show acceptable
model skill. The model skill to reproduce seasonal phosphate is scattered, but with $CF$ values lower or close to 1 and combined
values of $CF$ and $1 - r$ mainly falling within the inner and outer quarter circles. The latter indicates good or acceptable
performance, respectively. However, at ANHOLT, FLADEN and SLÄGGÖ, $CF$ and $1 - r$ values for winter and spring fall
outside the outer circle (Fig. 10c, blue and purple markers).

Despite the positive bias in nitrate in the Skagerrak-Kattegat transition zone, the model shows acceptable, mainly good,
performance for nitrate when evaluated for the entire period at all 14 stations (Fig. 10d, black markers), except at BY29.
The seasonal model performance for nitrate is less well reproduced than that for phosphate, however, most stations still show
acceptable seasonal performance, except at BY29, ANHOLT, Å13, FLADEN and SLÄGGÖ for all seasons, Å15 for winter
and spring and BY20 for winter (Fig. 10d, color markers). For the stations in the Skagerrak-Kattegat transition zone, this is
due to the time delay in phytoplankton bloom, which shifts the seasonal cycle in the model (e.g. Fig. 7a-f and Fig. B1a-f) and
contributes to the positive nitrate bias. This bias is confined above the mixed layer depth, therefore mainly affecting most of
the water column at shallow stations. Because the model considers an averaged depth within a grid cell, the maximum depth of
the model and the observations differ. The difference is considerable at shallow stations (with depths less than 100 m), namely
SLÄGGÖ, Å13, ANHOLT, FLADEN and W LANDSKRONA. Observations deeper than the maximum model depth are thus
not considered in this evaluation, which affect our results as nitrate below the mixed layer depth is better captured by the model
in this region. At BY29, the poor model skill comes from the bias below the mixed layer depth linked to the ocean model
salinity bias in the Baltic proper. At this station, the maximum model depth is $\sim$160 m, while observations go as deep as 180 m
(not shown). Unlike all the other deeper stations in the Baltic Sea (e.g. Fig.B2), the nitrate bias at BY29 also decreases with
depth but remains positive below $\sim$70 m, resulting in high $CF$ and $1 - r$. BY29 has not been used for model validation in
previous studies, however, BY15 and a nearby station (BY31) in the RCO-SCOBI model show similar results for a different
time period (1969-1998), where nitrate $CF$ values are high ($>$1) below 100 m (Eilola et al., 2009). This suggests that SCOBI
still struggles to reproduce nitrate concentrations in intermediate waters of the Baltic proper.

In the Baltic proper, good model skill for chlorophyll-a is found at six stations (i.e. BY4, BSCIII-10, HANÖBUKTEN,
BY10, BY15 and BY20), which also show good or acceptable model skill for both nitrate and phosphate. The model skill to

reproduce oxygen concentrations both seasonally and for the entire period is good at all 14 stations. This is also the case at the other stations in the Baltic Sea (Appendix B, table B1). To a lesser extent, the other biogeochemical variables also show good or acceptable model skill at many of these other stations, especially for phosphorus. The main inference of this analysis is

that model results at stations in the Skagerrak-Kattegat transition zone and Baltic proper are good, except at stations where the model delay in phytoplankton bloom or the positive nitrate bias due to a small oxygen positive bias affects most of the water column. Despite such specificities of the NEMO-SCOBI model, the analysis for individual stations is in good agreement with previous station results in other models (e.g. Eilola et al., 2009; Daewel and Schrum, 2013; Maar et al., 2011).

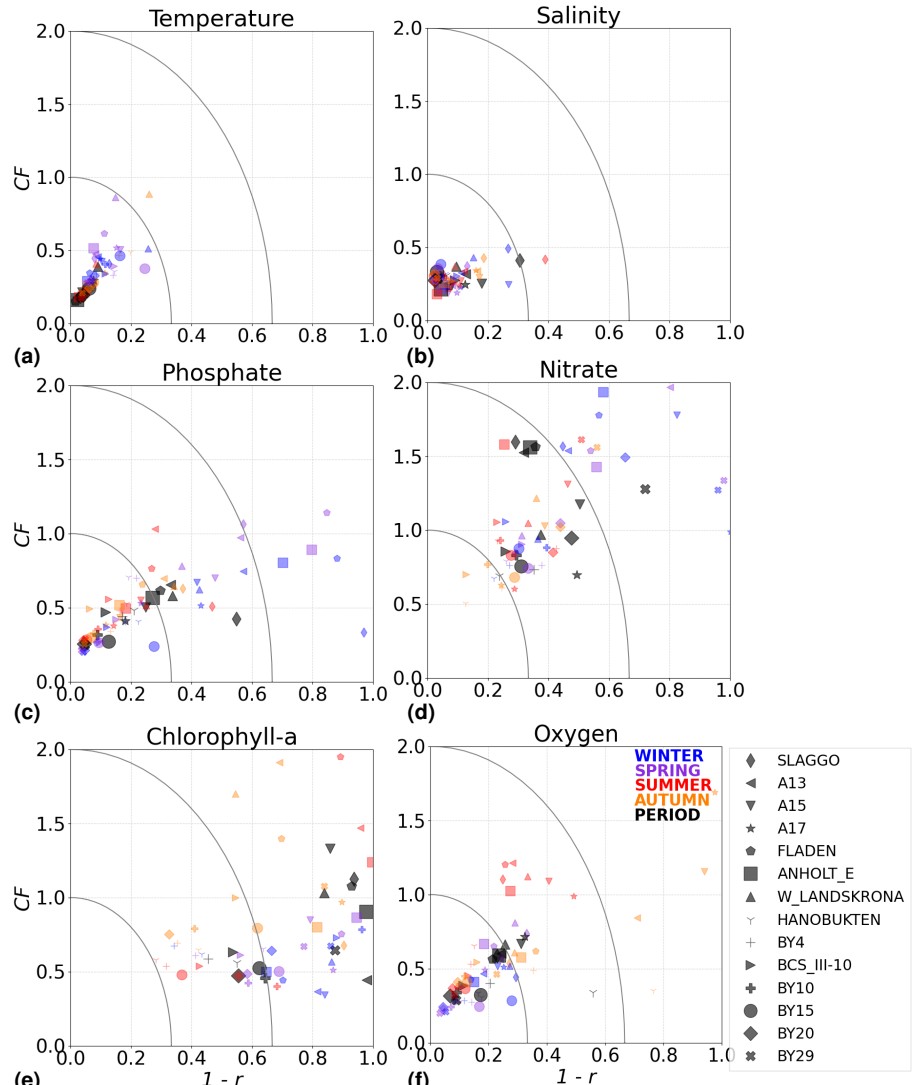

**Figure 10.** Model skill for the period 2001 to 2017 shown as a combination of Pearson correlation bias (1 - $r$) and Cost Function bias ($CF$) for selected stations in the Skagerrak-Kattegat area and the Baltic Sea. Shown biases are for the whole period (black) and for each season (colors) for water-column a) temperature, b) salinity, c) phosphate, d) nitrate, e) chlorophyll-a and f) oxygen. Winter months are from December to February, spring months are from March to April, summer months are from May to August and autumn months are from September to November. Markers within the inner quarter circle indicate good model skill, markers inside the outer circle indicate that the model skill is acceptable and markers outside the quarter circles indicate large biases. Stations ANHOLT and BY15 are highlighted with larger marks.

### 3.3 Model skill on regional scales

Physical and biogeochemical processes do not follow political or clear defined borders. However, the applied HELCOM-OSPAR assessment areas likely represent major features of the regional ecosystems as they are defined according to geography, bathymetry and stratification, notably that of OSPAR which follows stratification patterns described in van Leeuwen et al. (2015) and Capuzzo et al. (2013, 2015). We use these areas to evaluate the temporal and spatial model skill and with this identify dominant model regional features at a sub-basin and finer scales where possible. In general, this analysis shows that the biogeochemical parameter that is best captured by the model is oxygen, followed by phosphate, nitrate and then chlorophyll-a. The combined $CF$ and $1 - r$, especially when evaluated per HELCOM-OSPAR assessment area show, however, large scatter for all biogeochemical parameters (Fig. 11). Notably, phosphate and nitrate show variable model skill, which indicates a strong spatial-specific model response to the ocean dynamics and the applied physical and biogeochemical forcing. Below surface waters (Fig. B5), the model skill for phosphate and nitrate show higher $1 - r$ and $CF$ in most areas, suggesting that the scattered results are also partly linked to the distribution and frequency of observations. Indeed, observations are not homogeneously distributed in space and time (e.g. Fig. 12 and Fig. 13) and are seriously lacking in several areas, especially in the North Sea where phosphate and nitrate observations are mainly confined to the surface waters (Fig. 5) and only densely measured in near-coastal regions (van Leeuwen et al., 2023).

More specifically, oxygen shows good or acceptable model skill when evaluated for the entire domain and all its sub-basins (Fig. 11d, diamonds). Phosphate and nitrate show good and acceptable model skill, respectively, when evaluated for the entire domain, but the skills differ for each sub-basin (Fig. 11a and Fig. 11b, diamonds). Phosphate shows good and acceptable model skill in the Baltic Sea and the Skagerrak-Kattegat transition zone, respectively, in good agreement with previous North Sea-Baltic Sea 3D ocean models (Daewel and Schrum, 2013; Maar et al., 2011), but poor skill in the North Sea. The latter is linked to a mainly negative bias of winter phosphate in most surface waters of the North Sea (Fig. 12a). Nonetheless, 10 out of the 19 evaluated areas in the North Sea (areas 1, 5, 6, 8, 9, 11, 13, 23, 25 and 26; Fig. 11a, circles) show good or acceptable model skill for phosphate. Phosphate biases through all seasons are also found in surface waters of the Skagerrak-Kattegat transition zone and the Baltic proper (Fig. 12a). However, both these regions have been more frequently measured than the North Sea, contributing to better and more robust results of the combined $CF$ and $1 - r$ evaluation. The only areas that plot outside the outer circle in the Baltic Sea and the Skagerrak-Kattegat transition zone are the Gulf of Riga and the Bay of Mecklenburg, respectively (areas 33 and 39). Our results give better cost function results for phosphorus and oxygen in the Baltic Sea (Fig. 11, green diamonds) than those reported from a model ensemble using ERGOM, BALTSEM and RCO-SCOBI (Eilola et al., 2011a), but worse model skill for nitrate.

On the other hand, nitrate is better captured than phosphate when evaluated for the entire North Sea, but shows poor skill in the Skagerrak-Kattegat transition zone and the Baltic Sea (Fig. 11b, diamonds). In the Skagerrak-Kattegat transition zone, the nitrate bias is linked to the model underestimation of phytoplankton growth and the delay in its monthly maximum. Nevertheless, both the Baltic Sea and the Skagerrak-Kattegat transition zone have a $1 - r$ and a $CF$ close to 0.66 and 1, respectively, indicating similarities between model and observations in both variability and averages. In the Skagerrak-Kattegat transition

zone, most HELCOM-OSPAR assessment areas show acceptable nitrate model skill (Fig. 11b, circles), except in the Sound and in the Kiel Bay (areas 31 and 32, which plot outside the outer circle). Both these areas have narrow straits and complex land features that likely prevent the model from capturing their full local dynamics. In the Baltic Sea, four HELCOM assessment areas (areas 35, 37, 38 and 43 in Fig 1) show acceptable model skill, which includes areas where intermediate waters showed a positive nitrate bias at individual stations (e.g. BY5, BY38 and BY15 in the Bornholm Basin, the Western Gotland Basin and the Eastern Gotland basins; areas 35, 37, 38 in Fig. 11b). This suggests that the model overestimation of nitrate in intermediate waters together with that in surface waters shown at stations in the Skagerrak-Kattegat transition zone mainly affects the model skill in the Arkona basin (area 34), which plots outside the outer circle when evaluated at all depths (Fig 11b) and in surface and intermediate waters (Fig. B5b and Fig. B5c). However, model biases in surface waters for nitrate in the Arkona Basin are mainly found in winter and spring and along the southern coast (northern Germany and Poland; Fig. B6b).

Also, in the Northern Baltic Proper, the Gdansk Basin, the Gulf of Riga, the Gulf of Finland, the Åland Sea, the Quark and the Bothnian Bay the model skill for nitrate is less good. In the Northern Baltic Proper surface model bias is mainly confined along the coast and near the entrance of the Gulf of Finland, resulting in acceptable model skill for surface waters in this area (Fig. B6b and Fig. 11b). In the Bothnian Sea, intermediate waters give acceptable model skill for nitrate (Fig. B5c), but poor skill in surface waters (Fig. B5b). All other northern basins in the Baltic Sea show poor nitrate skill in intermediate waters. This suggests that the nitrate bias in the model for the northern basins of the Baltic Sea is not linked to that of the Skagerrak-Kattegat transition zone and the Baltic proper, but rather linked to specific regional inputs and local dynamics. The applied atmospheric nitrate input in the Baltic Sea, especially for years before 1995, may be overestimated here. In this case, nitrate concentrations in areas where phytoplankton is limited by phosphate could slowly increase with time due to non-yearly depletion. A small overestimation of nitrate in such areas (such as in the Gulf of Bothnia) would also lead to nitrate accumulation in these areas and explain the positive bias. Our results compare well to the $CF$ results for surface waters in Maar et al. (2011), where their highest values are also found near coastal areas in the Baltic proper and the Bothnian Sea for both phosphate and nitrate. Note that in their study, observations for the Gdansk Basin, the Gulf of Riga, the Gulf of Finland and the Bothnian Bay are missing. In fact, except for the Northern Baltic Proper and the Gdansk Basin, the Gulf of Riga, the Gulf of Finland, the Åland Sea, the Quark and the Bothnian Bay are the most poorly measured of the Baltic Sea, which prevents robust statistical evaluations in these areas.

On a finer regional North Sea scale, the model skill for nitrate is acceptable in several shallow areas in the southern and eastern part of the North Sea (areas 1, 7, 9, 11, 22 and 26), the Dogger Bank (area 5) and the Norwegian trench (area 6). While the Northern North Sea (area 25) plots far outside the outer circle, the Eastern North Sea (area 24) plots near the outside circle (with $CF$ close to 1.25 and 1 - $r$ close to 0.6), suggesting that areas less affected by the northern open boundary and direct riverine nutrient input (i. e. southern offshore areas) are best represented by the model. In addition, the Northern North Sea (area 25 in Fig. 1) is the largest assessment area of the North Sea and the least spatially measured regarding phosphate and nitrate (Fig. 12 and Fig. 13), making it difficult to analyze specific model flows in this area. Especially, since the nutrient bias in surface waters in the Northern North Sea is small (Fig. B6). Nonetheless, our biogeochemical results are comparable to those of Daewel and Schrum (2013), which is in overall good agreement with observations in the North Sea but with biases

in the southern coast of the North Sea. This region is characterized by strong tidal currents (Van der Molen, 2002) and is heavily impacted by runoff and nutrient input that results in a high spatio-temporal variability. This remains a challenge for most models to recreate as also highlighted in a recent model ensemble study (van Leeuwen et al., 2023).

The best model skill for Chlorophyll-a is in the North Sea (with a $CF$ lower than 1 and a 1 - $r$ slightly higher than 0.66), within which five of the evaluated HELCOM-OSPAR assessment areas show acceptable model skill (areas 1, 2, 9, 12 and 18 Fig. 11c, circles). These areas, located in the southern North Sea, are coastal well mixed areas (van Leeuwen et al., 2015) where phytoplankton is mainly N limited in spring (both shown in observations and model results; Fig. 14). This suggests that the model is able to reproduce chlorophyll-a even in regions where important nutrient discrepancies between model and observations exist as long as the limiting nutrient for phytoplankton growth in the model corresponds to that in observations. Small model imbalances between nitrate and phosphate can greatly affect the chlorophyll-a production, especially when the stoichiometry is fixed to the Redfield ratio (Fransner et al., 2018; Neumann et al., 2022). This could lead to large bias in chlorophyll-a through all seasons in many areas of the model domain (Appendix B, Fig. B6c). Oxygen measurements for the five areas where the model skill for chlorophyll-a is best (areas 1, 2, 9, 12 and 18 Fig. 11c, circles) are lacking, but all evaluated areas in the North Sea (i.e. with more than 100 observations; Fig. 5) show good model skill for oxygen below surface, except the Northern North Sea (area 25; Fig. 11, circles). The Northern North Sea is directly influenced by the open boundary conditions in the model. Applying simplified physical dynamics and biogeochemistry in the open boundaries likely limits our model results in this area.

The model skill analysis on a regional scale shows that the model response varies widely, giving an overall acceptable model skill for oxygen, nutrients and chlorophyll-a that are comparable to other model studies. Regional discrepancies between model and observations exist which are difficult to explain due to complex physical and biogeochemical interactions, but three potential causes could be identified affecting different regions: the local nutrient input (via rivers or atmosphere), the applied open boundaries and a missing process in the phytoplankton growth that delays the peak of the bloom in the model (further discussed in section 3.6). The spatio-temporal lack of observations greatly hinders a more detailed understanding of the system. Thus, larger regional scale processes are the main focus of the next section.

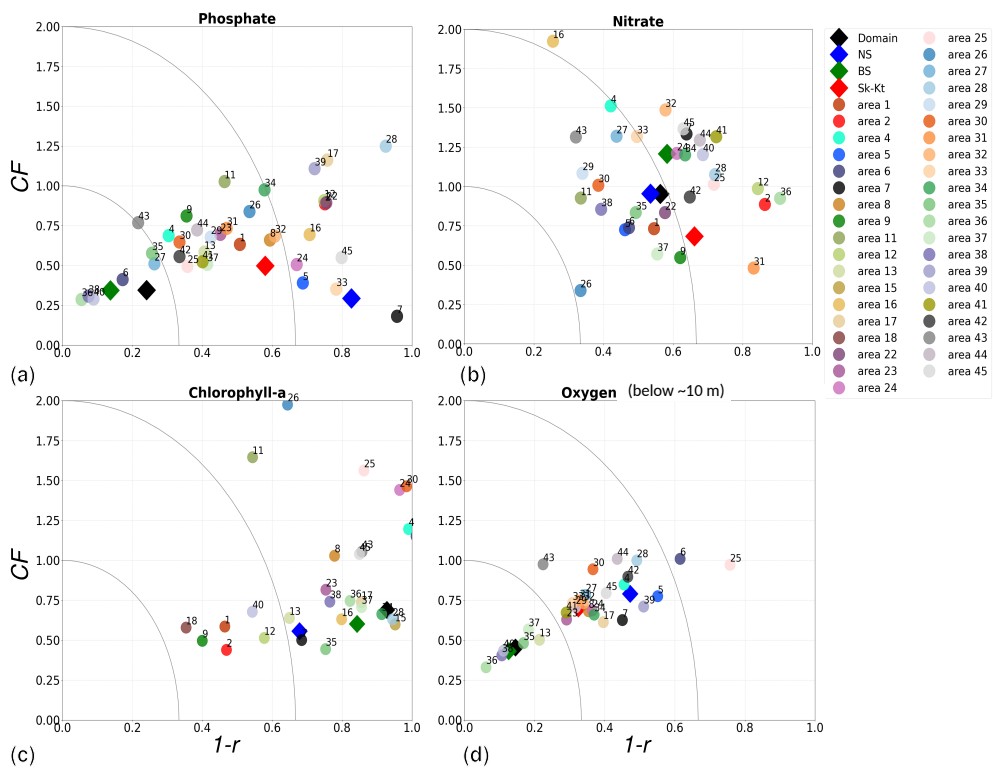

**Figure 11.** Model performance over the period 2001 to 2017 shown as a combination of Pearson correlation bias (as $1 - r$) and Cost Function bias (CF) for the Baltic Sea (BS), North Sea (NS) and Skagerrak-Kattegat (Sk-Kt) transition zone (diamonds) and areas in Fig. 1 (circles). Numbers in legend correspond to evaluated areas, note that for oxygen additional areas are not evaluated due to lack of observations (see section 2.2.3 and Fig. 5). Chlorophyll-a is evaluated for the top $\sim$10 m and the oxygen below surface waters.

## 3.4 The Baltic Sea-North Sea system

The coastal zone along France, Belgium, the Netherlands, Germany, and Denmark, is characterized by nitrate accumulation which reflects phosphate limitation of primary productivity in this region (Fig. 12, Fig. 13, Fig. 14 and Appendix B; Fig. B3 and Fig. B4). To a lesser extent, phosphate also accumulates close to the shoreline along the southeastern North Sea likely because productivity rates there are not high enough to fully consume the massive nutrient input by rivers, as seen both in observations and model results. Productivity in this region is likely limited by light (Holt et al., 2012). The nitrogen accumulation pattern is promoted by the reduction of phosphorus loads in the early 90s (Burson et al. e.g. 2016; Ly et al. e.g. 2014) and well captured by the model (Fig. 12, Fig. 13 and Fig. 14). The main river sources in this area are the Rhine, Elbe, Scheldt and Meuse plumes, where elevated nitrogen is yearly discharged (Fig. 3). In the northern North Sea, the pattern is inversed: almost full consumption of nitrate takes place in spring/summer while phosphate is preserved throughout the year. Phosphorus sources are mainly from point sources, from input via rivers and from advected water masses coming from the North Atlantic, which are enriched in nutrients due to mixing and internal wave generation along the shelf break (Gröger et al., 2013; Mathis et al., 2019; Huthnance et al., 2022).

High riverine nutrient inputs and shallow water depths promote high chlorophyll-a concentrations along the coasts (Holt et al., 2012). The elevated chlorophyll-a concentrations along the eastern UK coast have been shown to be likely related to frequent upwelling events under a predominant westerly wind regime (Winther and Johannessen, 2006). Towards the open ocean, chlorophyll-a concentrations tend to decrease. In particular, autumn and winter mixing in the central North Sea (van Leeuwen et al., 2015) distributes chlorophyll-a deep along the water column. The Norwegian coastal current is characterized by mesoscale meanders and eddies (Ikeda et al., 1989). Mesoscale eddies are also produced along the opposing currents of the northward flowing Norwegian coastal current and the southward flowing water masses entering the North Sea at the western slope of the Norwegian Trench (Winther and Johannessen, 2006). Hence, nutrients from deep waters can be mixed upwards, especially in winter and spring. When nutrients enter the stratified waters of the Norwegian coastal current, chlorophyll-a concentrations increase in this region. During autumn, the chlorophyll-a concentrations decrease due to increased vertical mixing caused by strong winds (Sündermann and Pohlmann, 2011). All these chlorophyll-a main features can be clearly seen in the chlorophyll-a maps for spring, summer and autumn, in both model results and observations (Fig. 12 and Fig. 13), indicating that the model is able to reproduce these North Sea characteristics. In addition, the spatial distribution of both nutrients and the N to P ratios are well captured by the model, which show important persistent gradients in the entire domain (Fig. 12, Fig. 13, Fig. 14 and Appendix B; Fig. B3 and Fig. B4). In particular, strong nutrient gradients are observed in the Skagerrak-Kattegat transition zone, which is in good agreement with previous findings for the Skagerrak (Danielssen et al., 1997). However, a consistent positive bias in nitrate occurs in the coastal southeastern North Sea, in the Skagerrak-Kattegat transition zone (as described before at individual stations in this region), near the Szczeciński Lagoon (Poland) and in the Gulfs of Gdańsk, Riga, Finland and Bothnia (Appendix B, Fig. B6). In the Skagerrak, the observations show an overall nitrogen limitation except during the spring months (Fig. 14b). In this area, the model remains phosphorus limited in all seasons due to the positive nitrogen bias (Fig. 14a).

In the more stratified Baltic Sea, nutrients and chlorophyll-a concentrations are spatially more homogeneous. Unlike in the North Sea, the high chlorophyll-a concentrations in the Baltic Sea are confined to the coasts (Fig. 12 and Fig. 13) due to limited occurrence of mesoscale turbulence and in turn poor mixing in the open Baltic Sea (Feistel et al., 2008). Both these physical features are well represented in the model (Hordoir et al., 2019). The open Baltic Sea is nutrient-fueled by the direct nutrient input along the coasts and by nutrients accumulated in sea water and sediments during winter. The nutrient inventory decreases in surface waters due to consumption by phytoplankton and export of sinking organic matter during the growth season, which spans from late-winter/early-spring to late-summer/autumn (Fig. 13), varying according to the region and between the open and coastal ocean. In the Baltic proper, primary productivity is limited by nitrate (clearly seen in Fig. 14) linked to high removal rates of nitrogen via denitrification and high release rates of inorganic phosphorus from the sediments (Eilola et al., 2009). This favours cyanobacteria blooms under elevated temperatures and reduced vertical mixing during (late-)summer (Janssen et al., 2004). In the model, the cyanobacteria bloom starts in summer but becomes only widespread in autumn (Appendix B, Fig. B7). This could be linked to a model overestimation of light attenuation in the open ocean (Appendix B, Fig. B8) limiting cyanobacteria growth in the summer months. In addition, explicitly considering the life cycle of cyanobacteria would significantly improve the timing of the growth of cyanobacteria (Hense and Burchard, 2010; Hieronymus et al., 2021). The cyanobacteria response likely affects the entire phytoplankton growth season in the model, which currently is generally underestimated in the open Baltic Sea (Fig. B6, bottom panel) and starts one month later compared to observations (Fig. 9, top panel). Hence, nutrient depletion is also perturbed in the model results. In the Bothnian Sea and Bothnian Bay, the seasonal cycle of nitrate is not well captured which is common in models using a fixed Redfield ratio (Fransner et al., 2018; Neumann et al., 2022).

In the southeastern coastal North Sea and in the Skagerrak-Kattegat transition zone the model delay in phytoplankton bloom is about three months. According to observations, chlorophyll-a concentrations in both these areas peak in late winter (around February), while in the model the maximum occurs in May. Here, the model late-winter/early-spring primary productivity is neither limited by nitrogen, phosphate nor silicate. In addition, the maximum growth rates were adjusted to favour diatoms under lower temperatures and high nutrient concentrations (Appendix A, section A1, Eq. A7 and Table A1). However, this is not sufficient to capture the correct timing of the phytoplankton bloom in the Skagerrak-Kattegat transition zone. This suggests that an additional limitation factor affects both the southern coastal North Sea and the Skagerrak-Kattegat transition zone. The model sensitivity of total phytoplankton growth rates to temperature and light attenuation is shown in Appendix A, Fig. A1 where nutrients, detritus, and zooplankton concentrations are kept constant at typical model values for February in the Kattegat. The model does not allow for high phytoplankton growth rates (or high chlorophyll-a concentration) at low temperatures and low light attenuation (high secchi depths). Indeed, the modeled temperature at ANHOLT is low (2.5 $C^o$), the secchi depth high (6.8 m) and the chlorophyll-a concentrations low ($\sim$1 $\mu g\,ChlaL^{-1}$). Because the model temperature for February in surface waters when averaged from 2001 to 2017 is very close to that shown by the observations (2.8 $C^o$), we attribute the phytoplankton bloom delay mainly to a too low sensitivity of phytoplankton growth to light in this region (Fig. A1, Appendix A). The delay in the model causes interannual nutrient imbalances that can be transported, for example along the Jutland

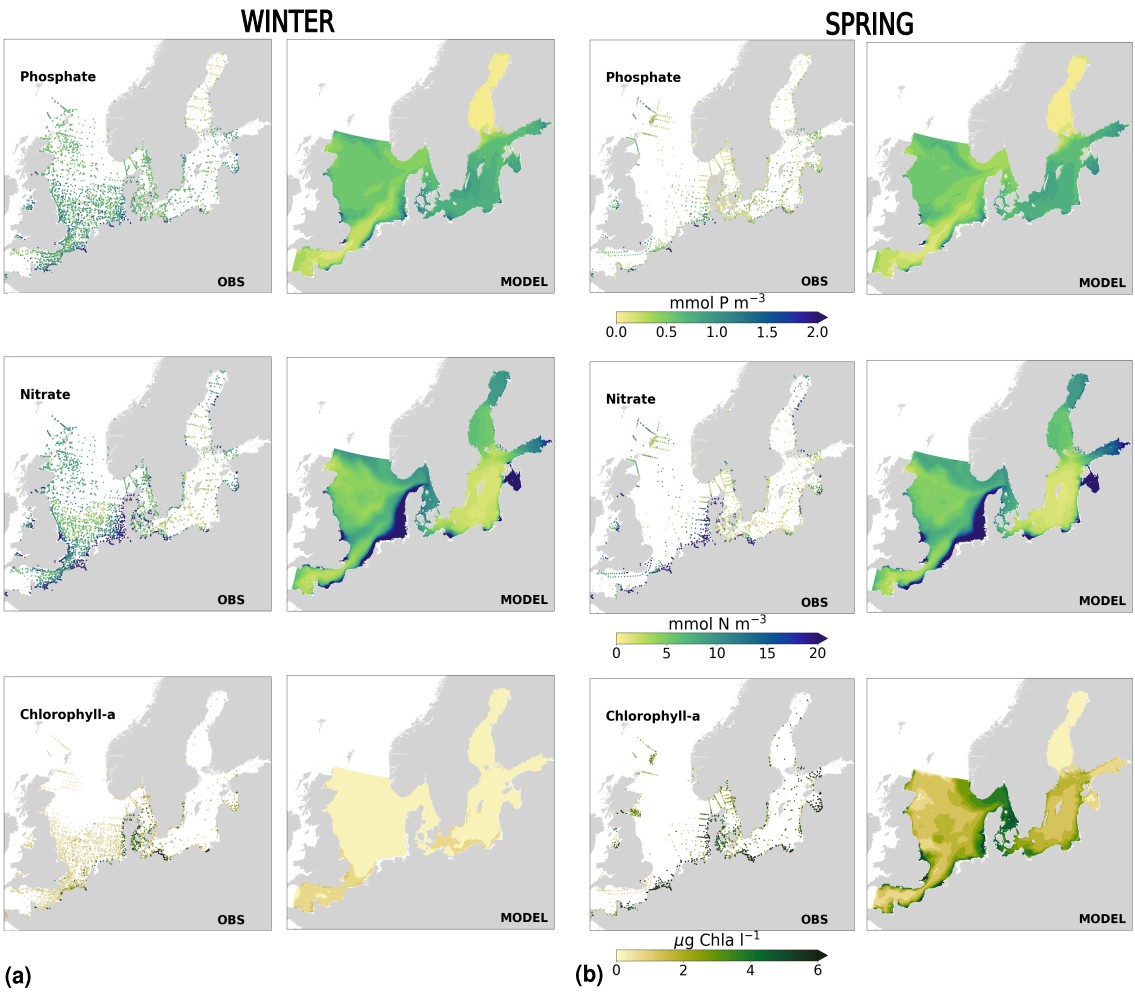

**Figure 12.** Spatial distribution of a) winter and b) spring for observations (left panels) and model results (right panels) for phosphate, nitrate and chlorophyll-a averaged over the period 2001 to 2017.

coastal waters into the Skagerrak, where this imbalance persists affecting the seasonal nutrient concentrations in the Kattegat (e.g. excess of phosphorus in summer, Fig 7 and Appendix B, Fig B1).

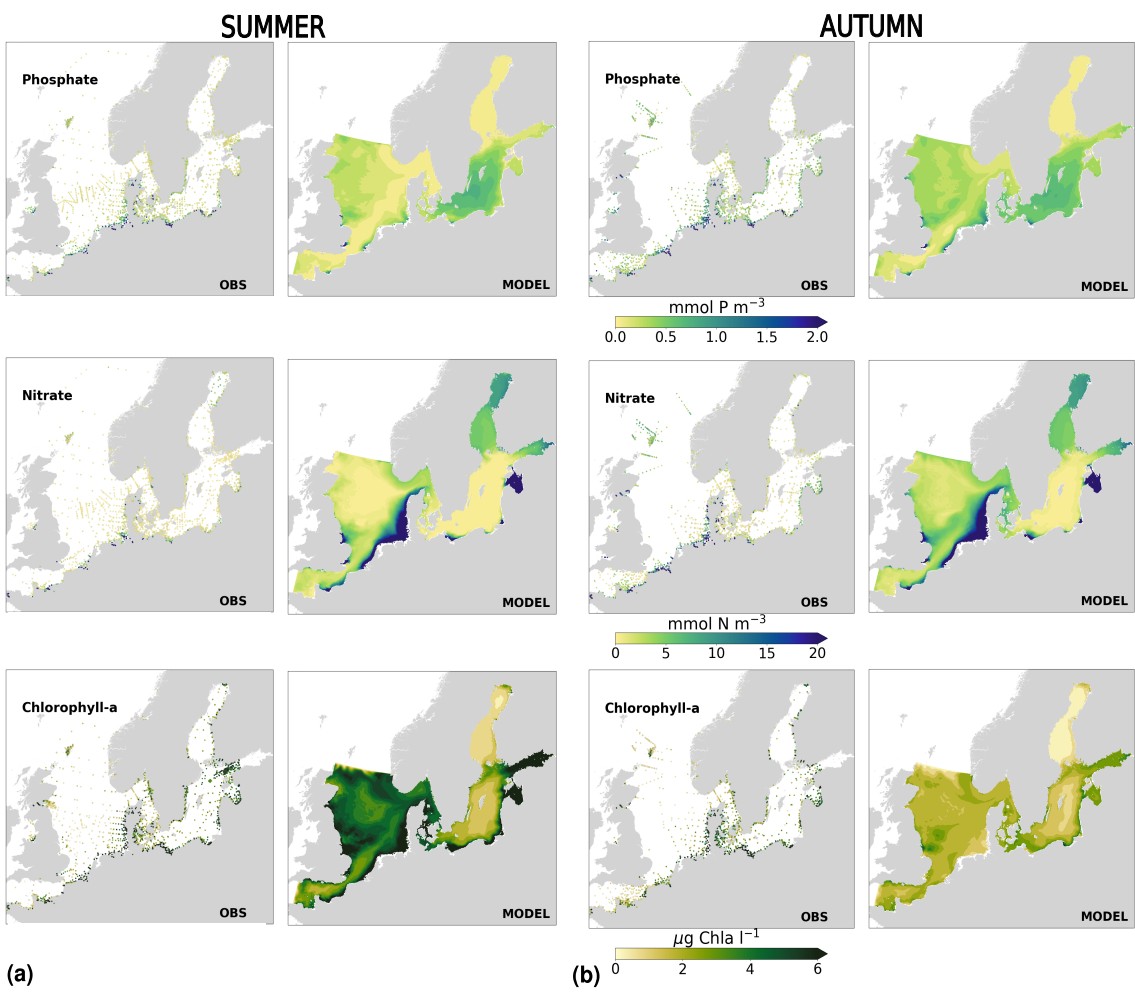

**Figure 13.** Spatial distribution of a) summer and b) autumn for observations (left panels) and model results (right panels) for phosphate, nitrate and chlorophyll-a averaged over the period 2001 to 2017.

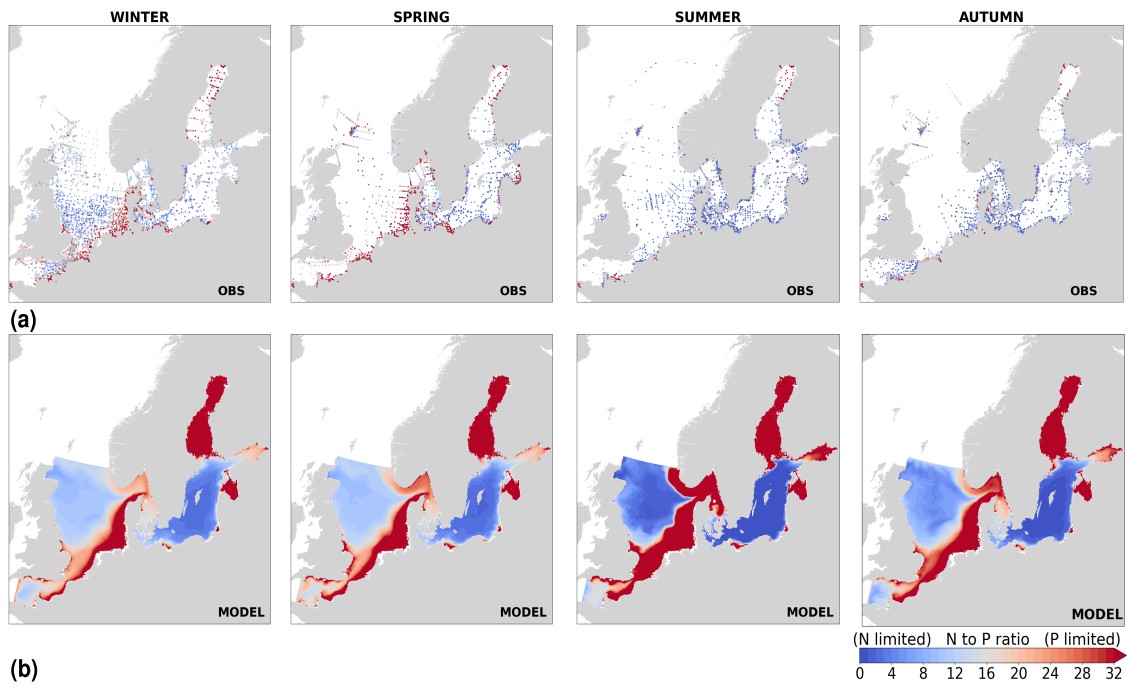

**Figure 14.** Seasonally nitrogen to phosphate ratios for the period 2001 to 2017 for a) observations and b) model results.

## 3.5 Relevance of the study

Previous ocean models coupled to SCOBI (e.g. Eilola et al., 2009; Almroth-Rosell et al., 2011) have the open boundary in the Kattegat. The main advantage of this model setup compared to those models, as well as to the typical model used to support HELCOM assessments (Gustafsson et al., 2012; Savchuk et al., 2012), is that NEMO-SCOBI allows for a study of the North Sea and the entire Skagerrak-Kattegat transition zone as the boundaries have been moved far from the Kattegat area. Covering the entire Swedish coast including the newly covered Skagerrak-Kattegat transition zone is particularly important for Swedish marine managers. The latter is a complex dynamic area difficult to simplify in order to represent correct (in/out)flows as boundary conditions (Gustafsson, 1997). This is especially true for long-term modeling as conditions in this region depend on the modeled processes in the adjacent basins, rather than prescribed boundary conditions. For climate runs it is, for example, difficult to prescribe boundary conditions at the high resolution needed in order to resolve the influence of processes in the North Sea and Skagerrak. Thus, extending the boundaries far from the Skagerrak-Kattegat transition zone allows for a better representation of its full dynamics.

When compared to other modelling studies, covering either one sea or both seas, our results show differences that are neither better nor worse than previous model results depending on the variable and area (e.g. Eilola et al., 2011a; Maar et al., 2011; Daewel and Schrum, 2013). This has also been demonstrated in a recent ensemble study on eutrophication that includes NEMO-SCOBI results (van Leeuwen et al., 2023), where for example chlorophyll-a concentrations were both over and underestimated by the different models used in the ensemble when compared to satellite and *in situ* observations in the southern North Sea and the English Channel (areas 7 and 11 in Fig. 1). It is nonetheless surprising that by expanding the model to cover such a different dynamic region, and with very little additional tuning, most processes in the North Sea are still well captured.

Here, we have not only validated our model results for individual stations, but also for areas officially used in international programs aiming to reduce eutrophication, such as OSPAR and HELCOM. To our knowledge, this is the first time that model results from one single model have been validated for the combined HELCOM-OSPAR assessment areas. Following the statement of Ducrotoy and Elliott (1998), Mee et al. (2008) and Koho et al. (2021) of a need for improved ecosystem models in these areas, we provide a step forward towards better understanding model area dependent performances and uncertainties. This also gives added value to contribute to joint OSPAR and HELCOM initiatives, especially on their work for healthy environments. Importantly, a variety of models simulating the biogeochemistry in similar areas should be used together, as they all differ significantly in their biogeochemical complexity and have different performing skills. The use of ensemble mean assessments has been shown to be good or even better than the results from individual models (Eilola et al., 2011a; van Leeuwen et al., 2023). We have shown that the NEMO-SCOBI model can be used to derive relevant indicators for HELCOM and OSPAR initiatives for the Baltic Sea-North Sea system, for which specific relevant improvements will be applied. Large model biases, especially that of nitrate, are also strongly linked to the applied forcing, which is continuously updated depending on available data and/or down scaling methods. Thus, the model will be used to produce novel climate and nutrient scenarios similar to those provided in the Climate Change Scenario Service (www.smhi.se/en/climate/future-climate/advanced-climate-change-scenario-service/oce/),

produced by the Swedish Meteorological and Hydrological Institute (SMHI), but now with a consistent model domain that covers the entire North Sea-Baltic Sea system.

## 3.6    Future work and knowledge gaps

Solving the phytoplankton bloom timing in the southeastern coastal North Sea and the Skagerrak-Kattegat transition zone in NEMO-SCOBI is a priority as it would significantly improve the model results, especially for nitrate concentrations and

seasonal behavior of the biogeochemical parameters. Our ecosystem model study suggests that besides nutrients, the light attenuation in the North Sea is key to determine the specific spatial distribution of phytoplankton communities, in agreement with findings in Ford et al. (2017) for the North Sea. Seasonal observations that relate to Kd (or secchi depths) are less abundant than those for nutrients in the entire domain, especially in the North Sea. However, when comparing NEMO-SCOBI results for secchi depths to winter-averaged observations reported in ICES for the North Sea during the period 2001 to 2017,

NEMO-SCOBI tends to underestimate secchi depths (by 1 to 5 m depending on the location) in the open waters of the North Sea, while it overestimates winter secchi depths (by 1 to 5 m) in a narrow fringe of the southeastern coastal waters (Appendix B, Fig. B8a). We have also compared our results to light attenuation averages for different areas in the North Sea from Capuzzo et al. (2013, 2015) and found that Kd values in NEMO-SCOBI are in very good agreement with those reported in these studies for most areas and seasons, with notably small differences ($\sim$2 m) in the open south-eastern and northern

North Sea for winter and spring. However, better constraining the spatial distribution of the light attenuation coefficient (Kd), included in the parametrization for phytoplankton growth (Appendix A; Eqs. A1 and A3), could significantly improve our results. In addition, the current parametrization of the light attenuation in NEMO-SCOBI does not allow for the co-occurrence of high phytoplankton growth rates, low temperatures and high secchi depths, as inferred from winter observations in the Skagerrak-Kattegat transition zone (Fig. 12a and Appendix A; Fig. A1). This suggests that the parametrization of light atten-

uation in NEMO-SCOBI is well adapted for open waters, but additional tuning is required for specific shallow areas, such as the Skagerrak-Kattegat transition zone and the southern coast of the North Sea.

Moreover, substance-specific attenuation coefficients from measurements are not well constrained. Maar et al. (2011) made a comparison between a model run considering a constant background value for Kd versus one with a salinity dependent Kd. The latter gave a better correlation between model results and observations, improving the timing of their model spring bloom. Their

approach remains an approximation of realistic Kd levels and calls for a dedicated study on light limitation for phytoplankton growth. For our model, one important factor affecting the light attenuation coefficient is the organic matter present in sea water. Therefore, future work will also consist in better capturing the detritus in the SCOBI model. However, detritus in the North Sea-Baltic Sea system is poorly observed and therefore, poorly constrained. Seasonal comparisons between the model detritus and organic phosphorus and nitrogen observations in surface waters (obtained by subtracting the inorganic nitrogen/phosphorus

from the total nitrogen/phosphorus in the ICES data set) suggest that the model underestimates detritus in coastal areas near point sources, especially in the southeastern North Sea and slightly overestimates it during winter in the central North Sea (not shown). One factor affecting detritus is the fraction of the organic matter coming from rivers that is actually bioavailable and not directly retained in coastal waters. Here, we have assumed a constant bioavailable fraction for the riverine organic nitrogen

and phosphorus (of 0.3 and 0.75, respectively) in the entire domain based on previous studies for the coast in the Baltic Sea (Nausch and Nausch e.g. 2007; Eilola et al. e.g. 2009; Asmala et al. e.g. 2017; Edman et al. e.g. 2018). Thus, input of organic matter from rivers, especially nitrogen, could be improved by better accounting for river specific organic matter retention in coastal waters. However, this fraction is highly uncertain, especially in coastal waters in the North Sea and would require additional sensitivity tests. Importantly, both the ICG-EMO and the Baltic Sea data sets from which our correction factors for nutrient loads are based come with large uncertainties (Savchuk et al., 2008; Lenhart et al., 2010; Gustafsson et al., 2012). From a sensitivity test where the river loads were changed (not shown), the largest effect in nutrient concentrations in surface waters, affecting both coastal and the open ocean, was found in the southern North Sea and in the Arkona basin. Additional effects were restricted to highly river influenced coastal areas.

Water column and benthic denitrification (Fig. 15) are two important processes that can remove nitrogen from the system. These are well studied in the Baltic Sea, but still poorly constrained on a seasonal basin-wide scale and long observational time series are also lacking. In the central Baltic Sea (east of Gotland), estimates of water column denitrification in autumn/summer for the years 2008 and 2010 are variable, but as high as $21\,\mathrm{mmol\,N\,m^{-2}\,d^{-1}}$ (Dalsgaard et al., 2013; Hietanen et al., 2012). Water column denitrification rates averaged from 2001 to 2017 for summer and autumn in the hindcast run are comparable to, but slightly lower than, these estimates (Fig. 15a). The total nitrogen removal from water column denitrification in the Baltic proper with persistent large hypoxic areas has been estimated to be 132 to $547\,\mathrm{kton\,N\,yr^{-1}}$ (Dalsgaard et al., 2013). In the model, the yearly rates in the Baltic proper vary between 40 to $129\,\mathrm{kton\,N\,yr^{-1}}$ during 2001 to 2017, with an average of $65\,\mathrm{kton\,N\,yr^{-1}}$ for the entire period. An improved mixing representation below the halocline in the Baltic proper would further improve the model oxygen concentrations in the intermediate waters of the Baltic proper. This would in turn, lead to higher denitrification rates and decreased nitrogen concentrations there. Consequently, the nitrogen transport to the adjacent basins (such as the Gulf of Riga, Finland and Bothnia) would likely decrease and therefore, the nitrate positive bias in such basins would be reduced. One way of improving the vertical mixing in NEMO would be to increase the vertical resolution of the model as discussed in Hordoir et al. 2019.

Benthic denitrification has been estimated in the Gulf of Bothnia (0 to $0.94\,\mathrm{mmol\,N\,m^{-2}\,d^{-1}}$; Stockenberg and Johnstone 1997; Bonaglia et al. 2017), in the Gulf of Finland (0.1 to $0.65\,\mathrm{mmol\,N\,m^{-2}\,d^{-1}}$; Tuominen et al. 1998; Hietanen and Kuparinen 2008), in the Northern Baltic Proper (0.014 to $0.3\,\mathrm{mmol\,N\,m^{-2}\,d^{-1}}$; Tuominen et al. 1998) and in the southern Baltic proper (0.012 to $0.69\,\mathrm{mmol\,N\,m^{-2}\,d^{-1}}$; Deutsch et al. 2010). In these basins, the benthic denitrification rates in the model are in good agreement with previous estimates, although at the lower end of the ranges (Fig. 15b). The seasonal variations in benthic denitrification in the Baltic Sea are poorly observed, but have been found to follow a marked seasonal cycle, with low rates in early spring increasing towards late summer to late autumn and decreasing towards late winter (Hietanen and Kuparinen, 2008). While this seasonal variation in benthic denitrification is not well captured by the model, the yearly rates are in good agreement with previous estimates. The nitrogen removal by benthic denitrification in the entire Baltic Sea has been calculated to be between 426 and $652\,\mathrm{kt\,N\,yr^{-1}}$ (Deutsch et al., 2010). In the model, the benthic removal rate of nitrogen in the Baltic Sea for the period 2001 to 2017 varies from 484 to $627\,\mathrm{kt\,N\,yr^{-1}}$, with a period average of $553\,\mathrm{kt\,N\,yr^{-1}}$. Improving

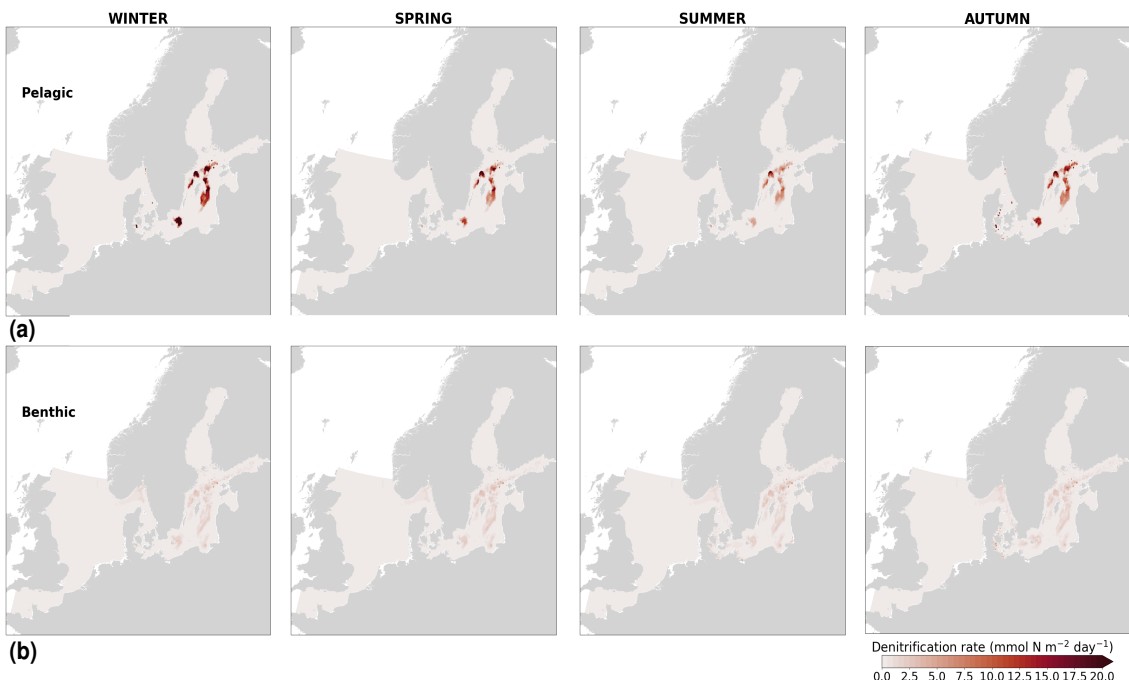

**Figure 15.** Model spatial distribution of seasonal averages of pelagic and benthic denitrification for the period 2001 to 2017.

the seasonal cycle of benthic denitrification in the model would likely improve the seasonal variability of nitrogen in basins with high denitrification rates.

 **4 Conclusions**

We have here presented and evaluated NEMO-SCOBI; a new coupled physical-biogeochemical model configuration for the Baltic and the North Sea. In conclusion, the model simulates biogeochemical variables well, reflects the physical and hydrodynamic processes, reproduces long-time trends and responds reasonably to anthropogenic nutrient sources along the coastal zone. This makes the model particularly suitable to be applied in future multi-stressor studies such as to test combined climate and nutrient scenarios (e.g. Wåhlström et al., 2020). It is therefore ready to be used to produce climate projection such as those in the SMHI Climate Scenario Service. Compared to other Baltic Sea models that have to prescribe climatological boundary values based on a limited number of observations in the Kattegat area (e.g. Eilola et al., 2009; Neumann et al., 2002), NEMO-SCOBI avoids many problems associated with a lateral boundary in the area of Kattegat and Skagerrak. This is of fundamental importance for the salt and oxygen inventory of the Baltic Sea as it controls the North Sea-Baltic Sea mass exchange. It was demonstrated that the model simulates this exchange in a physically consistent way with good skills for the oxygen dynamics in the Baltic Sea. NEMO-SCOBI also reveals a realistic seasonal cycle and interannual variability in most of the assessed variables, as well as model skill that can fully compete with existing models for the North Sea and the Baltic Sea. Thus, it can be used in further scientific applications, such as for a detailed analysis on long-time nutrient exchanges between basins and climate effects on eutrophication and oxygenation. It should be noted that robust statistical evaluations for long-time trends are particularly difficult to obtain in the Northern North Sea, the Gulf of Riga, the Gulf of Finland, the Åland Sea, the Quark and the Bothnian Bay due to great lack of observations in these areas. Including the entire North Sea-Baltic Sea system in one single model (as opposed to previous model versions coupled to SCOBI) allows for better identification of regions with similar biogeochemical behavior that are not limited to one or the other sea, as well as to study different processes occurring in the Kattegat-Skagerrak transition zone. NEMO-SCOBI can also keep contributing to European initiatives on de-eutrophication, water quality advice and support on nutrient reduction loads of both the North Sea and the Baltic Sea. However, additional care must be taken into account when evaluating regional seasonal cycles, especially for chlorophyll-a and nitrate. NEMO-SCOBI is, to our knowledge, the third physical-biogeochemical model covering the area of the Baltic and the North Sea (Daewel and Schrum 2013 and Maar et al. 2011). Future model inter-comparison studies of these three model systems have the potential to give valuable scientific insights into model performance and process understanding, and NEMO-SCOBI is thus an important addition to the model ensemble of this region.

*Code availability.* The NEMO-SCOBI code is available upon request in GitLab at https://git.smhi.se/fouo/nemo3.6

## Appendix A: Additional NEMO-SCOBI parametrizations

### A1 Phytoplankton growth in NEMO-SCOBI

Here, we only describe the phytoplankton growth as implemented in the current version and relevant related variables. Involved constant values are listed in Table A1. For a full overview and mass balance equations see Eilola et al. (2009). The parameterization for phytoplankton growth ($GROWTH$, Eq. A1) is defined as:

$$GROWTH_{1,2,3} = OXLIM \cdot LTLIM \cdot GMAX_{1,2,3} \cdot NUTLIM_{1,2,3} \cdot PHY_{1,2,3}. \tag{A1}$$

All three phytoplankton groups share the same oxygen dependency to prevent anabolism under anoxic conditions ($OXLIM$, Eq. A2) and light limitation ($LTLIM$, Eq. A3). These are defined as:

$$OXLIM = \frac{1}{1 + (\frac{\alpha_{ox}}{[O_2]})^{\beta_{ox}}} \quad \text{and} \tag{A2}$$

$$LTLIM = \frac{I_{PAR}}{I_{OPT}} \cdot EXP(1 - \frac{I_{PAR}}{I_{OPT}}), \tag{A3}$$

where the photosynthetic available radiation ($I_{PAR}$, Eq. A4) decreases exponentially with depth ($z$), and the optimum irradiation for photosynthesis ($I_{OPT}$ in W m$^{-2}$, Eq. A5) are described as:

$$I_{PAR} = \alpha_{PAR} \cdot I_0 \cdot EXP(-Kd \cdot z), \tag{A4}$$

$$I_{OPT} = MAX(I_{OPTMIN}, \alpha_{OPT} \cdot I_0). \tag{A5}$$

The solar radiation ($I_0$ in W m$^{-2}$) that reach the water surface is calculated at every time step within NEMO-Nordic and $I_{PAR}$ was set to account for light absorption due to biological fluxes. The vertical light attenuation coefficient ($Kd$) is affected by a constant background light attenuation ($Kd_w$) and by the concentrations of phytoplankton, zooplankton, yellow substances and detritus as follows:

$$Kd = Kd_w + \alpha_{phy} \cdot R_{chl:N} \cdot ([PHY1 + PHY2 + PHY3]) + \alpha_{zoo} \cdot ZOO + \alpha_{DETN} \cdot DETN + \lambda_{ys}, \tag{A6}$$

where $\alpha_{phy}$, $\alpha_{zoo}$ and $\alpha_{DETN}$ are the light attenuation coefficients per unit of chlorophyll-a, zooplankton and detritus, respectively and $\lambda_{ys}$ is a regionally prescribed vertical attenuation coefficient for yellow substances.

The three phytoplankton groups have the same fixed mortality rate (5% of the phytoplankton concentrations per day). They differ from each other by their maximum growth ($GMAX$, Eq. A7-A9), which is temperature dependent, their nutrient limitation of the growth ($NUTLIM$, Eq. A10-A12), and their sinking rates. Hence, they depend on both physical and chemical conditions of the water. The group PHY1 has the characteristics of "diatoms" which besides using nitrogen and phosphorus,

also use silica to build up their shells. They can grow rapidly at cold conditions and at higher nutrient concentrations, having an advantage over flagellates in turbulent conditions. The group PHY2 represents "flagellates and others" and benefits from stratified conditions. That is when surface temperatures are relatively high and the nutrient concentrations low above the thermocline. The group PHY3 has the characteristics of "filamentous cyanobacteria", which grow in warm low-saline waters and therefore, a salinity threshold of $S \leq 10$ is used for cyanobacteria only to grow in the Baltic Sea. As in previous versions of SCOBI, cyanobacteria also have the ability to fix molecular nitrogen ($N_2$) when nitrogen concentrations are low in the water. They have a tendency to remain close to the surface waters, so that in the model they are considered to be neutrally buoyant (i.e. their sinking speed is set to zero). The maximum growth for PHY1, PHY2 and PHY3 ($GMAX_1$, $GMAX_2$ and $GMAX_3$, respectively) is defined as:

$$GMAX_1 = \alpha_{PHY1} \cdot EXP(\beta_{PHY1} \cdot T), \tag{A7}$$

$$GMAX_2 = \alpha_{PHY2} \cdot EXP(\beta_{PHY2} \cdot T) \text{ and} \tag{A8}$$

$$GMAX_3 = \alpha_{PHY3} \cdot \frac{EXP(\beta_{PHY3} \cdot T)}{1 + EXP(TK1 - TK2 \cdot T)}. \tag{A9}$$

Note that the growth sensitivities to temperature ($T$) for all phytoplankton types have been tuned for NEMO-Nordic based on sensitivity analysis (see Table A1 for updated constant values). The silica limitation for group PHY1 was implemented following Michaelis-Menten kinetics. The half saturation constant for the uptake of silica by diatoms has been shown to be extremely variable depending on the species and water conditions (Thamatrakoln and Hildebrand, 2008). Here we take a rather conservative value (0.1 mmol Si m$^{-3}$, Table A1) after Paasche (1973) and Pasquer et al. (2005). The nutrient limitation for PHY1, PHY2 and PHY3 ($NUTLIM_1$, $NUTLIM_2$ and $NUTLIM_3$, respectively) in SCOBI is now as follows:

$$NUTLIM_1 = MIN(NLIM, \ PLIM, \ SiLIM), \tag{A10}$$

$$NUTLIM_2 = MIN(NLIM, \ PLIM) \text{ and} \tag{A11}$$

$$NUTLIM_3 = MIN(NLIM, \ PLIM), \tag{A12}$$

where $NLIM$, $PLIM$ and $SiLIM$ are the nitrogen, phosphate and silica limitation, respectively:

$$NLIM = \frac{[NO_3]}{K_{NO3} + [NO_3]} \cdot EXP(-\Phi \cdot [NH_4]) + \frac{[NH_4]}{K_{NH4} + [NH_4]}, \tag{A13}$$

$$PLIM = \frac{[PO_4]}{K_{PO4} + [PO_4]} \text{ and} \tag{A14}$$

$$SiLIM = \frac{[Si]}{K_{Si} + [si]}. \tag{A15}$$

The $K_{NO3,NH4,PO4,Si}$ and '[ ]' are the half saturation constants and concentrations for nitrate, ammonium, phosphate and silica, respectively and $\Phi$ is the strength of the ammonium inhibition for nitrate uptake. An example of the total $GROWTH$

as a function of $Kd$ and $T$ for typical model values in surface waters of the Kattegat is shown in Fig. A1. At low $Kd$ (high secchi depth) and low $T$, phytoplankton growth rates can only be small in the model.

Table A1: Constants as applied in NEMO-SCOBI for phytoplankton growth. Numbers in bold are updated or new values based on sensitivity analysis while other numbers follow those in Eilola et al. 2009 or Almroth-Rosell et al. (2015) (indicated by [1]). PHY1 stands for diatoms, PHY2 for Flagellates and others and PHY3 for filamentous cyanobacteria.

| Symbol | Description | Value | Unit |
|---|---|---|---|
| $\alpha_{ox}$ | Constant in the oxygen dependency to prevent anabolism under anoxic condition | 2 | $\mathrm{ml\,O_2\,l^{-1}}$ |
| $\beta_{ox}$ | Constant in the oxygen dependency to prevent anabolism under anoxic condition | 6 | - |
| $\alpha_{PAR}$ | Photosynthetic available radiation (PAR) fraction of the solar radiation at the sea surface | 0.5 | - |
| $I_{OPTMIN}$ | Constant minimum value for optimum irradiance | 25 | $\mathrm{W\,m^{-2}}$ |
| $\alpha_{OPT}$ | A constant fraction of the incident PAR | 0.25 | - |
| $Kd_w$ | Background light attenuation | 0.04 | $\mathrm{m^{-1}}$ |
| $\alpha_{chla}$ | Vertical light attenuation per unit chlorophyll concentration | **0.04** | $\mathrm{(mmol\,N\,m^{-3}\,m)^{-1}}$ |
| $R_{chl:N}$ | Chlorophyll to nitrogen ratio | 0.63 | $\mathrm{mmol\,N/mg\,Chla}$ |
| $\alpha_{zoo}$ | Vertical light attenuation per unit zooplankton concentration | **0.0008** | $\mathrm{(mg\,C\,m^{-3}\,m)^{-1}}$ |
| $\alpha_{DETN}$ | Vertical light attenuation per unit detritus concentration | **0.0008** | $\mathrm{(mg\,C\,m^{-3}\,m)^{-1}}$ |
| $\lambda_{ys}$ | Yellow substances vertical attenuation coefficient in | | |
| | the Bothnian Bay and the Gulf of Finland | 0.23[2] | $\mathrm{m^{-1}}$ |
| | the Bothnian Sea | 0.21[2] | $\mathrm{m^{-1}}$ |
| | the Gulf of Riga | 0.27[2] | $\mathrm{m^{-1}}$ |
| | the Baltic proper | 0.13[2] | $\mathrm{m^{-1}}$ |
| | the North Sea and the Skagerrak-Kattegat | **0.12** | $\mathrm{m^{-1}}$ |
| $\alpha_{PHY1}$ | PHY1 growth rate at 0 $^o$C | 0.75[1] | $\mathrm{day^{-1}}$ |
| $\alpha_{PHY2}$ | PHY2 growth rate at 0 $^o$C | 0.5[1] | $\mathrm{day^{-1}}$ |
| $\alpha_{PHY3}$ | PHY3 growth rate at 0 $^o$C | **0.6** | $\mathrm{day^{-1}}$ |
| $\beta_{PHY1}$ | PHY1 growth rate dependence on temperature | **0.05** | $\mathrm{^oC^{-1}}$ |
| $\beta_{PHY2}$ | PHY2 growth rate dependence on temperature | **0.085** | $\mathrm{^oC^{-1}}$ |
| $\beta_{PHY3}$ | PHY3 growth rate dependence on temperature | 0.0633 | $\mathrm{^oC^{-1}}$ |
| $TK1$ | PHY3 growth rate dependence on temperature | **24** | - |
| $TK2$ | PHY3 growth rate dependence on temperature | 2 | $\mathrm{^oC^{-1}}$ |
| $K_{NO3}, K_{NH4}$ | Half-saturation constants for nitrate and ammonium, respectively | | |
| | for PHY1 | 0.5 | $\mathrm{mmol\,N\,m^{-3}}$ |
| | for PHY2 and PHY3 | 0.25 | |
| $K_{PO4}$ | Half-saturation constants for phosphate | | |
| | for PHY1 | 0.1 | $\mathrm{mmol\,P\,m^{-3}}$ |
| | for PHY2 and PHY3 | 0.05 | |
| $K_{Si}$ | Half-saturation constants for silica for PHY1 | 0.1 | $\mathrm{mmol\,Si\,m^{-3}}$ |
| $\Phi$ | Strength of the ammonium inhibition of nitrate uptake | 1.5 | $\mathrm{(mmol\,N\,m^{-3})^{-1}}$ |

[1]: applied in the SCOBI version in Almroth-Rosell et al. 2015, [2]: applied in Eilola et al. 2009.

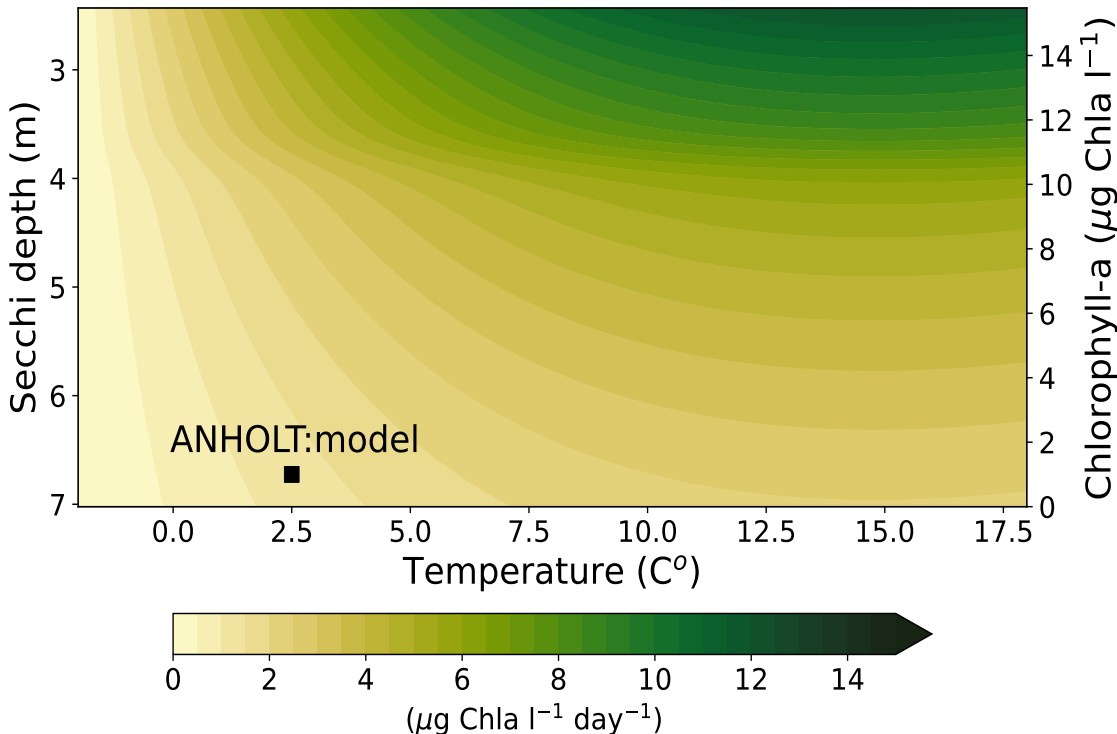

**Figure A1.** Sensitivity of phytoplankton growth to temperature and light attenuation ($Kd$) given by equation A6 and here shown as secchi depth = $1.45/Kd$, which is a general approximation based on Holmes (1970). Except for temperature and concentrations of $PHY1$, $PHY2$ and $PHY3$, all variables in equations A1 to A15 are fixed to averaged values for the period 2001 to 2017 in the Kattegat area at ~4 m depth. The corresponding total chlorophyll-a concentrations ($PHY1 + PHY2 + PHY3$) from which $Kd$ was obtained are shown as reference on the right x axis. The actual model value obtained at ANHOLT for February is highlighted (black square). As reference, the temperature, the secchi depth and the chlorophyll-a concentrations obtained from observations at ANHOLT are 2.8 C°, 8 m and 5 µg Chla L$^{-1}$.

## A2   Benthic fluxes in NEMO-SCOBI

In SCOBI, the sinking organic matter (phytoplankton and detritus) is deposited on the sediments and builds up the corresponding benthic nutrient pools: BSi, BOP and BN. The sinking rate of phytoplankton varies between functional types and follows the velocity sinking function of Penta and Walsh (1995). The sinking velocity of detritus is a function of depth, the detritus pool and a constant sinking velocity rate set to 2.5 meters per day in the water column and to 3.5 meters per day in the bottom most cell to account for aggregation processes, following (Neumann et al., 2002). The release of inorganic nutrients from benthic organic material has been modified to better capture the nutrient dynamics for both the Baltic Sea and the North Sea. Similarly to Almroth-Rosell et al. 2015, the total release of phosphorus from remineralized benthic organic material

($BOPOUT_{PO4}$, Eq. A16) consists of two pathways: the transfer of phosphorus from BOP to the sediment pool of mineral bound inorganic phosphorus ($BOPREM_{BIP}$, Eq. A17) and the direct release of phosphate to the overlying water column ($BOPREM_{PO4}$, Eq. A18). The release of benthic phosphorus is temperature dependent (described by the remineralisation rate term $\lambda_T$, Eq. A19), oxygen dependent and now also salinity limited (included in the limitation term $\delta_{O2S}$; Eq. A20). In the well mixed North Sea, $BOPREM_{BIP}$ is generally less important than in the Baltic Sea. Here, $BOPREM_{BIP}$ decreases

with increasing salinity ($S$) and decreasing bottom oxygen concentrations ($[O_2]_{bot}$), which then increases $BOPOUT_{PO4}$ accordingly, as follows:

$$BOPOUT_{PO4} = (BOPREM_{PO4} + BOPREM_{BIP}), \tag{A16}$$

$$BOPREM_{PO4} = (\alpha_{RC} - \lambda_{O2S}) \cdot \lambda_T \cdot BOP, \tag{A17}$$

$$BOPREM_{BIP} = \lambda_T \cdot BOP - BOPREM_{PO4}. \tag{A18}$$

where,

$$\lambda_T = \alpha \cdot EXP(\beta \cdot T), \tag{A19}$$

$$\lambda_{O2S} = \frac{f + g \cdot TANH(MS - S)}{1 + a \cdot EXP(-b \cdot ([O_2]_{bot} - c)) - \frac{d}{1 + (SS/S)^e}}. \tag{A20}$$

To avoid negative values for $BOPREM_{BIP}$, $BOPREM_{PO4} = \lambda_T \cdot BOP$ when $\lambda_{O2S} < 0.15$. The salinity dependency starts at salinities of 20 (dividend in Eq. A20). At even higher salinities, such as those in the North Sea ($>30$), the transfer of

phosphate to BIP does not occur (i.e. $BOPREM_{BIP}$ is approximately 0). However, the bottom water oxygen concentration ($[O_2]_{bot}$) remains the most important variable controlling the benthic transfer of phosphate to BIP. Under anoxic conditions (i.e. $O_2 \leq 0$) all phosphate from remineralisation is directly released to the water column ($BOPREM_{BIP} = 0.0$), independently of the salinity level.

    In addition to $BOPREM_{BIP}$, the BIP pool is increased by scavenging of PO4 under oxic conditions ($PO4SCAV_{BIP}$;

Eq A21) and decreased by the redox dependent release of inorganic phosphorus from iron-bound-particles ($BIPREL_{PO4}$; Eq. A22). Both depend on the oxygen concentrations in bottom waters and are parameterized as follows:

$$PO4SCAV_{BIP} = \alpha_{pscav} \cdot (1 - \frac{[BIP]}{[BIP] + K_{BIP}}) \cdot \frac{[O_2]_{bot} \cdot [PO_4]}{[O_2]_{bot} + K_{o2bot}} \quad \text{and} \tag{A21}$$

$$BIPREL_{PO4} = \alpha_{prel} \cdot \frac{[BIP]}{[BIP] + K_{BIP}} \cdot [BIP] \cdot (1 - \frac{[O_2]_{bot}}{[O_2]_{bot} + K_{o2bot}}) \tag{A22}$$

The water diffusivity of inorganic phosphorus is given by $\alpha_{pscv}$, as:

$$\alpha_{pscv} = [C1 + C2 \cdot \frac{T_{bot} - C3}{1 - ln(\phi)^2}] \cdot \frac{\phi}{\Delta X}. \tag{A23}$$

The constant values involved in both fluxes ($\alpha_{prel}$, $K_{BIP}$, $K_{o2bot}$, $\Delta X$, $C1_{sp}$, $C2_{sp}$, $C3_{sp}$ and $\phi$) are described in Table A2. Note that from these constants, only $\phi$ (the sediment porosity) differs from older SCOBI versions. This term was regional specific in previous versions. In NEMO-SCOBI, this term had to be simplified for numerical reasons, but will be made regional dependent at a later stage. In Eq. A23, $T_{bot}$ is the bottom water temperature between 0 and 25 °C. The BIP pool is also affected by the permanent burial of phosphorus ($BIPBUR$; Eq. A28), the resuspension of inorganic P due to wave and currents friction and sinking of WIP in bottom waters. The resuspension and the sinking of WIP depend on the shear stress, following Eq. A31.

For the release of nitrogen (in the form of ammonium) from benthic organic matter ($BNOUT_{NH4}$), the remineralisation rate is regulated by the temperature and BN (Eq. A24). The release of silica from benthic organic matter ($BSiOUT_{DSi}$) is assumed to be directly released to the overlying water column in the form of dissolved silica with half the dissolution rate compared to the remineralisation rate (Eq. A25):

$$BNOUT_{NH4} = \lambda_T \cdot BN, \tag{A24}$$

$$BSiOUT_{DSi} = \lambda_T/2 \cdot BSi. \tag{A25}$$

Other oxygen-dependent benthic processes, such as benthic denitrification and ammonium sequestration on particles, determine how much ammonium enters the water column from the sediments and follow the equations in Eilola et al. (2009). Here a nitrate limitation term is added to the benthic denitrification of pelagic nitrate ($BDEN_{NO3}$), which removes nitrate from the water column as follows:

$$BDEN_{NO3} = \lambda_T \cdot BN \cdot \frac{[NO_3]}{[NO_3] + K_{bden}}. \tag{A26}$$

Permanent burial of organic matter ($BOPBUR$, $BIPBUR$, $BNBUR$ and $BSiBUR$) depends on a shared but regional constant burial rate ($\alpha_{bur}$) and the accumulated material within sediments, as follows:

$$BOPBUR = \alpha_{bur} \cdot BOP, \tag{A27}$$

$$BIPBUR = \alpha_{bur} \cdot BIP, \tag{A28}$$

$$BNBUR = \alpha_{bur} \cdot BN \ \text{ and} \tag{A29}$$

$$BSiBUR = \alpha_{bur} \cdot BSi. \tag{A30}$$

The constant burial rates are prescribed per basin and respective values are shown in Table A2. The resuspension of benthic organic nutrients due to wave and current friction ($S$) depends on a prescribed critical shear stress ($\tau_{crit}$), which here differ in the Baltic Sea and the North Sea (Table B1), and the mean shear stress ($\tau$), following Almroth-Rosell et al. (2011):

$$S = \begin{cases} S_o \cdot (\frac{\tau}{\tau_{crit}} - 1) & \text{if } \tau > \tau_{crit} \\ W_s \cdot (1 - \frac{\tau}{\tau_{crit}}) & \text{if } \tau < \tau_{crit}, \end{cases} \tag{A31}$$

where $S_o$ is the maximum upward velocity of particles and $W_s$ the sinking velocity. Resuspension occurs when the bottom stress exceeds $\tau_{crit}$, otherwise the suspended material is (re)deposited in the sediments. Based on sensitivity analysis, a more conservative value for $\tau_{crit}$ is used in the Baltic Sea compared to previous SCOBI versions and a small $\tau_{crit}$ for the North Sea (Table A2) was added. This is because bottom waters in the North Sea are generally more dynamic than those in the Baltic Sea and therefore more sensitive to resuspension of benthic material (Almroth-Rosell et al., 2011; Thompson et al., 2011).

Table A2: Constants as applied in NEMO-SCOBI for benthic processes. Numbers in bold are updated values.

| Symbol | Description | Value | Unit |
|---|---|---|---|
| **Involved in benthic organic processes** | | | |
| $\alpha$ | remineralisation rate of benthic organic material at 0°C | 0.0005 | day$^{-1}$ |
| $\beta$ | Constant temperature for remineralisation of benthic organic matter | 0.15 | °CF$^{-1}$ |
| $\alpha_{RC}$ | Maximum phosphorus release capacity from the sediments at S = 0 | 1.15 | - |
| a | Constant in oxygen limitation for benthic phosphorus release | 0.5 | - |
| b | Constant in oxygen limitation for benthic phosphorus release | 1.5 | $l\,O_2\,ml^{-1}$ |
| c | Constant in oxygen limitation for benthic phosphorus release | 0.7 | $l\,O_2\,ml^{-1}$ |
| d | Constant in salinity limitation for benthic phosphorus release | 0.15 | - |
| e | Constant in salinity limitation for benthic phosphorus release | 20 | - |
| f | Constant in salinity limitation for benthic phosphorus release | 0.5 | - |
| g | Constant in salinity limitation for benthic phosphorus release | 0.5 | - |
| SS | Constant in salinity limitation for benthic phosphorus release | 5 | - |
| MS | Maximum salinity at which benthic phosphorus release occurs | 20 | psu |
| | | | |
| **Involved in benthic inorganics processes** | | | |
| $\alpha_{prel}$ | Maximum release rate of benthic inorganic phosphorus | 0.01 | day$^{-1}$ |
| $\Delta X$ | Length scale of the diffusion gradient of phosphorus | 0.01 | m |
| $K_{BIP}$ | Half saturation value of benthic inorganic phosphorus | 484 | $mmol\,P\,m^{-2}$ |
| $K_{o2bot}$ | Half saturation value of bottom water oxygen | $10^{-4}$ | m |
| $K_{bden}$ | Half saturation value for nitrate in benthic denitrification of pelagic nitrate | 0.1 | $mmol\,N\,m^{-3}$ |
| C1 | Constant regulating the scavenging of phosphorus | 7.34 x 10$^{-10}$ | - |
| C2 | Constant regulating the scavenging of phosphorus | 0.16 x 10$^{-10}$ | - |
| C3 | Constant regulating the scavenging of phosphorus | 25 | - |
| $\phi$ | Sediment porosity | **0.75** | $g\,cm^{-3}$ |
| $\tau_{crit}$ | Critical bottom stress value for resuspension: | | |
| | in the Baltic Sea | **0.2** | $N\,m^{-2}$ |
| | in the North Sea | 0.1 | $N\,m^{-2}$ |
| | | | |
| **Involved in burial** | | | |
| $\alpha_{bur}$ | Burial constant rate per basin: | | $10^{-4}\,m^2\,day^{-1}$ |
| | Bothnian Bay [1] | 2.2 | |
| | Bothnian Sea [2] | 4.1 | |

| Symbol | Description | Value | Unit |
|--------|-------------|-------|------|
| | Gulf of Finland [3] | 2.7 | |
| | Gulf of Riga | 4.1 | |
| | Baltic proper [4] | 0.6 | |
| | Bornholm Basins | 0.9 | |
| | Arkona Basins | 0.9 | |
| | Skagerrak-Kattegat [5] | 1.8 | |
| | North Sea [6] | **1.8** | |

[1]Includes the Quark; [2]Includes the western Åland Sea; [3]Includes the eastern Åland See (i.e., Archipelago Sea);

[4]Includes the Gdansk, Western Gotland and the Northern Baltic Proper basins;

[5]Includes the Bay of Mecklenburg, the Kiel Bay, the Sound and the Coastal NOR 3;

[6]Includes all areas in the North Sea.

## Appendix B:  Additional results

Additional results are presented in this section complementing those shown in the main text, mainly as further examples of model performance:

Table B1 - The model skill of phosphate, nitrate, chlorophyll-a and oxygen at additional stations (section 3.2).

Figure B1 and Figure  B2: The monthly-, the seasonally- and the yearly-averages over the period 2001 to 2017 at Å17 in the Skagerrak and BY5 in the Bornholm Basin, respectively. The biogeochemistry above 60 m at Å17 and ANHOLT is as described in the main text, section 3.1. However, below such depth at the deep stations of the Skagerrak-Kattegat transition zone (namely Å15 and Å17), the model is in good agreement with observations and shows little monthly-, seasonal- and annual-variability (e.g. Fig. B1). Note that nitrate is only underestimated by the model below 60 m at Å17. The biogeochemistry at BY5 is similar than that at BY15, as described in the main text, section 3.1 and only small differences can be observed at BY5, principally due to the fact that BY5 is shallower than BY15. For example, at BY5, no positive oxygen bias in intermediate waters (below ∼75 m) is displayed by the model as salinity is still well captured at this station and at these depths.

Figures B3 and B4: Time series of water column nitrate and phosphate at three of the stations in the southern North Sea that include the largest number of observations. Model results are compared to observations and show a general good agreement both in magnitude and seasonality. However, high nitrate concentrations linger in the model, especially at Walcheren due to a frequent lack of yearly summer depletion. Because the number of observations remains lower than those from the SHARK stations (especially at EastCoastNS), these stations have not been included in the statistical analysis (e.g. regression analysis and model skills per station).

Figure B5 - The spatio-temporal model performance at a fine regional scale for phosphate and nitrate at surface, intermediate and deep waters.

Figure B6 - The seasonal spatial distribution of differences between model results and observations for three main biochemical parameters in surface waters (namely, nitrate, phosphate and chlorophyll-a). The figure shows that the difference between model and observation vary per season and per variable, especially for chlorophyll-a. The smallest difference in phosphate are found in the Gulf of Bothnia for all seasons, in the northern North Sea in winter and in the southern North Sea in summer. For nitrate, the smallest differences between model and observations are mainly in the central North Sea and the Baltic proper in all seasons.

Figure B7 - The seasonal spatial distribution of the three included phytoplankton species in surface waters clearly show that in the model flagellates dominate in summer, while cyanobacteria are mainly restricted in the Baltic proper during Autumn, when diatoms and flagellates decrease and nitrate concentrations are low.

Figure B8 - The seasonal spatial distribution of observed secchi depths when compared to model results show that there is a general overestimation of the modeled secchi depths in the open ocean and the Skagerrak-Kattegat transition zone, but that the light penetration is best captured in the Baltic Sea during summer.

Table B1: List of the total number of observations (nobs), the 1 - correlation coefficient (1 - $r$) and the cost function ($CF$) for the period 2001 to 2017 at 12 stations in the Baltic Sea that are not shown in Fig. 10 for 4 main biogeochemical parameters. The 1 - $r$ and the $CF$ are evaluated for the entire period (p), for winter (w), for spring (sp), for summer (s) and for autumn (a). A "$-$" indicates no observations available for the corresponding evaluated time period. Numbers in black indicate good or acceptable model skill, where good model skill is highlighted in bold (i.e. when both 1 - $r$ and $CF$ are smaller than 0.35 and 1, respectively). Numbers in `typewriter` and *italic* fonts indicate poor model skill (i.e. when both 1 - $r$ and $CF$ are larger than 0.7 and 2, respectively), but "close to outer circle" and "far from the outer circle", respectively. When nobs is less than 500 the variable at that stations is not considered in this analysis.

| Station | nobs | 1-r (p) | C (p) | 1-r (w) | C (w) | 1-r (sp) | C (sp) | 1-r (s) | C (s) | 1-r (a) | C (a) |
|---|---|---|---|---|---|---|---|---|---|---|---|
| **PO4** | | | | | | | | | | | |
| B1 | 3582 | **0.3** | **0.7** | 0.4 | 0.9 | `0.6` | `1.1` | 0.3 | 1.2 | **0.2** | **0.5** |
| B7 | 948 | 0.6 | 0.7 | *1.1* | *1.6* | *1.0* | *0.9* | `0.7` | `0.7` | 0.4 | 1.0 |
| BY1 | 2898 | 0.5 | 0.9 | 0.5 | 0.9 | 0.8 | 1.7 | 0.4 | 0.9 | **0.3** | **0.6** |
| BY2 | 3075 | 0.4 | 0.7 | 0.5 | 0.8 | 0.7 | 1.4 | **0.3** | **0.8** | 0.2 | 0.5 |
| BY31 | 12606 | **0.1** | **0.3** | **0.04** | **0.2** | **0.1** | **0.3** | **0.04** | 0.3 | 0.04 | 0.3 |
| BY32 | 5544 | **0.04** | **0.3** | **0.04** | **0.2** | **0.1** | **0.3** | **0.04** | 0.3 | 0.03 | 0.3 |
| BY38 | 4727 | **0.1** | **0.4** | **0.1** | **0.3** | **0.1** | **0.4** | **0.1** | **0.4** | 0.1 | 0.4 |
| BY5 | 4576 | **0.2** | **0.5** | **0.2** | **0.5** | **0.2** | **0.9** | **0.2** | **0.5** | 0.2 | 0.4 |
| C3 | 1984 | **0.2** | **1.0** | **0.2** | **1.0** | 0.1 | 1.1 | **0.1** | **0.6** | 0.02 | 0.8 |
| F3 | 1166 | *0.8* | *1.4* | *0.9* | *1.5* | *0.6* | *1.6* | *0.7* | *9.2* | – | – |
| F9 | 1560 | `0.5` | `1.6` | *0.5* | *1.7* | 0.2 | 1.4 | 0.5 | 6.0 | – | – |
| P2 | 4180 | 0.4 | 0.6 | *0.8* | *0.8* | `0.7` | `1.1` | **0.3** | **1.0** | 0.4 | 0.7 |
| **NO3** | | | | | | | | | | | |
| B1 | 3580 | 0.5 | 0.7 | *0.5* | *2.3* | *0.8* | *1.0* | *0.7* | *1.9* | 0.5 | 0.7 |
| BY1 | 2800 | 0.5 | 0.9 | 0.6 | 0.8 | *0.9* | *1.5* | 0.5 | 1.3 | **0.3** | **0.8** |
| BY2 | 2967 | 0.4 | 0.9 | 0.6 | 0.8 | *0.9* | *1.5* | 0.4 | 1.4 | **0.2** | **0.7** |
| BY31 | 11875 | *0.9* | *1.9* | *1.3* | *1.8* | *1.2* | *1.8* | 0.8 | 2.2 | 0.8 | 2.4 |
| BY32 | 4725 | 0.6 | 1.1 | *1.0* | *1.8* | 0.8 | 1.5 | 0.5 | 0.9 | 0.5 | 1.3 |
| BY38 | 4338 | 0.5 | 1.1 | *0.8* | *1.6* | `0.6` | `1.3` | 0.4 | 1.1 | 0.4 | 1.0 |
| BY5 | 4374 | 0.5 | 0.7 | 0.4 | 0.8 | 0.5 | 0.9 | 0.6 | 0.8 | 0.4 | 0.6 |
| P2 | 4009 | *0.3* | *1.5* | *0.5* | *1.6* | 0.7 | 1.7 | 0.5 | 3.4 | *0.7* | *3.4* |
| **CHLA** | | | | | | | | | | | |
| B1 | 3581 | *0.9* | *0.9* | `0.7` | `0.9` | *1.0* | *1.0* | 0.8 | 1.4 | `0.7` | `1.1` |
| BY1 | 2067 | *0.8* | *0.8* | `0.7` | `0.9` | *1.0* | *0.8* | 0.4 | 1.2 | `0.7` | `0.8` |
| BY2 | 2856 | 0.8 | 1.0 | 0.6 | 0.8 | 1.0 | 1.0 | 0.5 | 1.6 | 0.6 | 0.7 |
| BY31 | 11499 | 0.4 | 0.5 | 0.6 | 0.5 | **0.3** | **0.5** | **0.3** | **0.5** | 0.4 | 0.5 |
| BY32 | 3285 | 0.5 | 0.5 | 0.6 | 0.6 | 0.4 | 0.5 | 0.4 | 0.5 | 0.6 | 0.6 |

**Table B1 – continuation**

| Station | nobs | 1-r (p) | CF (p) | 1-r (w) | CF (w) | 1-r (sp) | CF (sp) | 1-r (a) | CF (a) | | |
|---|---|---|---|---|---|---|---|---|---|---|---|
| BY38 | 3916 | 0.6 | 0.5 | 0.7 | 0.6 | 0.7 | 0.5 | 0.4 | 0.6 | 0.6 | 0.6 |
| BY5 | 3751 | *0.8* | *0.5* | 0.6 | 0.7 | 0.6 | 0.7 | 0.9 | 0.4 | 0.5 | 0.8 |
| P2 | 3861 | *0.9* | *0.2* | *0.8* | *0.4* | *1.0* | *0.8* | *1.0* | *1.8* | *0.9* | *1.6* |
| **O2** | | | | | | | | | | | |
| B1 | 558 | *0.8* | *0.4* | **0.2** | **0.7** | **0.3** | **1.0** | *1.2* | *0.5* | 0.6 | 1.2 |
| BY1 | 2898 | **0.3** | **0.5** | **0.2** | **0.4** | **0.3** | **0.5** | 0.5 | 1.0 | **0.3** | **0.5** |
| BY2 | 3070 | **0.3** | **0.5** | **0.2** | **0.4** | **0.3** | **0.5** | 0.5 | 1.0 | **0.3** | **0.5** |
| BY31 | 7660 | **0.1** | **0.3** | **0.1** | **0.3** | **0.03** | **0.2** | **0.1** | **0.3** | **0.1** | **0.4** |
| BY32 | 4557 | **0.1** | **0.3** | **0.1** | **0.2** | **0.05** | **0.2** | **0.1** | **0.3** | **0.1** | **0.4** |
| BY38 | 4208 | **0.1** | **0.3** | **0.1** | **0.3** | **0.1** | **0.2** | **0.1** | **0.4** | **0.2** | **0.4** |
| BY5 | 4559 | **0.1** | **0.4** | **0.1** | **0.3** | **0.1** | **0.3** | **0.1** | **0.4** | **0.2** | **0.5** |
| P2 | 4199 | **0.2** | **0.5** | **0.2** | **0.4** | **0.3** | **0.6** | **0.3** | **1.1** | **0.5** | **0.7** |

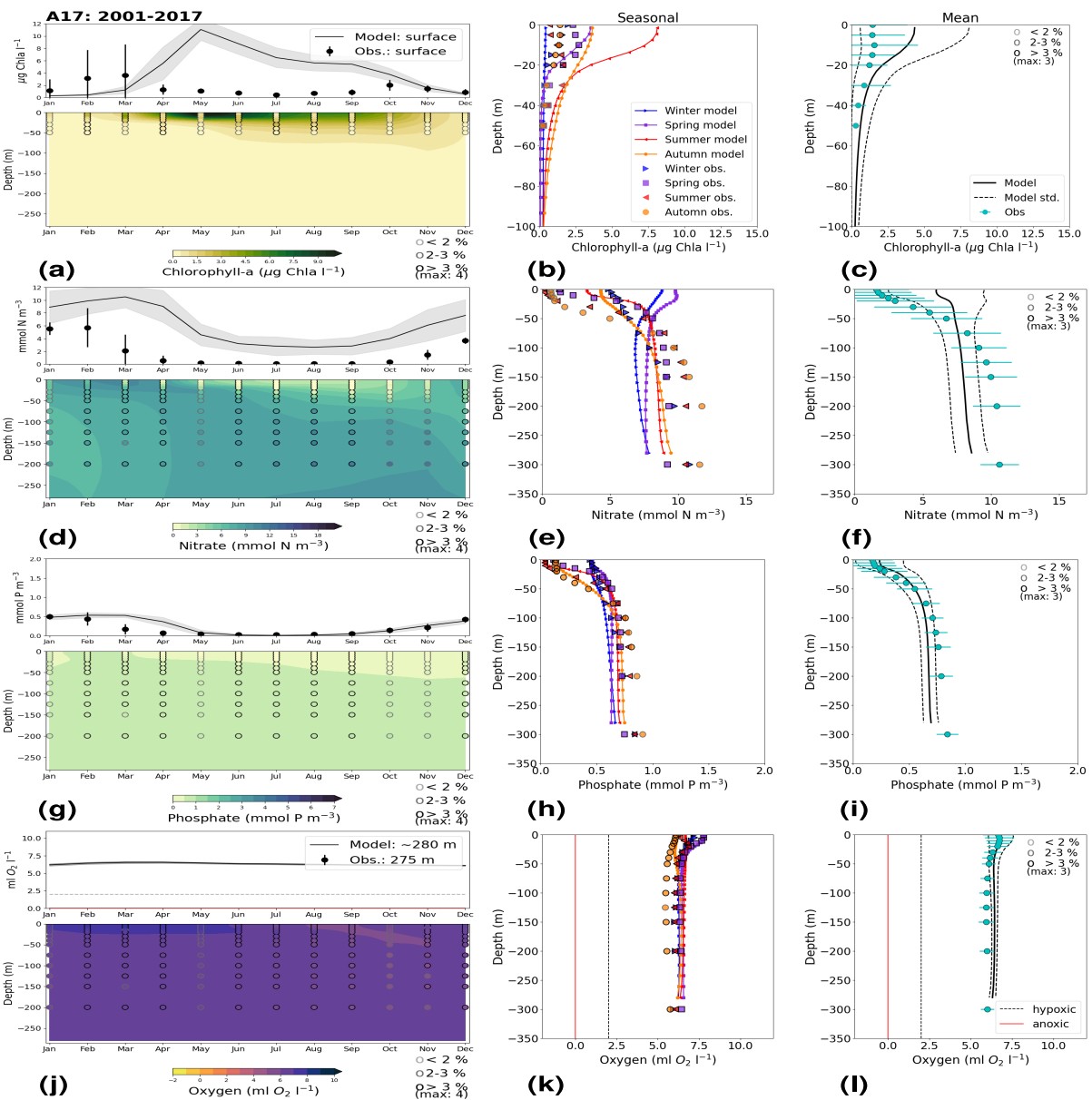

**Figure B1.** Monthly-, seasonal- and period-averages of the main biogeochemical variables at Å17 for 2001-2017. Variables are a-c) chlorophyll-a, d-f) nitrate, g-i) phosphate and j-l) dissolved oxygen for both model and observations. Monthly averages (a, d, g, and j) are shown over the entire water column (colors) and a close up for surface waters for all variables, except for dissolved oxygen where a close up of near bottom waters is shown instead. Near bottom is here considered to be the depth within the last model depth that has the most observations. The standard deviation in time for each averaged monthly value is shown for the model as a gray shaded area and as bars for the observations. The standard deviation of the period means (c, f, i and l) are also display for both model (dashed lines) and observations (cyan crosses). The observation coverage in all plots is shown as open symbols with shades of grays as indicated in the legend.

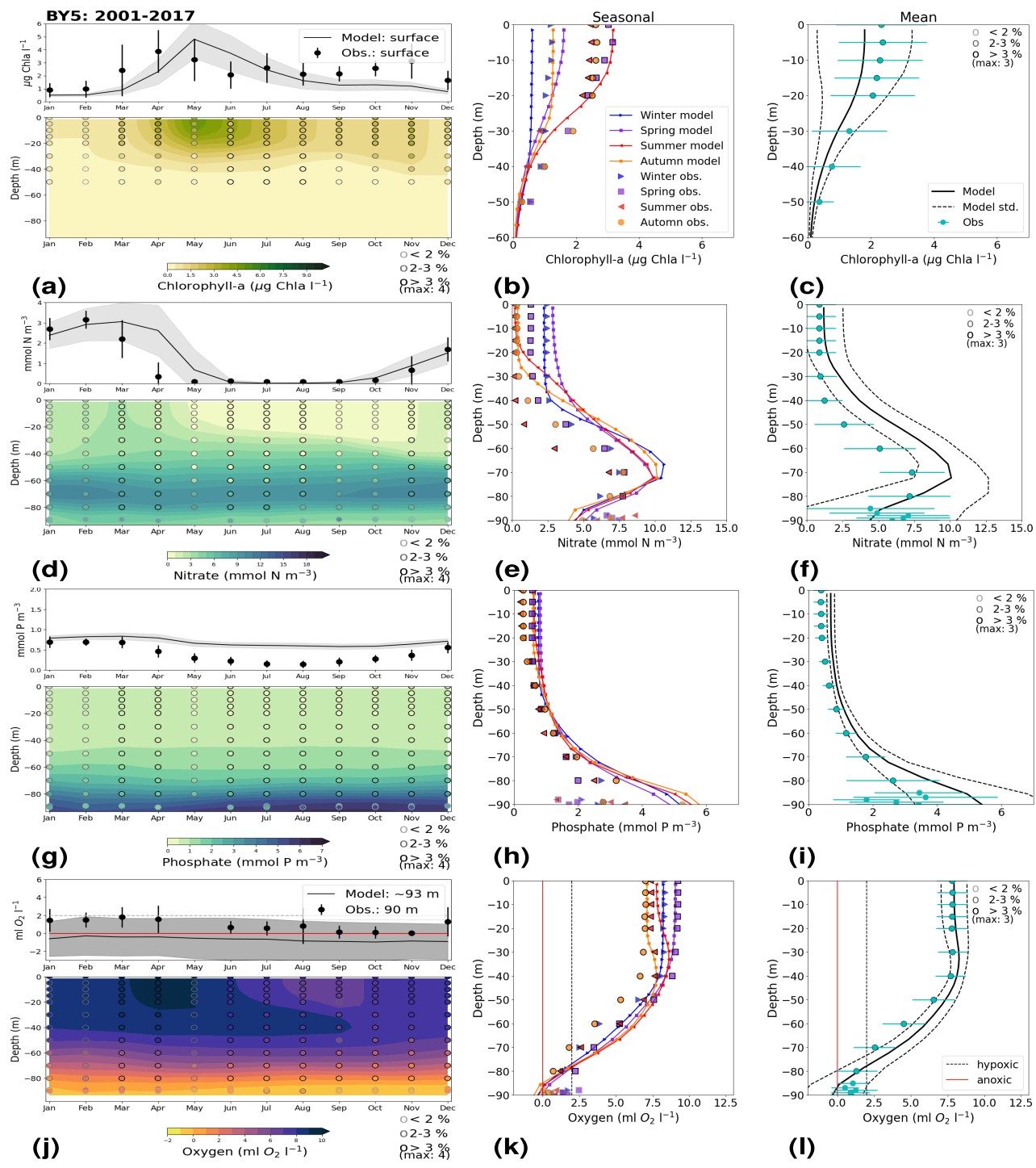

**Figure B2.** Annual and inter-annual variability of main biogeochemical parameters at BY5. Detailed description is as in Fig. B1

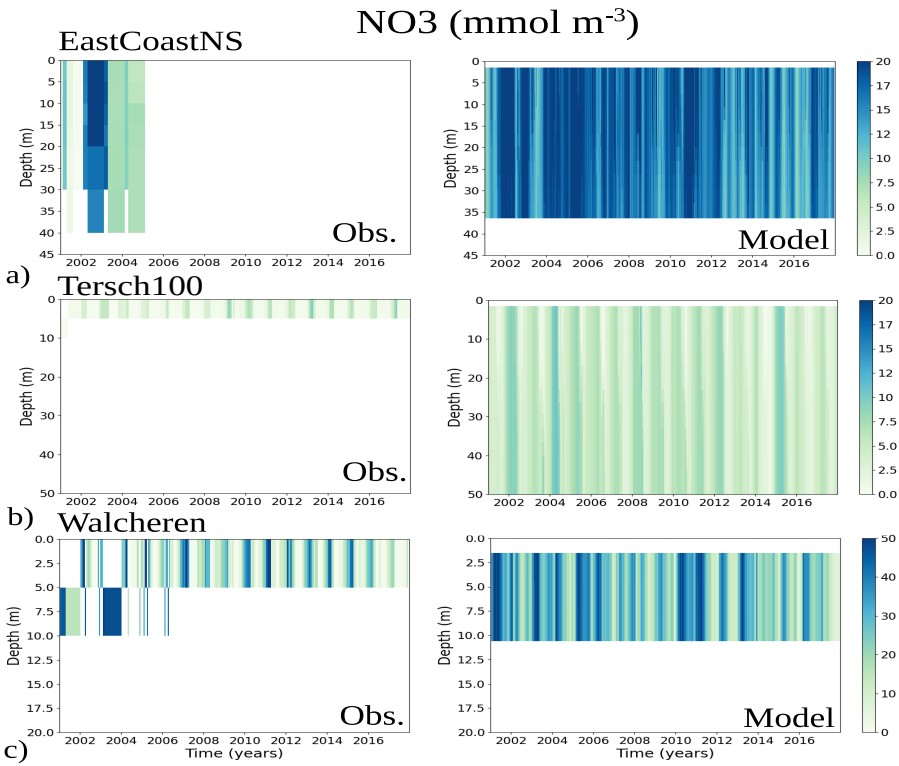

**Figure B3.** Water column nitrate for 2001 to 2017 from observations (left) and model (right) for 3 southern North Sea stations: a) East-CoastNS, b) Tersh100 and c) Walcheren. Walcheren is only represented by 2 vertical layers in the model, as NEMO-SCOBI is not meant to resolve such sallow waters.)

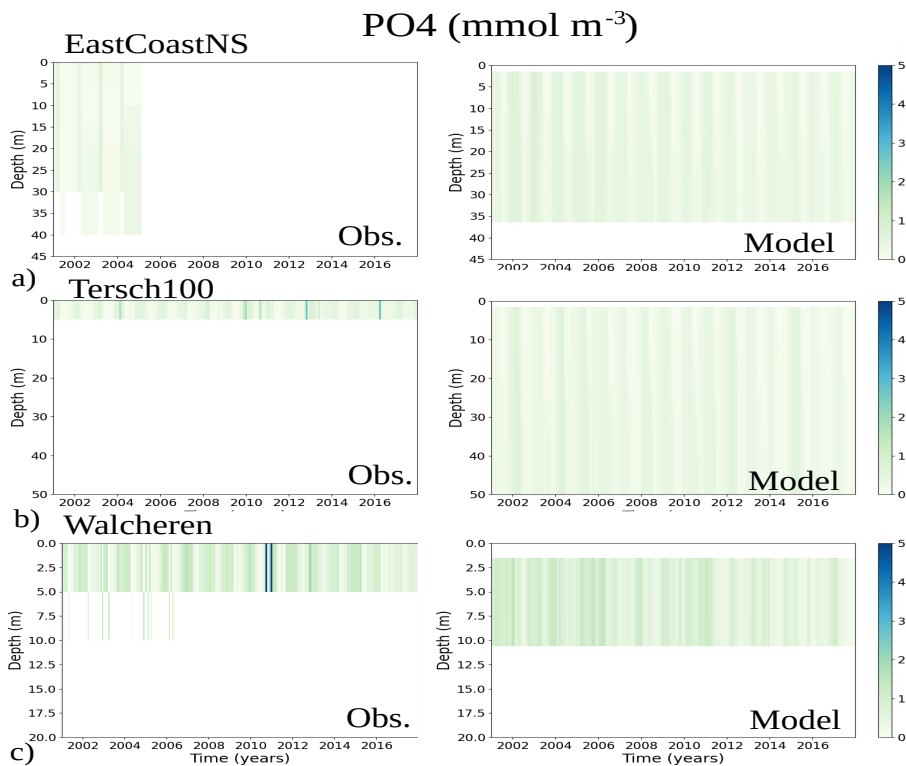

**Figure B4.** Water column phosphate for 2001 to 2017 from observations (left) and model (right) for 3 southern North Sea stations: a) EastCoastNS, b) Tersh100 and c) Walcheren.)

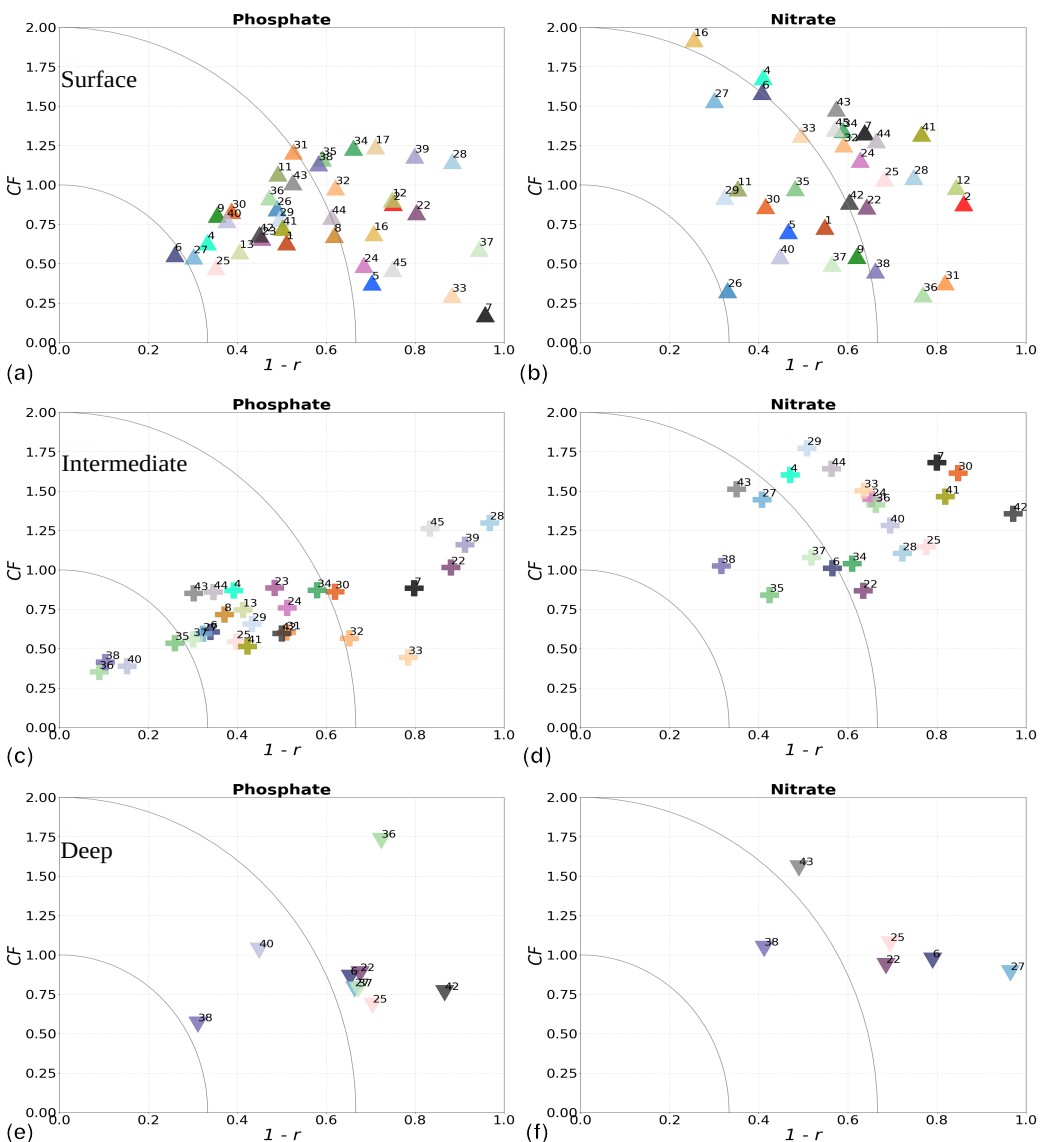

**Figure B5.** Model performance for phosphate and nitrate over the period 2001 to 2017 shown as a combination of Pearson correlation bias (1 - $r$) and Cost Function bias ($CF$) for the Baltic Sea-North Sea system evaluated per areas in Fig. 1 at surface (above 10 m), intermediate (in between 10 and 100 m) and deep (below 100 m) waters. Areas with too little number of observations are not evaluated (see section 2.2.3).

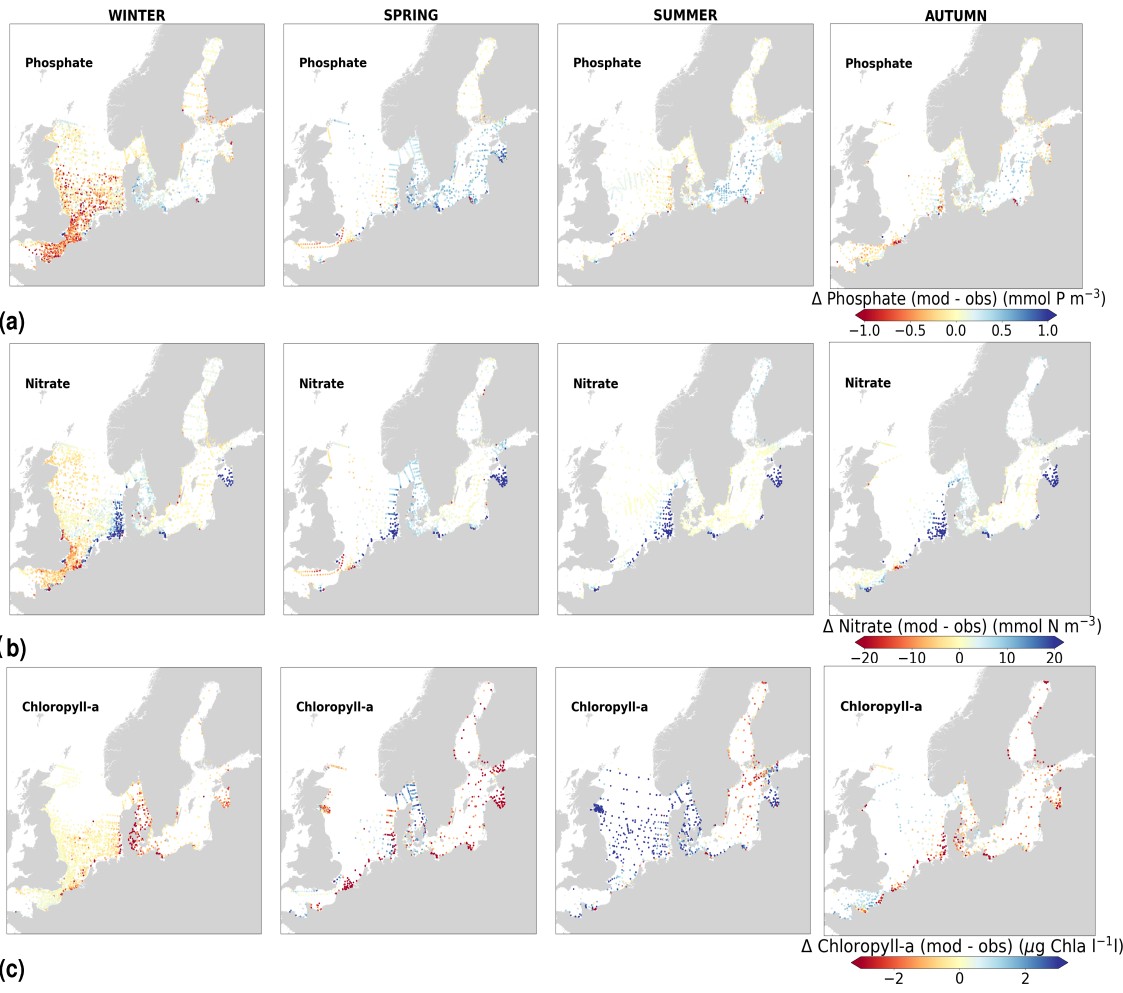

**Figure B6.** Seasonally spatial distribution of the difference between model results and observations for the period 2001 to 2017 for PO$_4$, NO$_3$ and Chlorophyll-a.

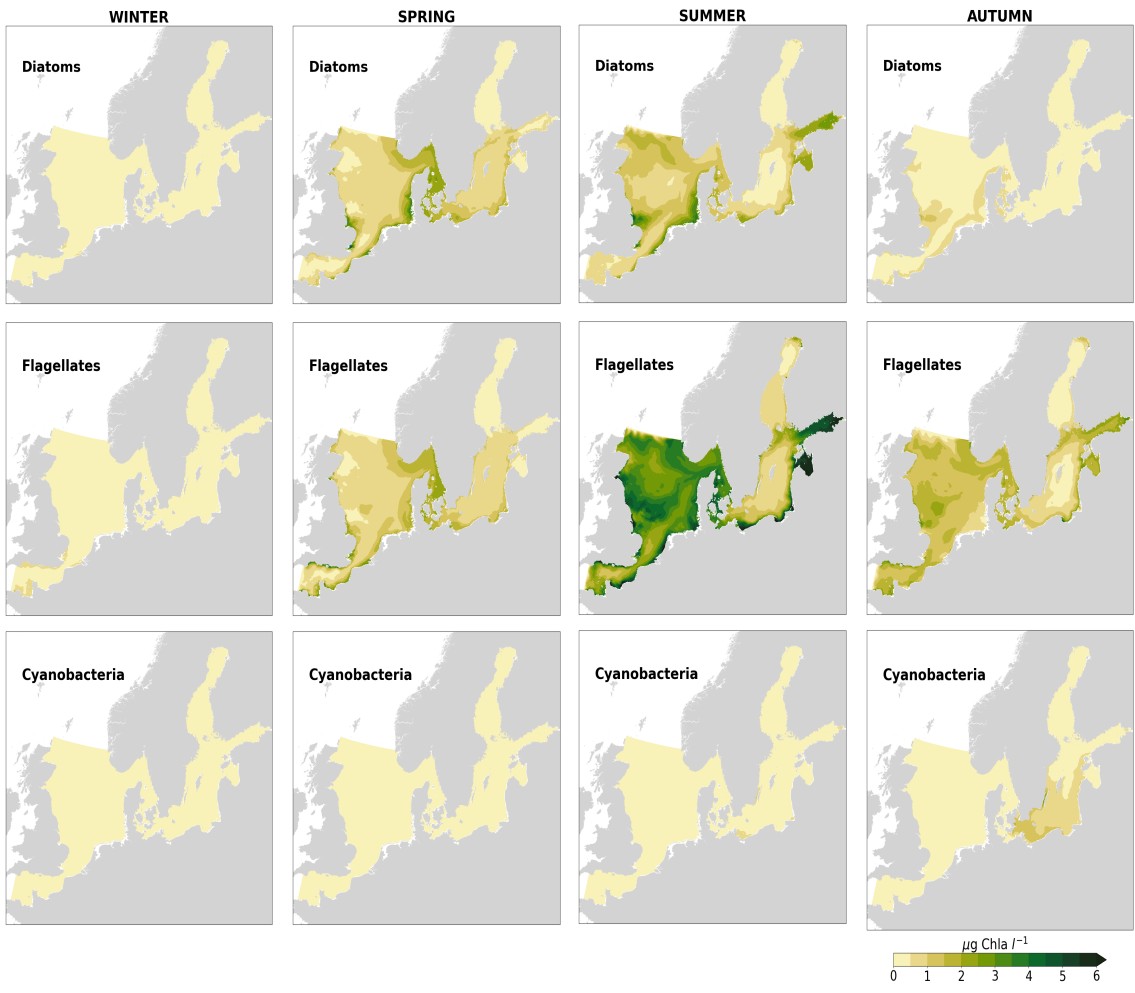

**Figure B7.** Seasonally averaged Production by Diatoms, Flagellates and Cyanobacteria in the model for period 2001 to 2017.

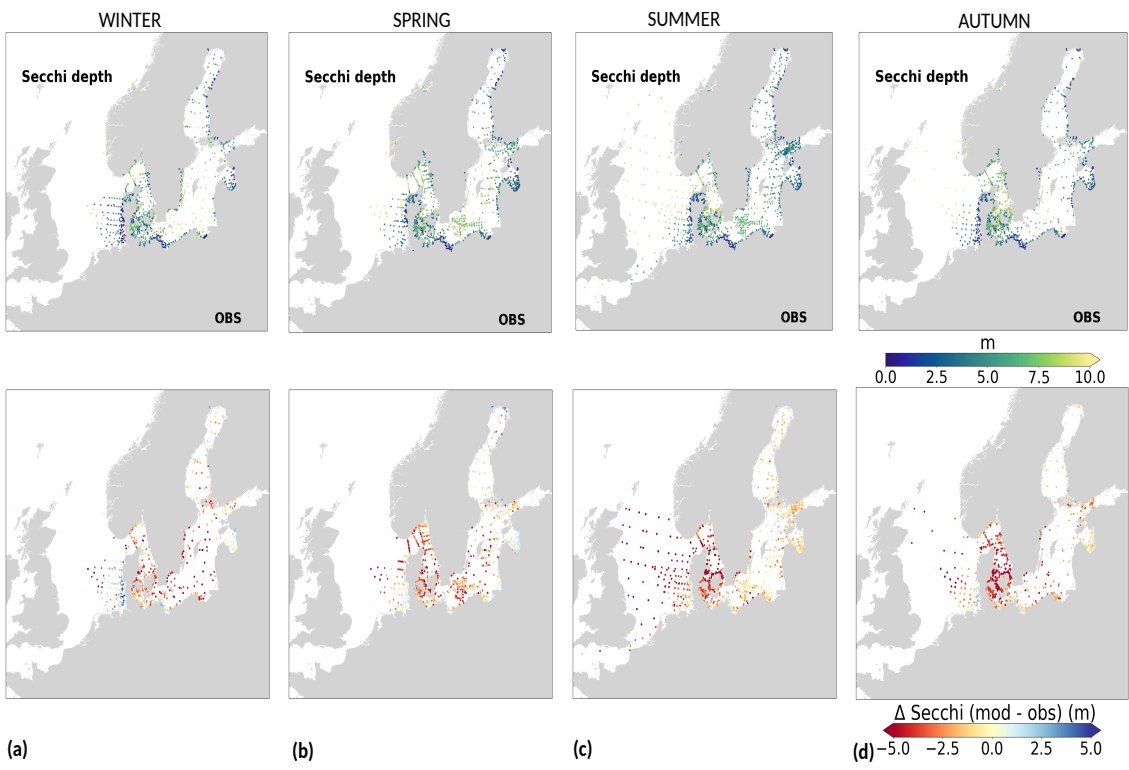

**Figure B8.** Observed light penetration depths shown as averages for a) winter, b) spring, c) summer and d) autumn of secchi depths and corresponding differences compared to model values for the period 2001 to 2017. Model secchi depths are approximated as the inverse of the light attenuation coefficient (secchi depth = 1.45/$kd$).

*Author contributions.* IK adapted the code of SCOBI to be coupled to NEMO. RH worked on the coupling of NEMO-Nordic to SCOBI. SF and FF set NEMO-SCOBI to the super computer and performed the first tests. IRB, LAr, STF and EAR designed the work. IRB, EAR, SEB, MG, MH, LAx, STF and JH were involved in setting the model forcing, performing and monitoring the run. IRB compiled observations and performed the main analysis work. All authors contributed in discussing and writing the manuscript.

*Competing interests.* We declare no competing interests

*Acknowledgements.* This study was funded by the Swedish Agency for Marine and Water Management (HaVs- och vattenmyndigheten; HaVs) within the framework OSPAR - 'ICG-EMO' and 'Spring oxygen and chlorophyll-a indicators in the Baltic Sea'. Additional financial support was given by the Swedish government via its climate adaption focus area. We would like to thank the working groups within OSPAR and HELCOM, in particular the 'intersessional correspondence group on eutrophication modelling (ICG-EMO)' for regular discussion supporting this work. We thank Kari Eiola and Nathan Grivault for support on technical model details.

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
