# Peer review of "Validation of the coupled physical-biogeochemical ocean model NEMO-SCOBI for the North Sea-Baltic Sea system"

_Biogeosciences, 2023_

## Referee Comment (RC3)

**Review** of the manuscript *Validation of the coupled physical-biogeochemical ocean model NEMO-SCOBI for the North Sea-Baltic Sea system*, by Ruvalcaba-Baroni et al, submitted to **Biogeosciences**

**Manuscript overview**

This manuscript provides an extensive overview of the validation performed on a new coupled model set-up. This set-up existed for the hydrodynamic model (NEMO-Nordic), but not previously for the biogeochemical model (SCOBI). The validation therefore focusses mainly on the biogeochemical part. Model performance is not bad for such a model, and model extensions are also reported fully in the manuscript and appendix. The manuscript has no other focus than presenting the validation, and concludes this model set-up is not better or worse than others previously published by other research groups. As such, it is deemed to be a valuable addition to model ensemble studies.

**Review overview**

The manuscript is in general well written (some grammatical errors remain) and includes an exhaustive validation exercise. The authors focus a lot of attention on the fact that this set up includes both the wider North Sea and the Baltic Sea: most model set-ups do either one or the other due to the different governing mechanisms. For this set up the biogeochemical model was extended, from its natural domain the Baltic, to cover the North Sea and Channel areas. Therefore is seems strange that the validation is mainly focused on the Baltic and the Kattegat/Skagerrak area (for which SCOBI was designed) and hardly on the new areas it now has to represent. The authors truthfully cite a lack of observational evidence in the newly covered areas, but some station validation is surely possible. The biogeochemical model tends to capture the phosphorous and oxygen dynamics pretty well, but this is what I would expect from a model designed for and tuned to the Baltic Sea. The North Sea has very different dynamics, and although hypoxia and anoxia can occur there as well they are not a defining feature of the modern system. The authors themselves state that the model performs better in P-limited areas than N-limited areas, and the North Sea is mainly the latter. Modelled phytoplankton consists of diatoms, flagellates and cyanobacteria, but the latter hardly play a role in the salty North Sea and Channel: what groups could be added for a better representation of phytoplankton in the North Sea? Would addition of *Phaeocystis spp*. be an option (the North Sea nuisance species), and how good is the model at representing primary production at pycnocline depth? Figures 7 and 9 seem to indicate an overestimation of the top mixed layer depth, which should be discussed more. Overall, the model misses the seasonal dynamics of both systems (e.g. timing of spring bloom, autumn bloom), which could be due to the light climate, silica dynamics, phytoplankton parametrization or the nutrient inputs (temperature is usually easy to get right). Without additional analysis it is hard to say what is the main cause, particularly as this might differ per region. But some light attenuation validation could be added, as could a comparison with continuous Chla observations (few locations, usually short temporal coverage) or comparison with Chla satellite observations to get a better grip on this issue. I miss an in-depth analysis and discussions on these topics in the current manuscript. But the manuscript itself is worth publishing, as the model represents a valuable addition to both North Sea and Baltic modelling efforts.

**Recommendation**

Moderate revision

I like the manuscript but would like to see further validation results added to it for 1. the North Sea area and/or Channel area (main) and 2. the riverine forcing used (appendix). This would require no new simulations but new post-processing. I would also like to see figures 6 and 8 reorganized and

figure B3 moved from the appendix to the main article. If necessary, figure 5 could be banned to the appendix to make way for figure B3.

**Detailed Comments**

1. Line 14-16: The validation is in agreement with assessment areas … ? Do you mean that you are using assessment areas for the validation (i.e. method), or that validation within these assessment areas confirms with reported values in those same areas (i.e. validation result)?

2. Line 19: the references are not in alphabetical or chronological order.

3. Line 44-45: too many comma's and bad grammar. Not sure what is meant here: "such areas" refers to the deeper parts of the North Sea (previous line), but those are not coastal.

4. Line 50: "rereferred to *as* cyanobacteria" and I miss a reference for the statement that cyanobacteria do not grow in the North Sea.

5. Line 78: bad grammar, I suggest ", which is particularly true in"

6. Line 80: "but *contain* biases for"

7. Line 87-90: Not sure why this text is here, not relevant to the subject of this manuscript.

8. Fig 1: I would say "observational SHARK stations". The acronym is later used with capitals, without those it is rather confusing here. Indeed, SHARK is only explained in line 205, so some explanation is due here.

9. Line 131: please refer to the figure before the textual explanation, to make it easier on the reader.

10. Line 148: do you mean that the phytoplankton parameters were tuned to represent both the Baltic and North Sea areas? That is to say, the parametrization the model had previously was tuned to the Baltic and these parameters did not fit with North Sea simulations and so needed adjustment? If so, can you speculate why this was necessary? What processes/groups/functionality difference is there between these areas that make this adjustment necessary?

11. Fig 2 : this is a spaghetti diagram, hard to read for the many arrows. I think it is a bad idea to include so much detail that the model visual abstract (which is what this is) becomes visually unattractive. I would leave out the coloured arrows explanation, readers can see for them selves if a flux stays in the pelagic or not. Or use different line styles. Maybe group it a bit more, with all pelagic nutrients together in a circle and 1 arrow going in and out if all nutrients are needed? And why are all the fluxes named in the caption rather than in a separate table?

12. Line 167: please provide a reference for the applied reduction factor, assuming this is a generally available dataset. If it is not I don't quite see why the authors would use this particular product.

13. I am getting a bit confused about the riverine forcing applied. Am I correct in thinking that you used

    - discharge values calculated by a hydrological model, which were adjusted evenly across the domain for a known, uneven model error?

    - nutrient values based on the same hydrological model, but adjusted for each year and basin to observational values based on two different observational data sets?

    If so, then I think it unlikely that any modeller could replicate your efforts, as they cannot replicate this forcing set. And it makes me wonder why the observational sets were not used directly. This mix up of 3 different sources complicates interpretation of results, which are reported in eutrophication-relevant variables. Please provide more detail on this forcing set

(in an appendix), as well as a comparison for a few selected rivers (e.g. some of the larger dots in figure 3) of the applied discharge and nutrient loads compared to the observations that you state you also have. Can some of your mismatches in coastal zones be related to this forcing data?

14. Line 233: the reference year was chosen because of high nutrient values. Where those simulated values or observational values?

15. Line 245: explanation of the applied seasonal delineation (meteorological? astronomical?) is only given in the caption of figure 10. Please provide this here.

16. Line 280: this has been stated before.

17. Section 3.1: the authors should include a North Sea station here, maybe one on the Danish or Dutch transects or an individual research station from the UK. It may not have everything the authors want but an extension of the SCOBI model into the North Sea and Channel areas should be validated in detail there. Stations like the Oystergrounds (NL), West Gabbard (UK) or L4 (UK) spring to mind, though the latter is I think just outside of the domain. These may have standard surface monitoring and limited at depth monitoring, but it is better than nothing. They also have high resolution observational data for a few years, generally. In the very least a North Sea station comparison will provide more detail on the local Chl-a seasonal signal and bloom timing capacity of the model there (difficult to derive from figure 12).

18. Line 315-317: can you provide an overview of the trends in table form in the appendix? Now it is hard to see and compare trends.

19. Figures 6 and 8: I would suggest restructuring these. A label over results in a graph is a no-go, in any case. Suggestion: make a two column graph (which these are already). Top left: the legend for surface values. Rest of left: surface graphs for T, S, NO3, PO4, Chl-a. Top right: legend for bottom values. Rest of right: bottom graphs for T, S, NO3, PO4, O2. The legend for O2 can be removed and explained in the caption. This would make the graph more accessible as surface or bottom processes can be viewed at a glance (vertically) while top and bottom values can still be compared easily (horizontally).

20. Line 330-333: "no guarantees that the measurements did not fail to", the double negative here makes this sentence hard to read. I presume your point is that observational evidence is discrete in time and so can easily miss the peak of the spring bloom. This is a very valid and important point to make, which merits unambiguous text.

21. Line 350-354: the model correctly predicts inflow of North Sea waters into the Baltic proper, though bottom temperature and salinity values are too low compared to observations. But this is a feature of the existing hydrodynamical model, NEMO-Nordic, which was already used in the presented domain before, and calibrated and validated there. I would not expect the extension of the SCOBI model to influence these dynamics.

22. Figures 7 and 9: the little cyan plusses (not "crosses" as it says in the caption, that would be "x") are very hard to see. Can this be done by shading instead? I do love the surface values on top of the depth graphs, very nicely done!

23. Line 435: figure B3 is mentioned here, I would prefer to see figure B3 in the main text rather than in the appendix. If there is a limited number of figures allowed, I would suggest moving figure 5 to the appendix instead, as it does not show simulated results. Within B3 the markers are very hard to see, can you make then larger? Some of the colours are quite light, resulting in a number without a visible marker in my printed version: enlargement might help with this too.

24. Line 463: "in the Baltic Sea, four HELCOM-OSPAR assessment areas". Surely these are HELCOM assessment areas?

25. Line 477-4780: surely you can see in your simulation results if accumulation happens or not?

26. Line 483-485: this is an important message for the monitoring organisations, please make it stronger.

27. Line 492-493: grammatically incorrect sentence and it doesn't make much sense.

28. Line 494-495: surely this is not about which model is better? Grammatically also incorrect, I assume the model by Daewel et al captures the southern coast of the North Sea just fine. Maybe not in biogeochemical terms, but the coastline itself is in the model so it captures it.

29. Line 505: sentence is too long and loses it grammatical structure by the end. Please rephrase.

30. Line 513-516: please speculate on what the missing process for phytoplankton growth could be. And add riverine nutrient validation to the appendix (e.g. figure 3 but with an applied forcing-observational evidence focus), to better quantify the nutrient input issue. How well does your input capture suspended matter from fluvial sources?

31. Line 524: have you considered the following works?

    *Capuzzo, E., Stephens, D., Silva, T., Barry, J., & Forster, R. M. (2015). Decrease in water clarity of the southern and central North Sea during the 20th century. Global change biology, 21(6), 2206-2214.*

    *Capuzzo, E., Painting, S. J., Forster, R. M., Greenwood, N., Stephens, D. T., & Mikkelsen, O. A. (2013). Variability in the sub-surface light climate at ecohydrodynamically distinct sites in the North Sea. Biogeochemistry, 113, 85-103.*

    And how does this work relate to your findings?

32. Line 526: you mean the Rhine, arguably the largest river to exit into the North Sea, has no influence here? Surely not.

33. Line 532-535: figure 12 shows no observational support for this. How do you know your model is not simply overestimating the local light climate?

34. Line 542-544: maybe, but a comparison with satellite observations could verify this point better spatially.

35. Line 547: in figure B4 the matching points are hard to see as they are white, overemphasizing the discrepancies. Can you use a blue-yellow-red colourbar here to emphasize where model and observational evidence do agree, and where there is simply an observational dessert? The same applied to figure 13, where observational points with a N:P ratio of (near) Redfield values are invisible.

36. Line 570-575: spring bloom timing is mainly driven by temperature and light conditions in the North Sea, so a discussion on the simulated light climate is due here. Diatoms have evolved to be more light receptive than most other phytoplankton species, so they lead the spring bloom. Does the biogeochemical model allow for a proper succession of species? Figure B5 indicates it does, but a general seasonal succession graph (daily resolution, maybe for the different basins) would be better to display the model's inner workings. A difference of 3 months in spring bloom timing is a lot, even for a large scale biogeochemical model.

37. Figure 12: I love this graph but at the current size results are hard to compare to observational evidence. Can these graphs be enlarged? The colourbars are also hard to read.

38. Line 580: "allows *for a study of the* North Sea"

39. Line 584: *", rather than prescribed boundary conditions"*

40. Line 585: again, this should not be a model contest on who performs best.

41. Line 595-597: you have shown that your model is capable of simulations from which you can derive relevant indicators for HELCOM and OSPAR, taking into account model performance

and bias. Certainly for Chl-a there would be caveats, but most models have these. But you have not shown that the model can be used for climate projections with specific relevant improvements, as you have not made these improvements yet. And there is no detailed information in the manuscript on what these improvements would be: several ideas have been floated but there was no priority list of "things to implement in the model". I would remove the latter part of this statement. For example, line 650 list improving the seasonal cycle of benthic denitrification, but contains no statement to how important this is with regard to other suggested improvement (e.g. cyanobacteria life cycle inclusion), or how this will be achieved.

42. Line 596-597: this is why I want to see a validation of the applied riverine forcing. The atmospheric deposition bias was discussed, but the reader lacks information on the riverine input bias.

43. Line 613: how about suspended sediment?

44. Line 633: please provide references for the claim that the model compares well with previously published estimates (assuming you mean other publications than Dalsgaard et al, 2013).

---

## Author Comment (AC1)

Note: Authors answers are given in bold

**Reviewer #1**

The paper of Ruvalcaba-Baroni et al. represents an important step forward in coupled physical-biogeochemical modeling of the North Sea and Baltic Sea regions as a single domain. The combination of NEMO with SCOBI is logical and according to the authors knowledge is only the third such effort from the modeling community after the papers of Maar et al. (2011) and Daewel & Schrum (2013), based on DMI-BSHcmod/ERGOM and ECOSMO, respectively. Given the need for model ensembles to improve overall understanding of biogeochemical functioning of marine systems, this study is very relevant, even more so with current efforts to better integrate management of the two seas under study.

The paper is generally of very high quality and builds upon the base of previous work in the same group with respect to both the development of NEMO-Nordic and SCOBI. Therefore I do not see any fundamental problems with the research and support its publication. That said, it is interesting to observe the discrepancies between SCOBI-modeled parameters and observational data in certain regions, which show the current limitations in knowledge and hopefully will guide the authors towards future improved iterations of the model. Some of these discrepancies deserve a bit more elucidation in the text, or at least better structuring of the sections, so as not to leave the most important discussion points to the end of the paper. There are also a small number of technical clarifications I would draw attention to, and some suggestions for alternative phrasing and setting the context of the study. These may all be considered minor revisions so I give them as a Line-by-Line list below.

Kind regards,

Tom Jilbert University of Helsinki, Finland

**We thank Tom Jilbert for nice words and accurate comment/suggestions, which we address in details below and will improve our manuscript.**

1) Line 45-50: Not clear why primary producer assemblages are mentioned here for NS but no equivalent description for BS. I suggest to introduce the physical aspects first, then biogeochemical and finally the plankton assemblages.

**We indeed only mention cyanobacteria in the Baltic Sea, as these are the most problematic species there. However, we will restructure the paragraph and add more details on primary producers in the Baltic Sea.**

2) Line 66: Replace "recycling of benthic phosphorus minerals" with "recycling of phosphorus from sediments"

**We will change the wording as suggested.**

3) Line 70-71: Please update this setting of the context with a citation of the following publication and potentially references therein: "The Baltic and North Sea Strategic Research and Innovation Agenda BANOS SRIA 2021 The final BANOS SRIA draft of the proposed, new, joint Baltic and North Sea Research and Innovation Programme -BANOS BANOS CSA Deliverable 1.5".

**We will adjust the context to include this relevant work (Koho et al., 2021).**

4) Caption of Fig. 1: Capitalize SHARK

**We will capitalize this.**

5) Table 1: Modify mg CHL m-3" to mg Chl-a m-3"

**We will change this.**

6) Figure 2 legend: Typo in atmosphere

**We will correct this.**

7) Line 183: Typo 100 kton N/yr

**We will correct this.**

8) Line 195: Does this mean "reduced to 0.3 x and 0.75 x the original value, respectively"? Please clarify. Also clarify (as implied in Fig. 2) that the resulting nutrient flux to the sea is entirely in the dissolved fractions and comment briefly on simplifications with respect to e.g. fluxes of TP, that in reality are largely particulate.

**Yes, the reduction factors are: 0.3 x detritus P, and 0.75 x detritus N. This means that we have reduced the riverine input of particulate organic phosphorus and nitrogen (in our model assumed to be detritus of nitrogen and phosphorus) by 75% and 30%, respectively, to account for coastal retention and bioavailability. This is based on values at the Swedish coast (Edman and Anderson, 2014; Eilola et al., 2009). These factors are largely unknown for most of other coastal areas in the model domain and therefore the same reduction factors are used throughout the domain. We will ensure that this is clear in the text. Regarding the resulting nutrient flux, figure 2 shows the total**

**phosphorus and nitrogen (so the sum of dissolved inorganic and detritus) that enters the model domain. We will ensure this becomes clear in the text. Please also see our reply to comment 30.**

9) Line 286: Modify to e.g. "depending on their proximity to one another"

**This will be changed as suggested.**

10) Line 301: Typo "to analyze"

**This will be corrected.**

11) Line 315: Replace 80s with 1980s

**This will be changed as suggested.**

12) Line 317: Typo observations are too low

**This will be corrected.**

13) Line 326: Replace dont with do not

**This will be corrected.**

14) Line 346: Typo statistically

**This will be corrected.**

15) Line 347: Remove "not studied here", it confusing to state this

**This will be removed as suggested.**

16) Line 349: Rephrase to e.g. "There is a lack of observational data for bottom water oxygen during the period 1975-1995".

**There is indeed a lack of observations before the 1995 at ANHOLT and at most stations in the Skagerrak-Kattegat region. However, it is not limited to oxygen. While nitrate observations are totally lacking before ∼ 1995, phosphate and oxygen observations are scarcer (with fewer data points per year) before the year around 1995. Note that the year where observations become more abundant is different at each station in the Skagerrak-Kattegat, but fall between 1992 to 2000. However, we agree that the sentence is confusing. This will be rephrased as follows:**

**"In the Skagerrak-Kattegat region (e.g., at ANHOLT, fig. 6) observational data is largely lacking before 1992-1995, including that for oxygen in bottom waters and therefore model trends may be more representative of the system for historic values."**

17) Line 407: Typo less than

**This will be corrected.**

18) Line 420: Typo extent

**This will be corrected.**

19) Line 421: Modify to The main inference of...

**This will be corrected.**

20) Line 424: Modify to "Despite such specificities of..."

**This will be changed.**

21) Line 457: Modify to its (no apostrophe)

**This will be corrected.**

22) Line 471: Modify to "bias is" or "biases are"

**This will be corrected.**

23) Line 497: Modify to most models

**This will be corrected.**

24) Line 572: How could low oxygen theoretically inhibit primary production? Not easy to understand what is meant here.

**"low oxygen" will be removed from the sentence.**

25) Line 574: The model vs. data discrepancy in the timing of the late winter/spring bloom in the Kattegat is one of the key question marks raised in the study, and indeed it is highlighted in Fig. 7. I think it deserves more elucidation at this point in the discussion, for example an assessment of the degree to which light or temperature might be inhibiting the early onset of the bloom in the model. The authors return to this in Section 3.6 Future work and data gaps but in the current version the reader is left hanging for an explanation after it is established that there is no nutrient limitation at the time in question.

26) **We will add, most likely in appendix, plots of the sensitivity of modelled phytoplankton growth to light and temperature. We will then add more information on it's relevance in the text here. Please note that we attribute the bloom delay occurring in the Kattegat-Skagerrak mostly to the fact that the model does not capture the seasonal variations of the light attenuation depth in this area. In addition, the model temperature is underestimates during spring at for example ANHOLT (by approx. two degree, see figure 1 below). This indicates that the stratification in the Skagerrak-Kattegat is not sufficient in the model during spring and may inhibit phytoplankton growth at the exact time of the blooming period. We realize that we have not included the latter in the text and will be added as a discussion point. Please also see our reply to reviewer #3, on pages 4 and 5, comment 0.8.**

[Figure]

Figure 1: Seasonal profiles averaged over 2001 to 2017 for temperature at ANHOLT, where blue colors are for winter, purple for spring, red for summer and orange for autumn.

27) Line 590: Again please check the BANOS SRIA for up-to-date statements about the need for integrated modeling of BS-NS system

**Yes, we will.**

28) Line 595: Citations in brackets

**The missing brackets will be added.**

29) Line 604: Modify to "in Ford et al. (2017)..."

**Brackets for this citation will be removed.**

30) Line 616-623: This is a very important section of the discussion: the model assumes (if I understand correctly) that a certain bioavailable fraction of nutrients enters the sea, directly into the dissolved phase state variables PO4, DSi, NO3, NH4 (Fig. 2). This is of course a large simplification of reality, where there is transfer between bioavailable and non-bioavailable fractions within the coastal filter, as well as heterogeneous removal of nutrients. Different types of coastlines may behave quite differently in this regard, see eg. Asmala et al., L&O 62 (2017). Are there any of the observed model vs. data discrepancies e.g. in near shore nutrient or Chl-a concentrations, that could be affected by this simplification? If so, it would deserve some mention higher up in the discussion for those specific areas.

**The bioavailability factors are only used on the particulate organic matter that enters the sea with the rivers. The factors are to one part accounting for the different quality of the organic matter from land, of which some part is refractory and will not decompose fast enough to be a part of the marine nutrient circulation within the model system. In the ocean model, this factor also accounts, to some extent, for the filtering effect of the coastal zone. In SCOBI, the bioavailable detritus (particulate organic material that has been decreased from the rivers) enters the sea into**

the detritus pools for the specific nutrient and then go through the transport, grazing, sinking and decomposition processes. Thus, the bioavailable fraction of the organic pool does not directly enter the dissolved nutrient (see figure 1 in the main text). We will better clarify this in the text.

Regarding the filtering effect, the reviewer is correct that different types of coastlines have different response to nutrient removal. The bioavailibilty factors used here are typical values for the Swedish coast and the Baltic Sea Eilola et al. (2011) , which are also not too far from those in Asmala et al. (2017) (16% of nitrogen and 53% of phosphorus removal from land versus 30% and 75% assumed here). These factors are, however, poorly quantified for the North Sea. As a first approach we assumed a homogeneous value for our entire domain following the approach of Eilola et al. (2011). In addition, we did perform a scenario with the exact same settings as that presented here, but without any retention factor. When comparing results from the latter to the hindcast scenario here, the spatial effect of the retention factor was minor, especially for nitrogen which decreased by one or two mmol/m3 almost homogeneously in our domain. The chlorophyll-a was not much affected by this and therefore we dedicated only minor discussions to this in lines 620-624. We think that the major effect of this is confined to specific coastal areas, probably those limited by P. However, we agree with the reviewer that this is relevant and deserves more explanation in the text. This also links to our reply to reviewer #3, comment 12 and 13 where we say that we will look further to better detect which coastal regions are most affected by the rivers. We will also add the reference of Asmala et al., 2017 in line 621.

31) Line 666: typo therefore

This will be corrected.

32) Appendix Line 737: Important: should this read "BOPREM-BIP decreases with increasing salinity and decreasing bottom oxygen concentrations..." ? That is the implication of the equations and following text, and the logical relationship.

Yes, that is correct and 'decreasing' will be added before 'oxygen concentrations'.

33) Appendix Line 771: Typo from

This will be corrected.

34) Appendix Table A2: Comment on the validity of using a single porosity value for entire NS-BS system.

We are fully aware that porosity is not homogeneous in the Baltic Sea and the North Sea - see for example a model study of benthic phosphorus cycling in the Baltic Sea by co-authors of this study, where porosity was region-specific (Almroth-Rosell et al., 2011, 2015). The results presented here are, however, from the first version of NEMO-SCOBI which indeed is simplified and has currently a hard-coded porosity. It is definitely an important point in our list of future development for NEMO-SCOBI, where it will be made regional specific. A comment on this will be added in the text.

**References**

Almroth-Rosell, E., Eilola, K., Hordoir, R., Meier, H. M., and Hall, P. O. (2011). Transport of fresh and resuspended particulate organic material in the Baltic Sea - a model study. *Journal of Marine Systems*, 87(1):1–12.

Almroth-Rosell, E., Eilola, K., Kuznetsov, I., Hall, P. O., and Meier, H. M. (2015). A new approach to model oxygen dependent benthic phosphate fluxes in the Baltic Sea. *Journal of Marine Systems*, 144:127–141.

Asmala, E., Carstensen, J., Conley, D. J., Slomp, C. P., Stadmark, J., and Voss, M. (2017). Efficiency of the coastal filter: Nitrogen and phosphorus removal in the Baltic Sea. *Limnology and Oceanography*, 62(S1):S222–S238.

Edman, M. K. and Anderson, L. G. (2014). Effect on pCO2 by phytoplankton uptake of dissolved organic nutrients in the Central and Northern Baltic Sea, a model study. *Journal of Marine Systems*, 139:166–182.

Eilola, K., Gustafsson, B. G., Kuznetsov, I., Meier, H., Neumann, T., and Savchuk, O. (2011). Evaluation of biogeochemical cycles in an ensemble of three state-of-the-art numerical models of the Baltic Sea. *Journal of Marine Systems*, 88(2):267–284.

Eilola, K., Meier, H. M., and Almroth, E. (2009). On the dynamics of oxygen, phosphorus and cyanobacteria in the Baltic Sea; A model study. *Journal of Marine Systems*, 75(1-2):163–184.

Koho, K., Andrusaitis, A., Sirola, M., Ahtiainen, H., Ancans, J., Blauw, A., Cresson, P., Raedemacker, F., et al. (2021). The Baltic and North Sea Strategic Research and Innovation Agenda, BANOS SRIA 2021. *BANOS CSA D*, 1.

---

## Author Comment (AC2)

**Answer to reviewer 2: Review of the manuscript Validation of the coupled physical-biogeochemical ocean model NEMO-SCOBI for the North Sea-Baltic Sea system by Ruvalcaba Baroni et al., (2023)**

Note: Authors answers are given in bold

**Reviewer #2**

I have completed the review of the manuscript Validation of the coupled physical-biogeochemical ocean model NEMO-SCOBI for the North Sea-Baltic Sea system by Baroni et al., (2023). In this work the Authors describe a run made using the newly ocean-biogeochemical model NEMO-SCOBI that has been specifically developed and tuned for the North Sea-Baltic Sea System. I found the manuscript well written and the validation of the model properly carried out by the Authors. On the other hand, I think that the manuscript needs some major revisions whose reasons are listed below

**We thank reviewer #2 for his time and useful comments, which we address in detail below.**

Comments

1. While I was reading the manuscript, although it is clear that having a coupled model for the North Sea-Baltic Sea represents a clear advancement for the scientific community, it was not clear to me if its performances are better or worst that other 3D modeling tools available for the region that are cited by Authors at line 80 for example. Are the performances of NEMO-SCOBI better or worse than other modeling tools? Does NEMO-SCOBI improve the biases of other modeling tools or not? Im asking that since a potential user should have all this information to decide to use your modeling tool/simulated data than others provided by another modeling tool.

**As mentioned in lines 450-455 and specified in line 585, our results show that NEMO-SCOBI is neither better nor worse than previous model results depending on the variable and area. This means that in some areas our model performs better than other models for a specific variable and in others not (details are in sections 3.2 and 3.3). Importantly, the aim of this paper is to evaluate NEMO-SCOBI, not to make a model intercomparison, which requires a total different methodology and extensive additional work (see for example Gröger et al., 2022). Because no model is perfect, the best way is to combine results from several models and use ensembles, when possible. Currently, there are only three biogeochemical-ocean models (including NEMO-SCOBI) that account for both the North Sea and the Baltic Sea, and therefore our work represents an important step to create representative ensembles from independent models that cover both the North Sea and the Baltic Sea. We have also discussed in detail (in section 3.4) the strong and weak points of NEMO-SCOBI model, as well as its possible application as a stand alone model. These points are also summarized in section 3.5. As concluded in line 663, NEMO-SCOBI can fully compete with previous models that account for the North Sea and the Baltic Sea. However, we will read through the text to see where we can make this more clear and extend section 3.5. One of the major benefits in a model covering both North Sea and Baltic Sea is that the open boundary is moved out from the transition zone between the seas (as mentioned in lines 580-581). This is a very important step to better study the Baltic Sea and the Skagerrak-Kattegat area as it allows to study these areas without interactions with the open boundary. We will highlight this more in the text.**

2. Strictly correlated with that: please provide more quantitative information about these performances. Good, Comparable, Acceptable and so on are not informative from my point of view. Please quantify in the manuscript the biases and the values of the trends and their statistical significance.

**As written in section 2.2.3, and mentioned in lines 265-270, the words "good" and "acceptable" have a statistical value attached to them that is also used in other studies - see for example Edman and Anderson (2014); van Leeuwen et al. (2023). They are specified according to the Pearson correlation coefficient (r) as 1-r and the cost function (CF), which are statistical methods broadly used for model validation against observations. In summary, the model performance is considered to be "good" when both 1-r and CF of an assessed parameter fall within an inner quarter circle (see figs. 10 and 11). If the CF evaluation is equal to or lower than 1, the model values are very similar to the observed values. A value of 1-r that is equal to or lower than 0.33 means that the variability in the model for the evaluated parameter matches well that of the observations. When combining both 1-r and CF, if values fall withing the inner circle, it means that both the absolute values and the variability of the model are in good agreement. The model performance is considered to be "acceptable" if the combined value of 1-r and CF fall within the inner and outer quarter circle (i.e. when 1-r and CF values are in between 0.33,1 and 0.66,2). If 1-r and CF values are larger than 1 and 2, respectively, the model data is far from observations, meaning that there are significant differences in both absolute values and variability between model and observations. We will add clarifications where relevant in the results section. Note that the value of the absolute bias is also given where relevant in the text (e.g., line 308 and line 357).**

**Regarding the trends, the regression analysis and the corresponding p-values (which give the statistical significance) are calculated and mentioned where relevant in the results (e.g, lines 319, 321, 326, 346 and 364). We do not see the need for more statistics in this case, as the goal of this paper is not to detect specific temporal trends, as mentioned in line 234.**

1) Line 11: What do you mean with differences? please explain

**We mean that the model values and the values given by measurements for chlorophyll-a and nitrate differ considerably in the mentioned areas both in time and space. We will rephrase this sentence as follows:**

**"However, there are important differences for chlorophyll-a and nitrate between model results and observations in coastal areas of the southeastern ..."**

2) Line 26 : Modelled or Observational studies? Please explain

**Previous biogeochemical studies in the Kattegat-Skagerrak; both modelling and observations. This will be added into the text.**

3) Line 31-40 : What about the intermediate layers of the basin?

**This paragraph will be adjusted (also in response to reviewer #1, comment 1. More detail information on the water column structure will be added.**

4) Line 54: What do you mean with point sources. Please explain

**A point source is any single identifiable source of pollution from which pollutants/nutrients are discharged, such as sewage. This clarification will be added in the text.**

5) Line 70-76: Please provide more information about these differences since it is the starting point for showing that this model is a step forward for the scientific community

**We do not fully understand what differences the reviewer is referring to. If the reviewer refers to the difference between previous studies and our study, this is mentioned in lines 74-84 - i.e., our model covers both seas and only 2 other 3D models have this similar domain. In models that cover either the Baltic Sea or the North Sea, the Skagerrak or the Kattegat represent the boundaries of the models, which implies having simplified dynamics in these areas (this is written in lines 579-585 and also mentioned in our conclusions, line 659). Having several models covering the same area is a big advantage, as each model comes with their respective pros and cons. We have a dedicated section where we discuss the advantages that NEMO-SCOBI provides. Therefore, we will not add information on this in the introduction, but we will better highlight this in section 3.5. Please also see our reply 0.1 to reviewer #3.**

6) Fig.1: It would be great to have super imposed here the bathymetry of the domain (eventually using contours)

**Adding the bathymetry superimposed to this map it will be too messy and confusing. However, we will consider to add the bathymetry next to this figure or in appendix.**

7) Line 107: Did you assess the drifts in the tracers to assess whether 14 years are (lets say) enough for the spin up or not? In many ocean/biogeochemical models even 30 years are not enough to stabilize the numerical solutions.

**Yes, the years that are actually considered for validation correspond to those with less drift for biogeochemical parameters. The exact year is difficult to pick due to lack of observations before the 1980s. Therefore, we also performed several other runs also starting from 1961 and compared the drift. The initial drift generally decreased around the year 1975 in all runs. In addition, the biogeochemistry was not initialized from scratch - i.e. with homogeneous 3D tracer fields. The spinup was started from earlier SCOBI runs that represent already the physical-biogeochemical conditions as imprint in the initialization followed by the actual spinup. We will better clarify this in the text, line 168.**

**For the physical initial conditions we used restart files for the year 1973 from the simulation in Hordoir et al. (2019) that were the closest to the observations for physical properties at the start of the simulation (adding another 12 years of spin up).**

8) Line 113: what kind of grid are you using? Regular, structured/unstructured? Please explain

**We use a regular grid. This information will be added**

9) Line 119: Is Iron not important in your domain of study? (since I remember that iron is important in the global ocean).

**In case of dissolved iron in the water column as a limiting substance for the phytoplankton growth, the answer is no. Iron is not important in the domain of our study. However, for the**

**phosphorus cycling iron may be an important factor for the ability of the sediment to adsorb dissolved phosphate. One of the greatest challenges in implementing iron in this domain is the lack of observational data of iron in both the deep water and in the sediment as well as in the supply from atmosphere and land. Until enough observations are available, iron is assumed not to be a limiting factor.**

10) Section 2.1.2 I would include in the paragraph the treatment of light and PAR (that I see located in the SM). How is primary production parametrized? Did you use Q10 formulation? Please explain here.

**The full equations for primary production in NEMO-SCOBI are detailed in appendix A1 (equations A1 to A14) as written in lines 152-153. The specific equations where PAR is used is shown in equations A1 to A5. These equations shows that the modelled growth of phytoplankton is depending on temperature and can be limited by nutrients or light (i.e. we do not use Q10).**

11) Line 155-170: How tick is your sponge layer in the ocean domain? Why do not you use ORAS5 that is more recent than ORAS4? Did you use bulk formulas for latent and sensible heat fluxes? What formulation of albedo did you use in your ocean model (since it influences the quantity of shortwave radiation reflected by the surface)

**Because NEMO was originally intended for global setups with periodic boundary conditions, it has developed differently than other ocean model. Except for the nesting option called AGRIF, there is no sponge layer option in NEMO. Here, we have used the same settings as in Hordoir et al. (2019) for open boundary conditions (key _bdy). This means that we do not consider a sponge layer per say, instead we use a default thickness of one grid cell along the open boundaries. However, to make the model stable near the boundary, the viscosity/diffusivity coefficient has been significantly increased (by a factor of $\sim 10$) above the halocline near the boundary, as explained in detail by Hordoir et al. (2019).**

**The reason we use ORAS4 instead of ORAS5 is simply that we have not yet prepared ORAS5 OBCs, but we may do it in the future. We used the CORE bulk formulation as mentioned in Hordoir et al. (2019).**

**Regarding the albedo, we used the original formulation and constants for the ocean, ice and snow as given in LIM3, which is the ice model to which NEMO is coupled with (Vancoppenolle et al., 2008). The LIM3 has been broadly used for climate simulations and operational oceanography. Its code is open access and therefore detailed formulation can be seen here**
**In summary, the feedback of the albedo of ice promotes high absorption of shortwave radiation. The constants used could of course be tested and better tuned for the Baltic Sea and the North Sea, but this has not yet been done.**

12) Fig3-4 I would put the runoff as first panel in both figures.

**We prefer having the top panels with nutrients (TP and TN) as these are the most relevant for this work.**

13) Line 289: What do you mean with similar? please explain

**we will add more detail as follows:**

**"This is because the model response at these two stations is similar, both in magnitude and trend behaviour, to that at stations within their corresponding region ..."**

14) Figure 6 and after: please move the small panel Model Obs .. outside the first panel since it covers partially the lines beneath.

**We will improve this figure.**

15) Line 388 What do you mean with good period

**This will be rephrased as follows:**

**All these stations show good model skill for both the seasonal and the entire period evaluation for temperature and salinity (Fig. 10a and Fig. 10b).**

16) Paragraph 3.4. I think that this paragraph should go before the comparison between models and observations at the single stations. This could provide a general overview of the model performances better that the comparison with single point.

**We have taken this feed back into consideration, but came to the conclusion that we do not agree with the reviewer suggestion. The comparisons per station are given with time (as times series and interannual averages). This gives a better overview of when the model has large bias or not. In addition, figures for only 4 stations are shown, but the analysis is done for 27 stations (as written in line 210), which are regrouped by regions and therefore, also give a spatial overview. The analysis is then summarized with the model evaluation plots (Figs 10 and 11) in sections 3.2 and 3.3.**

17) Line 520-541, Line 551-560: These parts should go in the introduction since they provide an interesting description of the Baltic-North Sea system.

**We will add information on this in the introduction**

18) Figure 13-14: Put in first row the observations.

**This will be changed according to reviewer's suggestions**

**References**

Edman, M. K. and Anderson, L. G. (2014). Effect on pCO2 by phytoplankton uptake of dissolved organic nutrients in the Central and Northern Baltic Sea, a model study. *Journal of Marine Systems*, 139:166–182.

Gröger, M., Placke, M., Meier, H. E. M., Börgel, F., Brunnabend, S.-E., Dutheil, C., Gräwe, U., Hieronymus, M., Neumann, T., Radtke, H., Schimanke, S., Su, J., and Väli, G. (2022). The baltic sea model intercomparison project (bmip) – a platform for model development, evaluation, and uncertainty assessment. *Geoscientific Model Development*, 15(22):8613–8638.

Hordoir, R., Axell, L., Höglund, A., Dieterich, C., Fransner, F., Gröger, M., Liu, Y., Pemberton, P., Schimanke, S., Andersson, H., et al. (2019). Nemo-Nordic 1.0: a NEMO-based ocean model for the Baltic and North seas–research and operational applications. *Geoscientific Model Development*, 12(1):363–386.

van Leeuwen, S. M., Lenhart, H.-J., Prins, T. C., Blauw, A., Desmit, X., Fernand, L., Friedland, R., Kerimoglu, O., Lacroix, G., van der Linden, A., Lefebvre, A., van der Molen, J., Plus, M., Ruvalcaba Baroni, I., Silva, T., Stegert, C., Troost, T. A., and Vilmin, L. (2023). Deriving pre-eutrophic conditions from an ensemble model approach for the North-West European seas. *Frontiers in Marine Science*, 10.

Vancoppenolle, M., Fichefet, T., Goosse, H., Bouillon, S., Beatty, C. K., and Maqueda, M. M. (2008). Lim3, an advanced sea-ice model for climate simulation and operational oceanography. *Mercator Newsletter*, 28:16–21.

---

## Author Comment (AC3)

**Answer to reviewer 3: Review of the manuscript Validation of the coupled physical-biogeochemical ocean model NEMO- SCOBI for the North Sea-Baltic Sea system, by Ruvalcaba Baroni et al, submitted to Biogeosciences**

Note: Authors' answers are given in bold

**Reviewer #3**

**Manuscript overview**

This manuscript provides an extensive overview of the validation performed on a new coupled model set-up. This set-up existed for the hydrodynamic model (NEMO-Nordic), but not previously for the biogeochemical model (SCOBI). The validation therefore focusses mainly on the biogeochemical part. Model performance is not bad for such a model, and model extensions are also reported fully in the manuscript and appendix. The manuscript has no other focus than presenting the validation, and concludes this model set-up is not better or worse than others previously published by other research groups. As such, it is deemed to be a valuable addition to model ensemble studies.

**Review overview**

The manuscript is in general well written (some grammatical errors remain) and includes an exhaustive validation exercise. The authors focus a lot of attention on the fact that this set up includes both the wider North Sea and the Baltic Sea: most model set-ups do either one or the other due to the different governing mechanisms. For this set up the biogeochemical model was extended, from its natural domain the Baltic, to cover the North Sea and Channel areas. Therefore is seems strange that the validation is mainly focused on the Baltic and the Kattegat/Skagerrak area (for which SCOBI was designed) and hardly on the new areas it now has to represent. The authors truthfully cite a lack of observational evidence in the newly covered areas, but some station validation is surely possible. The biogeochemical model tends to capture the phosphorous and oxygen dynamics pretty well, but this is what I would expect from a model designed for and tuned to the Baltic Sea. The North Sea has very different dynamics, and although hypoxia and anoxia can occur there as well they are not a defining feature of the modern system. The authors themselves state that the model performs better in P-limited areas than N-limited areas, and the North Sea is mainly the latter. Modelled phytoplankton consists of diatoms, flagellates and cyanobacteria, but the latter hardly play a role in the salty North Sea and Channel: what groups could be added for a better representation of phytoplankton in the North Sea? Would addition of Phaeocystis spp. be an option (the North Sea nuisance species), and how good is the model at representing primary production at pycnocline depth? Figures 7 and 9 seem to indicate an overestimation of the top mixed layer depth, which should be discussed more. Overall, the model misses the seasonal dynamics of both systems (e.g. timing of spring bloom, autumn bloom), which could be due to the light climate, silica dynamics, phytoplankton parametrization or the nutrient inputs (temperature is usually easy to get right). Without additional analysis it is hard to say what is the main cause, particularly as this might differ per region. But some light attenuation validation could be added, as could a comparison with continuous Chla observations (few locations, usually short temporal coverage) or comparison with Chla satellite observations to get a better grip on this issue. I miss an in-depth analysis and discussions on these topics in the current manuscript. But the manuscript itself is worth publishing, as the model represents a valuable addition to both North Sea and Baltic modelling efforts.

**We thank reviewer #3 for the interest and the thorough review of the manuscript. The reviewer pin points already here important questions that we dissect and address one by one below:**

"Therefore is seems strange that the validation is mainly focused on the Baltic and the Kattegat/Skagerrak area (for which SCOBI was designed) and hardly on the new areas it now has to represent. The authors truthfully cite a lack of observational evidence in the newly covered areas, but some station validation is surely possible."

**0.1) The reviewer has a good point. However, it should be noted that monitoring programs are handled very differently in the North Sea than in the Baltic Sea. In the North Sea, single stations with long time series are greatly lacking or not publicly available, mainly confined to coastal areas and have less observations per year than in the Baltic Sea. The latter means that monthly or even seasonal profiles cannot always be obtained, and therefore, plots such as 7 or 9 are not that useful for the North Sea and statistically not representative. Many observations in the North Sea come from cruises limited in time that do not always cover the same exact transect from year to year. In addition, from our point of view, we showed that the evaluation per area is also very informative for detecting regional model biases. Following earlier studies (e.g. Pätsch et al., 2017) in the North Sea, we have prioritized the spatially compiled observations instead of isolated station data. Also note that both the Kattegat and the Skagerrak are new areas with respect to models that cover either the Baltic or the North Sea. In such models, both these areas constitute the boundaries of the models, which implies having simplified dynamics in these areas (this is written in lines 579-585 and also mentioned in our conclusions, line 659). We will add more discussion on this in section 3.5, to make it more clear that both Kattegat and Skagerrak are better represented in models covering both the North Sea and the Baltic Sea. In this respect, the North Sea, even though it can now be studied with NEMO-SCOBI, is mainly used here to better capture the complex dynamics of the Skagerrak-Kattegat transition zone. As a Swedish governmental institution, covering the entire Swedish coast including both the Skagerrak and Kattegat areas is important. Expanding the model to cover the North Sea allows for better dynamics in both these complex areas. This will be better emphasize in the text. Importantly, we have analyzed results from several stations in the Skagerrak and Kattegat areas (see figure 1 in the main text), therefore covering also both of these new areas. Please also see our reply to comment 17.**

The biogeochemical model tends to capture the phosphorous and oxygen dynamics pretty well, but this is what I would expect from a model designed for and tuned to the Baltic Sea. The North Sea has very different dynamics, and although hypoxia and anoxia can occur there as well they are not a defining feature of the modern system. The authors themselves state that the model performs better in P-limited areas than N-limited areas, and the North Sea is mainly the latter.

**0.2) Our analysis indicates that chlorophyll-a is best captured in areas where the same nutrient is limiting in both model and observations (lines 498-503). The reviewer could have been confused by line 417 (in section 3.2): "Because phosphate is better captured in the model, chlorophyll-a is also best captured at sites where phytoplankton is limited by phosphate ...". This phrase refers to stations in the Baltic proper (and not the North Sea). However, we agree that this is confusing and misleading. Line 417 will be rephrased as follows:**

"In the Baltic proper, good model skill for chlorophyll-a is found at 6 stations (i.e. BY4, BSCIII-10, HANOBUKTEN, BY10, BY15 and BY20), which also show good or acceptable model skill for both nitrate and phosphate".

**0.3) It is actually the other way around. It is surprising that by expanding the model to a such different dynamic region, and with very little additional tuning, most processes in the North Sea are still well captured. This point will be better highlighted in the text. It is only specific coastal areas (mentioned in line 457) which are not well represented in NEMO-SCOBI and in other models as well. Such areas are affected by multiple uncertain factors (e.g., riverine input, non-Redfield ratios for phytoplankton growth, light penetration depth, etc). The fact that phosphorus is better captured in the model in both the North Sea and the Baltic Sea could be indeed linked to a better model representation/parametrization of the phosphorus cycle than that for the nitrogen cycle. It can also be related to a better phosphorus forcing than that for nitrogen. It is true that the oxygen and phosphorus cycle in the Baltic Sea are key study points to understand this system, but this knowledge also applies to the North Sea.**

Modelled phytoplankton consists of diatoms, flagellates and cyanobacteria, but the latter hardly play a role in the salty North Sea and Channel: what groups could be added for a better representation of phytoplankton in the North Sea? Would addition of Phaeocystis spp. be an option (the North Sea nuisance species)

**0.4) Yes, the cyanobacteria species included in SCOBI do not grow in the North Sea, but they do grow in other areas of the domain. As in other biochemical models covering a similar model domain (e.g., ERGOM Maar et al. (2011)), we consider diatoms and flagellates to be the dominant species in the North Sea and do not plan to add additional species for the North Sea in this version of the model. Note that the category PHY2 in SCOBI accounts for "flagellates and others" (line 126) and therefore implicitly accounts for other species such as Phaeocystis spp. However, explicitly including a phytoplankton group that reacts on changes in the N to P ratios, as seen in Phaeocystis spp in Dutch coastal waters Riegman et al. (1992), could indeed improve results in the southern coast of the North Sea. The results presented here are results from the first version of NEMO-SCOBI, which is under constant development. The reviewer suggestion will be certainly considered for future improvements.**

how good is the model at representing primary production at pycnocline depth? Figures 7 and 9 seem to indicate an overestimation of the top mixed layer depth, which should be discussed more.

**0.5) The chlorophyll-a values at the pycnocline are in very good agreement with observations (both seen in the Hovmöller diagrams and the seasonal and period averaged profiles of Figures 7, 9, B1, B2 to panels). The overestimation above the pycnocline seen in the summer months in the Hovmöller diagrams comes from the delay in the timing of the phytoplankton bloom. The maximum chlorophyll-a concentrations is generally well captured by the model at all stations, but comes too late in the year at several station in the Skagerrak-Kattegat and the Baltic proper. This is mentioned in section 3, more specifically in line 335 for the Skagerrak-Kattegat (AN-HOLT) and in line 516 for the Baltic proper (BY15). Station-specific differences in chlorophyll-a are also explained in detail for ANHOLT and BY15 and discussed in lines 367-377. However, we could add a line summarizing the fact that the overestimation above the pycnocline for the summer months comes from the bloom delay in lines 570-575.**

Overall, the model misses the seasonal dynamics of both systems (e.g. timing of spring bloom, autumn bloom), which could be due to the light climate, silica dynamics, phytoplankton parametrization or the nutrient inputs (temperature is usually easy to get right). Without additional analysis it is hard to say what is the main cause, particularly as this might differ per region. But some light attenuation validation could be added, as could a comparison with continuous Chla observations (few locations, usually short temporal coverage) or comparison with Chla satellite observations to get a better grip on this issue. I miss an in-depth analysis and discussions on these topics in the current manuscript. But the manuscript itself is worth publishing, as the model represents a valuable addition to both North Sea and Baltic modelling efforts.

**0.6) To capture all aspects of the seasonal cycle is challenging. We have, however, shown that the seasonal cycle for all assessed parameters are well captured at least in several regions of the model (e.g. all dots falling within the inner and outer quarter circle in Figure 10 and written in lines 418-420). In the text, we do highlight the specific parameters that are not well captured for specific months and areas in order to understand what to improve in the model. We also acknowledge that the reviewer consider the manuscript worth publishing.**

**0.7) Regarding the phytoplankton bloom delay, it is mainly occurring at some stations in the Skagerrak-Kattegat transition zone and the Baltic proper, as highlighted in lines 421-424. To our knowledge, no current biogeochemical model of the Baltic Sea and/or the North Sea have good representation of chlorophyll-a everywhere in their domain. Much of the modelling community work is focusing in improving this, but it is not trivial work. NEMO-SCOBI is not an exception and therefore we are currently working, in a parallel study, in improving chlorophyll-a concentrations, where we indeed aim to include high temporal resolution data and satellite data sets. For the present study, this is not feasible, as it requires much more additional information that would make the paper indigestible. For example, satellite data should come with additional analysis on its own, especially for chlorophyll-a, and additional evaluation periods as well as additional stations would be required. Importantly, in this paper we focus on long-temporal trends. In addition, the model has been partially validated against chlorophyll-a satellite data in the North Sea in a recent study (van Leeuwen et al. (2023)). We will add the latter information.**

**0.8) Regarding the light attenuation validation, the reviewer is correct and this is indeed an important parameter for phytoplankton growth. In fact, as mentioned in line 562, we have analyzed the light attenuation in the model and it is overestimated in the Baltic proper and the Skagerrak-Kattegat transition zone. The figure for observations and model results averaged over 1975 to 2017 for winter and spring are shown below, as an example. We will specifically add a line summarizing this information and specify that the model overestimation of the light attenuation in the Skagerrak-Kattegat is likely linked to the delay of phytoplankton bloom in lines 574-577. Please see also our reply to reviewer #1.**

[Figure]

(a) Modelled secchi depth in surface waters for winter months averaged over 1975 to 2017.

(b) ICES secchi depth in surface waters for winter months averaged over 1975 to 2017. Note that the scale is the same for both model and observations.

[Figure]

(a) Modelled secchi depth in surface waters for spring months averaged over 1975 to 2017.

(b) ICES secchi depth in surface waters for spring months averaged over 1975 to 2017. Note that the scale is the same for both model and observations.

**Recommendation**

Moderate revision

I like the manuscript but would like to see further validation results added to it for 1. the North Sea area and/or Channel area (main) and 2. the riverine forcing used (appendix). This would require no new simulations but new post-processing. I would also like to see figures 6 and 8 reorganized and figure B3 moved from the appendix to the main article. If necessary, figure 5 could be banned to the appendix to make way for figure B3.

**0.9) In our reply to the reviewer's first comment (reply 0.1), we have detailed the reasons why we have chosen not to validate model results for the North Sea at single stations. Please note, however, that we are aware of higher temporal resolution observational data for recent years and will be included in future work. We would like to pin point again that validation for stations in both the Kattegat and Skagerrak (new areas for this model) have been done and shown (Figures 1,6,7,10 and B1).**

**0.10) Regarding figure B3, we are not sure that moving figure B3 to the main text will be better, as it shows similar (almost repeating) information as figure 11, due to the fact that observations are mainly confined to surface waters. However, we will consider this within the author team (please also see our more detail answer to comment 23, page 11). Regarding figures 6 and 8, please see our reply to comment 19.**

**0.11) We would prefer to not ban figure 5, as it clearly shows the lack of observations in the North Sea with respect to the Baltic Sea. This means that extra care must be taken when analysing trends in the North Sea, that some analysis cannot be performed and that analysis for the North Sea will not have the same statistical significance as those for the Baltic Sea. This is one of the main reasons we did not show observations for the North Sea at single stations. Regarding the river forcing, please see our more detailed answer below (comments 12 and 13).**

**Detailed Comments**

1) Line 14-16: The validation is in agreement with assessment areas ... ? Do you mean that you are using assessment areas for the validation (i.e. method), or that validation within these assessment areas confirms with reported values in those same areas (i.e. validation result)?

**This will be rephrased as follows:**

**Hence, these observations represent only a snapshot of nature and there are no guarantees that the measurements actually captured the chlorophyll peaks and may ...**

2) Line 19: the references are not in alphabetical or chronological order.

**This will be corrected to be in chronological order**

3) Line 44-45: too many commas and bad grammar. Not sure what is meant here: such areas refers to the deeper parts of the North Sea (previous line), but those are not coastal.

**This will be rephrased**

4) Line 50: rereferred to as cyanobacteria and I miss a reference for the statement that cyanobacteria do not grow in the North Sea.

**This will be rephrased (also in response to reviewer #1, comment 1), but the overall message is that the filamentous (and sometimes toxic) cyanobacteria that we parameterize in the model in the Baltic Sea do not grow in the North Sea, because the latter is too salty for this species. We will add the following reference:**

**Olofsson, M., Suikkanen, S., Kobos, J., Wasmund, N., Karlson, B., 2020. Basin-specific changes in filamentous cyanobacteria community composition across four decades in the Baltic Sea. Harmful Algae 91, 101685. https://doi.org/10.1016/j.hal.2019.101685**

5) Line 78: bad grammar, I suggest , which is particularly true in"

**This will be rephrased according to reviewer's suggestion**

6) Line 80: but contain biases for

**This will be corrected**

7) Line 87-90: Not sure why this text is here, not relevant to the subject of this manuscript.

**We agree with the reviewer that these lines are not relevant in this section. We will therefore move the sentence to the section "relevance of this study" (section 3.5).**

8) Fig 1: I would say observational SHARK stations. The acronym is later used with capitals, without those it is rather confusing here. Indeed, SHARK is only explained in line 205, so some explanation is due here.

**Yes, this is a typo and shark should be capitalized (also in response to reviewer 1).**

9) Line 131: please refer to the figure before the textual explanation, to make it easier on the reader.

**We agree with the reviewer that it would be better and will therefore move line 131 to line 121.**

10) Line 148: do you mean that the phytoplankton parameters were tuned to represent both the Baltic and North Sea areas? That is to say, the parametrization the model had previously was tuned to the Baltic and these parameters did not fit with North Sea simulations and so needed adjustment? If so, can you speculate why this was necessary? What processes/groups/functionality difference is there between these areas that make this adjustment necessary?

**The explanation for these adjustments are given in appendix A1 (lines 674-678):**

**"In the SCOBI version coupled to NEMO, rates and dependencies for phytoplankton growth were modified with respect to previous SCOBI versions in order to account for silica limitation of diatoms (not included in earlier versions), to improve the occurrence of dominant groups in both the North Sea and the Baltic Sea and to limit cyanobacteria growth in the Skagerrak-Kattegat transition zone and in the North Sea."**

**For more clarity, this information will be moved to the main text, instead.**

11) Fig 2 : this is a spaghetti diagram, hard to read for the many arrows. I think it is a bad idea to include so much detail that the model visual abstract (which is what this is) becomes visually unattractive. I would leave out the coloured arrows explanation, readers can see for them selves if a flux stays in the pelagic or not. Or use different line styles. Maybe group it a bit more, with all pelagic nutrients together in a circle and 1 arrow going in and out if all nutrients are needed? And why are all the fluxes named in the caption rather than in a separate table?

**This is a matter of taste, but we agree that to meet the color blind test, the usage of colors for the different arrows are not the best choice. We will try different line styles instead. Gathering nutrients in one circle would simplify too much the figure and hide relevant information as all nutrients do not have the exact same pathway or dependency. Regarding the fluxes, we could add a table with all fluxes but we do not think it will be relevant for the main text as it would be too much detail. It can then be added in appendix, but the fluxes still would need to be mentioned in the main text in relation with the figure. If the reviewer really wants the table, we can add it, but we think our choice to mention them in the caption, even if we agree that it is not perfect, it is still the best choice in this case.**

12) Line 167: please provide a reference for the applied reduction factor, assuming this is a generally available dataset. If it is not I dont quite see why the authors would use this particular product.

**The runoff data used here (as mentioned in line 165) comes from a dedicated run performed with the EHYPE model, thus especially done for this study. The EHYPE model was forced with an atmospheric database called Hydro-GFD (Berg et al., 2020), which overestimated precipitation by about 10 percent, according to a validation done by the meteorological group at our institute. Hence the river discharge correction factor of 0.9. This will be better clarified in the text. Therefore, there is no published reference for this particular EHYPE run, but it has been tested against standard check ups within the EHYPE team before delivering it for this work. It also accounts for more realistic number of river outlets than those in observed runoff data sets., which will also be emphasized in the text. In addition, the EHYPE model Donnelly et al. (2009), is a hydrological model that has been well validated and broadly used in many different settings (e.g. Donnelly et al., 2016; Hundecha et al., 2016; Nijzink et al., 2018; Macian-Sorribes et al., 2020. In fact, together with LISFLOOD and VIC, EHYPE is one of the most popular models applied for large-regional scale Piniewski et al. (2022)).**

13) I am getting a bit confused about the riverine forcing applied. Am I correct in thinking that you used

- discharge values calculated by a hydrological model, which were adjusted evenly across the domain for a known, uneven model error?

**Yes. A constant reduction factor was applied over the entire model domain, as mentioned in our reply to comment 12.**

- nutrient values based on the same hydrological model, but adjusted for each year and basin to observational values based on two different observational data sets?

**Yes, one data set is used for the Baltic Sea Gustafsson et al. (2012) and the other for the North Sea Lenhart et al. (2010). Note that the ICG-EMO data, even if it contains data for the Baltic Sea, it neglects important river outlets in this sea.**

If so, then I think it unlikely that any modeller could replicate your efforts, as they cannot replicate this forcing set. And it makes me wonder why the observational sets were not used directly. This mix up of 3 different sources complicates interpretation of results, which are reported in eutrophication-relevant variables. Please provide more detail on this forcing set (in an appendix), as well as a comparison for a few selected rivers (e.g. some of the larger dots in figure 3) of the applied discharge and nutrient loads compared to the observations that you state you also have. Can some of your mismatches in coastal zones be related to this forcing data?

**This is indeed a large part of the work we did. It remains, however, a side part for this manuscript. There are many possible ways to compile river forcing and we selected the method that we think best fits our purpose.**

**The EHYPE data set covers all relevant rivers and models physically consistent and well calibrated changes in river input according to meteorological forcing. Therefore, as written in lines 188-190, captures well the interannual variability (both for runoff and nutrients). However, it does not yet account for the anthropogenic effect on nutrients (detailed in lines 189-191). The latter is extremely relevant for our study or else our model results for nutrients will be meaningless. Importantly, the ICG-EMO and the Baltic Sea riverine data sets are not directly applied in this study because the EHYPE data includes much more river outlets than in both of these data sets combined (636 *vs* 208 outlet points, respectively). Another relevant point, is that the ICG-EMO and the Baltic Sea riverine data sets are not pure observations per say. They have been derived from observations, meaning that several assumption had to be applied to those data sets. Real observations for nutrients (for example those reported per country) lack not only spacial resolution, but also temporal resolution (mostly available per year). Also, the Baltic Sea data (Gustafsson et al., 2012) provides monthly loads per basin, which need to be redistributed in the different rivers of the corresponding basin in the Baltic Sea. Thus, both the ICG-EMO and the Baltic Sea data sets come with large uncertainties. We will add information on this in the main text.**

**Regarding the effect of the river forcing on our model results, the nutrient trends in the model closely follow those seen in the riverine data set in many regions of the model domain, especially in the Skagerrak-Kattegat area and coastal areas. Thus, improving the riverine data set will certainly lead to improved model results. In fact, improving the riverine data sets in both the Baltic Sea and the North Sea is an active field of research. For example, the ICG-EMO data set is constantly being updated and discussed within a large group of experts from which we are part of (see acknowledgements). More information of the importance of the river forcing will be added in the main text. Note, however, that we will not necessarily show results for one or few specific rivers, as this implies validating riverine data sets which is out of the scope of this manuscript.**

**Currently, the riverine data applied in this study is available on demand but we will look for a place to make it freely downloadable, transparent and citable. Note that, due to its high spatio-temporal resolution, our riverine data set is quite heavy, even if provided in netCDF files and it**

**may not be possible to store it in an open platform. If this is the case, we will add that the dataset is available on demand and provided with a full description.**

**The reviewer has a good point regarding the re-productivity of our riverine data set and, as also mentioned in comment 12, we will look for a platform that allows for large data sets to be stored, explained, freely downloadable and citable. Note that due to high spatio-temporal resolution this data set is quite heavy, even if provided as netCDF files. If storage in an open platform is not possible, we will add that the dataset is available on demand and provided with a full description.**

14) Line 233: the reference year was chosen because of high nutrient values. Where those simulated values or observational values?

**The period was chosen when nutrients where high in both the model and the observations. This will be clarified in the text.**

15) Line 245: explanation of the applied seasonal delineation (meteorological? astronomical?) is only given in the caption of figure 10. Please provide this here.

**We will add this information in line 245 as well.**

16) Line 280: this has been stated before.

**We understand that the phrasing is similar, however, line 255 refers to the fact that we do not show areas with less than 100 observations in figure 5. In line 280, we refer to the method for the evaluation of the model, where we do not evaluate such areas. Thus, the information, even if similar, do not refer to the exact same thing.**

17) Section 3.1: the authors should include a North Sea station here, maybe one on the Danish or Dutch transects or an individual research station from the UK. It may not have everything the authors want but an extension of the SCOBI model into the North Sea and Channel areas should be validated in detail there. Stations like the Oystergrounds (NL), West Gabbard (UK) or L4 (UK) spring to mind, though the latter is I think just outside of the domain. These may have standard surface monitoring and limited at depth monitoring, but it is better than nothing. They also have high resolution observational data for a few years, generally. In the very least a North Sea station comparison will provide more detail on the local Chl-a seasonal signal and bloom timing capacity of the model there (difficult to derive from figure 12).

**We thank the reviewer for providing recommendations of further stations. In our study we used the ICES data, which should include all publicly available data for the North Sea. Figure 12 together with figure B4 already show that the general spatial pattern of the phytoplankton bloom in the North Sea is actually well captured, but that the model mainly underestimates it in the southern coast of the North Sea in winter, spring and autumn, while in summer it overestimates it. They also give information of the timing of the bloom and show that chlorophyll-a values in the model do increase when observation values also increase in most areas in the North Sea, except at very coastal areas during winter and autumn. This indicates that the timing is generally well captured in the North Sea, except for these very coastal areas (so a very small part of the domain) where the blooming lasts longer in observations than in the model. We therefore argue that single stations in the North Sea will not provide much more new information in terms of long-term**

**trends and general patterns. However, we will further look into specific stations, keeping in mind that this is an ocean model and not a coastal model, and add text if new relevant information is found.**

**Regarding the high resolution chlorophyll-a, we are aware that these exist for recent years (this is also the case for the Baltic Sea), but have chosen not to include them in this particular study as they do not cover the complete studied period (2001 to 2017). Again, the focus of this work is to validate the model performance on long-temporal trends and general biogeochemical patterns, which we have explained in detail in the text. We agree with the reviewer that high resolution data is extremely valuable and relevant for model improvements and we are currently including such observations (as well as satellite data) for further analysis. However, we think it is too much information to add in this study, as such observations should come with additional detail description. Please also see our reply above (page 4, reply 0.7).**

18) Line 315-317: can you provide an overview of the trends in table form in the appendix? Now it is hard to see and compare trends.

**We will summarize the trends shown in figures 6 and 8 in a readable table and add it where relevant.**

19) Figures 6 and 8: I would suggest restructuring these. A label over results in a graph is a no-go, in any case. Suggestion: make a two column graph (which these are already). Top left: the legend for surface values. Rest of left: surface graphs for T, S, NO3, PO4, Chl-a. Top right: legend for bottom values. Rest of right: bottom graphs for T, S, NO3, PO4, O2. The legend for O2 can be removed and explained in the caption. This would make the graph more accessible as surface or bottom processes can be viewed at a glance (vertically) while top and bottom values can still be compared easily (horizontally).

**The figures will be improved (also in reply to reviewer # 2, comment 14).**

20) Line 330-333: no guarantees that the measurements did not fail to, the double negative here makes this sentence hard to read. I presume your point is that observational evidence is discrete in time and so can easily miss the peak of the spring bloom. This is a very valid and important point to make, which merits unambiguous text.

**The sentence will be rephrased as mentioned in our reply to comment 1:**

**"no guarantees that the measurements actually captured the chlorophyll peaks..."**

21) Line 350-354: the model correctly predicts inflow of North Sea waters into the Baltic proper, though bottom temperature and salinity values are too low compared to observations. But this is a feature of the existing hydrodynamics model, NEMO-Nordic, which was already used in the presented domain before, and calibrated and validated there. I would not expect the extension of the SCOBI model to influence these dynamics.

**The reviewer is correct: the code for the dynamics in NEMO-Nordic has, indeed not been changed and the SCOBI model does not affect the hydrology of NEMO-Nordic. However, the applied physical forcing has changed, especially that for runoff. As mentioned in lines 378, the**

new river forcing significantly improved the surface salinity results in the Baltic Sea when compared to results in Hordoir et al. (2019), but increased the existing negative bias in intermediate and deep waters of the Baltic proper. The Baltic Sea in NEMO-Nordic is, fairly sensitive to changes in fresh water input and, while we are currently working on improving this bias, here we have chosen to compromise the known bias in deep waters in the Baltic proper for improved surface results.

22) Figures 7 and 9: the little cyan plusses (not crosses as it says in the caption, that would be x) are very hard to see. Can this be done by shading instead? I do love the surface values on top of the depth graphs, very nicely done!

**Thanks! We have already tried the shading and it is not great as it hides the standard deviation for the model (or make it less visible). However, we will further look on how to improve the averaged profile figures.**

23) Line 435: figure B3 is mentioned here, I would prefer to see figure B3 in the main text rather than in the appendix. If there is a limited number of figures allowed, I would suggest moving figure 5 to the appendix instead, as it does not show simulated results. Within B3 the markers are very hard to see, can you make then larger? Some of the colours are quite light, resulting in a number without a visible marker in my printed version: enlargement might help with this too.

**We will indeed enlarge the markers of figure B3. Regarding it's position in the manuscript, we had the same discussion within co-authors before submission and decided that it was too detailed information for the main text. If adding it to the main text, a detailed discussion on it should be added as well, describing main differences between areas, parameters and position in the water column. This will significantly complicated the flow and readability of the manuscript. The main relevance of this figure is to show where the model performs good or not. For example, if wanting to use the model results for a special variable in a particular area, one can look at this figure and have an idea if such results can be trusted or not. It also helps the developers of NEMO-SCOBI to see if changes in the code or forcing improves both these model performance figures (namely figure 10 and B3). Importantly, the figure results for the North Sea below surface waters are uncertain due to the great lack of observations (as clearly seen in figure 5). Moreover, we have summarized the relevant patterns in section 3.4 and discussed the main processes that could have an effect on the seen model biases in section 3.6, which is the main focus of this manuscript. It is also somewhat repetitive information as many of the area points show similar positions than those of Figure 10, especially when looking at surface values.**

24) Line 463: in the Baltic Sea, four HELCOM-OSPAR assessment areas. Surely these are HELCOM assessment areas?

**We agree that this is confusing, this will be changed.**

25) Line 477-4780: surely you can see in your simulation results if accumulation happens or not?

**Yes. We do see an accumulation over time in the time series at stations in the Gulf of Bothnia when compared to observations. However, there are only a few data points for nitrate at for example F3 and F9 in the entire time series and even if they are clearly below model results in recent years, it is difficult to say much more than this.**

26) Line 483-485: this is an important message for the monitoring organisations, please make it stronger.

**It is indeed a relevant point and we will emphasize it more in the text.**

27) Line 492-493: grammatically incorrect sentence and it doesnt make much sense.

**We agree with the reviewer that this line is confusing. "The Northern North Sea" is the official name for the central area in the OSPAR assessment areas. We refer to "the HELCOM-OSPAR assessment areas" to the areas adapted to our model domain, which means that they are not exactly the same, notably for OSPAR. The OSPAR assessment areas include also the North Atlantic and therefore some of the areas around the boundaries are cut in our domain. What we mean in this sentence is that the central part of the North Sea is the largest assessment area in figure 1 but also the least measured, which makes the model skill evaluation unreliable. We will rephrase this sentence.**

28) Line 494-495: surely this is not about which model is better? Grammatically also incorrect, I assume the model by Daewel et al captures the southern coast of the North Sea just fine. Maybe not in biogeochemical terms, but the coastline itself is in the model so it captures it.

**Correct, it is certainly not a competition between models. We are only highlighting the areas where most models fail to capture the nutrient dynamics. This will be rephrased as follows:**

**"Nonetheless, our biogeochemical results are comparable to those of Daewel and Schrum (2013): an overall good agreement with observations in the North Sea but with biases in the southern coast of the North Sea."**

29) Line 505: sentence is too long and loses it grammatical structure by the end. Please rephrase.

**We will split the sentence in two.**

30) Line 513-516: please speculate on what the missing process for phytoplankton growth could be. And add riverine nutrient validation to the appendix (e.g. figure 3 but with an applied forcing-observational evidence focus), to better quantify the nutrient input issue. How well does your input capture suspended matter from fluvial sources?

**The missing processes we refer to in this sentence are later on explained in section 3.5. For clarity, we will add at the end of the sentence "(further discussed in section 3.5)". Regarding suspended matter from riverine input, again, here we do not validate the riverine data set. In addition, we have modified EHYPE nutrient inputs to match those of observations and therefore we do not really understand the question. This said, we do mention (in lines 614) that there are large uncertainties regarding organic matter (detritus) in both seas. This also concerns all riverine data sets, as suspended and particulate organic matter is not necessarily automatically measured in all countries.**

31) Line 524: have you considered the following works?

Capuzzo, E., Stephens, D., Silva, T., Barry, J., & Forster, R. M. (2015). Decrease in water clarity of the southern and central North Sea during the 20th century. Global change biology, 21(6), 2206-2214.

Capuzzo, E., Painting, S. J., Forster, R. M., Greenwood, N., Stephens, D. T., & Mikkelsen, O. A. (2013). Variability in the sub-surface light climate at ecohydrodynamically distinct sites in the North Sea. Biogeochemistry, 113, 85-103.

And how does this work relate to your findings?

**We thank the reviewer for providing additional interesting references. We will look into both references and add findings if relevant for this study.**

32) Line 526: you mean the Rhine, arguably the largest river to exit into the North Sea, has no influence here? Surely not.

**Thank you for catching this. It is a typo and the Rhine will be added.**

33) Line 532-535: figure 12 shows no observational support for this. How do you know your model is not simply overestimating the local light climate?

**Lines 532-535 do not refer to our results (no reference to Fig. 12 there), but discuss findings in the literature. The sentence will be rephrased as follows:**

**"The elevated chlorophyll-a concentrations along the eastern UK coast has been shown to be likely related to frequent upwelling events under a predominant westerly wind regime ...**

**Please note that we do mention in line 562 that our model does overestimate the light attenuation depth. See also our reply 0.8 on light limitation and attached figures.**

34) Line 542-544: maybe, but a comparison with satellite observations could verify this point better spatially.

**Please sea our reapply in page 4, comment 0.7.**

35) Line 547: in figure B4 the matching points are hard to see as they are white, overemphasizing the discrepancies. Can you use a blue-yellow-red colourbar here to emphasize where model and observational evidence do agree, and where there is simply an observational dessert? The same applied to figure 13, where observational points with a N:P ratio of (near) Redfield values are invisible.

**We will see how to improve the visibility of matching points. The matching points are now in gray (not white), so we agree that they are difficult to distinguish not from missing data but between an island/land point from an actual observation point.**

36) Line 570-575: spring bloom timing is mainly driven by temperature and light conditions in the North Sea, so a discussion on the simulated light climate is due here. Diatoms have evolved to be more light receptive than most other phytoplankton species, so they lead the spring bloom. Does the biogeochemical model allow for a proper succession of species? Figure B5 indicates it does, but a general seasonal succession graph (daily resolution, maybe for the different basins) would be better to display the models inner workings. A difference of 3 months in spring bloom timing is a lot, even for a large scale biogeochemical model.

**Yes, the growing succession of species have been plotted and analyzed as these were tuned for the North Sea. It is not the main issue causing the bloom delay in the Skagerrak-Kattegat. In the text, we have highlighted the main factors that are likely the cause: the light limitation and the fact that its seasonality is not well captured in the Skagerrak-Kattegat area. We will make this point more clear in the text (in lines 562).**

37) Figure 12: I love this graph but at the current size results are hard to compare to observational evidence. Can these graphs be enlarged? The colourbars are also hard to read.

**Thanks! The figure is full page (A4), so enlarging it may be difficult. This may depend on how the journal handles it in the final version. However, we can enlarge the fonts.**

38) Line 580: allows for a study of the North Sea

**This will be changed according to reviewer's suggestion.**

39) Line 584: , rather than prescribed boundary conditions

**This will be changed according to reviewer's suggestion.**

40) Line 585: again, this should not be a model contest on who performs best.

**We fully agree, but still comparison is required to understand model discrepancies. We leave the sentence as it is also in response to reviewer #2, who is keen on knowing the performance of this model in comparison with others.**

41) Line 595-597: you have shown that your model is capable of simulations from which you can derive relevant indicators for HELCOM and OSPAR, taking into account model performance and bias. Certainly for Chl-a there would be caveats, but most models have these. But you have not shown that the model can be used for climate projections with specific relevant improvements, as you have not made these improvements yet. And there is no detailed information in the manuscript on what these improvements would be: several ideas have been floated but there was no priority list of things to implement in the model. I would remove the latter part of this statement. For example, line 650 list improving the seasonal cycle of benthic denitrification, but contains no statement to how important this is with regard to other suggested improvement (e.g. cyanobacteria life cycle inclusion), or how this will be achieved.

**A priority list is difficult to provide, because all mentioned future changes have a relevant role and can lead to significant improvements. However, solving the timing of the chlorophyll-a bloom is the first thing we will consider for further improvements. Therefore, it is the first point mentioned in the discussions of the section 'Future plan and knowledge gaps'. For this to be clear, we will rephrase the first sentence of this section as follows:**

**"Solving the phytoplankton bloom timing in the southeastern coastal North Sea and the Skagerrak-Kattegat transition zone in NEMO-SCOBI is a priority as it would significantly improve the model results, ..."**

**Note that this work necessarily implies improving the light penetration depth (as mentioned in line 612). In addition, we have shown that the main spatio-temporal patterns are well captured**

and with the evaluation made here, we have a very good idea of where results can be trusted and where not. Of course, we do want to improve our model results and further understand them. However, this is not a major problem for the future projections, because that work will not focus on seasonal patterns, but on yearly averages (or even period averages). We have shown that these are in good agreement with observations and therefore, the model can be used as it is. For climate projection the aim is to keep the model as simple as possible, but ensuring that the main processes are captured. This is to be able to perform long-term runs at a reasonable computational cost.

42) Line 596-597: this is why I want to see a validation of the applied riverine forcing. The atmospheric deposition bias was discussed, but the reader lacks information on the riverine input bias.

**Again, the focus of the paper is not on river validation. However, we will add more discussion on the river's effect. Please see our detailed reply to comment 12 and 13.**

43) Line 613: how about suspended sediment?

**The model does not take into account suspended inorganic sediments. However, it accounts for resuspended material from the sediments and it is taken into account for the light penetration depth as they go back to the corresponding nutrient pool in the water column (see for example N1 and P5 fluxes in figure 1).**

44) Line 633: please provide references for the claim that the model compares well with previously published estimates (assuming you mean other publications than Dalsgaard et al, 2013).

**In line 633, we indeed compare the model values to the values mentioned in Dalsgaard et al, 2013, which are mentioned just above (line 631). We agree with the reviewer that this is confusing and will rephrase these lines as follows:**

**line 630 "The total nitrogen removal from water column denitrification in the Baltic proper with persistent large hypoxic areas has been estimated to be..."**

**and**

**line 633 "This compares well with previous published estimates" will be removed from the text.**

**References**

Donnelly, C., Andersson, J., and Arheimer, B. (2016). Using flow signatures and catchment similarities to evaluate a multi-basin model (E-HYPE) across Europe. *Hydr. Sciences Journal*, 61(2):255–273.

Donnelly, C., Dahne, J., Lindström, G., Rosberg, J., Strömqvist, J., Pers, C., Yang, W., Arheimer, B., et al. (2009). An evaluation of multi-basin hydrological modelling for predictions in ungauged basins. *IAHS publication*, 333:112.

Gustafsson, B. G., Schenk, F., Blenckner, T., Eilola, K., Meier, H., Müller-Karulis, B., Neumann, T., Ruoho-Airola, T., Savchuk, O. P., and Zorita, E. (2012). Reconstructing the development of Baltic Sea eutrophication 1850–2006. *Ambio*, 41(6):534–548.

Hordoir, R., Axell, L., Höglund, A., Dieterich, C., Fransner, F., Gröger, M., Liu, Y., Pemberton, P., Schimanke, S., Andersson, H., et al. (2019). Nemo-Nordic 1.0: a NEMO-based ocean model for the Baltic and North seas–research and operational applications. *Geoscientific Model Development*, 12(1):363–386.

Hundecha, Y., Arheimer, B., Donnelly, C., and Pechlivanidis, I. (2016). A regional parameter estimation scheme for a pan-European multi-basin model. *Journal of Hydrology: Regional Studies*, 6:90–111.

Lenhart, H.-J., Mills, D. K., Baretta-Bekker, H., van Leeuwen, S. M., van der Molen, J., Baretta, J. W., Blaas, M., Desmit, X., Khn, W., Lacroix, G., Los, H. J., Mnesguen, A., Neves, R., Proctor, R., Ruardij, P., Skogen, M. D., Vanhoutte-Brunier, A., Villars, M. T., and Wakelin, S. L. (2010). Predicting the consequences of nutrient reduction on the eutrophication status of the North Sea. *Journal of Marine Systems*, 81(1):148–170.

Maar, M., Möller, E. F., Larsen, J., Madsen, K. S., Wan, Z., She, J., Jonasson, L., and Neumann, T. (2011). Ecosystem modelling across a salinity gradient from the North Sea to the Baltic Sea. *Ecological Modelling*, 222(10):1696–1711.

Macian-Sorribes, H., Pechlivanidis, I., Crochemore, L., and Pulido-Velazquez, M. (2020). Fuzzy postprocessing to advance the quality of continental seasonal hydrological forecasts for river basin management. *Journal of Hydrometeorology*, 21(10):2375–2389.

Nijzink, R., Almeida, S., Pechlivanidis, I., Capell, R., Gustafssons, D., Arheimer, B., Parajka, J., Freer, J., Han, D., Wagener, T., et al. (2018). Constraining conceptual hydrological models with multiple information sources. *Water Resources Research*, 54(10):8332–8362.

Pätsch, J., Burchard, H., Dieterich, C., Gräwe, U., Gröger, M., Mathis, M., Kapitza, H., Bersch, M., Moll, A., Pohlmann, T., et al. (2017). An evaluation of the North Sea circulation in global and regional models relevant for ecosystem simulations. *Ocean Modelling*, 116:70–95.

Piniewski, M., Eini, M. R., Chattopadhyay, S., Okruszko, T., and Kundzewicz, Z. W. (2022). Is there a coherence in observed and projected changes in riverine low flow indices across Central Europe? *Earth-Science Reviews*, page 104187.

Riegman, R., Noordeloos, A. A., and Cadée, G. C. (1992). Phaeocystis blooms and eutrophication of the continental coastal zones of the North Sea. *Marine Biology*, 112:479–484.

van Leeuwen, S. M., Lenhart, H.-J., Prins, T. C., Blauw, A., Desmit, X., Fernand, L., Friedland, R., Kerimoglu, O., Lacroix, G., van der Linden, A., Lefebvre, A., van der Molen, J., Plus, M., Ruvalcaba Baroni, I., Silva, T., Stegert, C., Troost, T. A., and Vilmin, L. (2023). Deriving pre-eutrophic conditions from an ensemble model approach for the North-West European seas. *Frontiers in Marine Science*, 10.

---

## Author Response (AR1)

**Changes done on the revised manuscript Validation of the coupled physical-biogeochemical ocean model NEMO-SCOBI for the North Sea-Baltic Sea system by Ruvalcaba Baroni et al.**

**1 Summary of main changes in reply to reviewers**

In our revised manuscript, we have addressed point-by-point all 3 reviewer's comments. Point-by-point modifications are detailed in section 3. The changes mainly consist of rephrasing, adding more detailed information on, for example, the riverine forcing, and showing additional figures. We believe these changes have significantly improved the manuscript. More specifically, we have:

1) in reply to reviewer 1, added a figure in Appendix A, Fig. A1, showing the sensitivity of phytoplankton growth to temperature and light. For this, the full light attenuation parameterization was also added in Appendix A, together with the definition of corresponding symbols (lines 774-778, equation A6). The values for each constant were also added to Table A1 and related information was added to the main text in lines 612-622.

2) in reply to reviewers 1 and 3, added a figure in Appendix B, Fig. B8, showing the observed spatial distribution of secchi depth averaged over the period 2001 to 2017. The difference between the model and observations is also shown. The figure is described in Appendix B and referred to in the main text in lines 601 and 671.

3) in reply to reviewer 3, added two new figures in Appendix B, Figs. B3 and B4, of the time series for nitrogen and phosphorus from model and observations at three stations in the southern North Sea. This required renaming most of the figures in Appendix B. These figures are described in Appendix B and referred to in the main text in lines 557 and 582.

4) in reply to reviewer 3, split figure 12 in two to improve the readability of its subfigures. Now Fig. 12 is then Fig. 12 and Fig. 13. This means that the former Fig. 13 is now Fig. 14.

5) in reply to reviewer 3, prepared documentation for the river forcing that was applied in this study. This will be made public, together with the dataset, upon acceptance of the manuscript in ZOTERO as Ruvalcaba Baroni et al. (XXXX) with DOI number 10.5281/zenodo.10185658.

6) in reply to reviewer 3, added a table (Table 2) summarizing the tredns of nutrient, chlorophyll-a and oxygen at ANHOLT and BY15.

**2 Additional changes**

1) We have added 4 co-authors and their affiliation (line 1) to our manuscript as they have been involved in the writing of large parts of the code for SCOBI, for the coupling of NEMO-Nordic to SCOBI and for setting the first runs of NEMO-SCOBI on the supercomputer. They have also contributed to the discussions of the results and writing parts of the text. Consequently, we have modified the section "Author contribution".

2) We have changed the line style of Figure 4 to meet the color blind test.

3) We have also improved the conclusions to better highlight the relevance of our work as follows (lines 732-733 and 753-756):

**"We have here presented and evaluated NEMO-SCOBI; a new coupled physical-biogeochemical model configuration for the Baltic and the North Sea."**

**"NEMO-SCOBI is, to our knowledge, the third physical-biogeochemical model covering the area of the Baltic and the North Sea (Daewel and Schrum 2013 and Maar et al. 2011). Future model inter-comparison studies of these three model systems have the potential to give valuable scientific insights into model performance and process understanding, and NEMO-SCOBI is thus an important addition to the model ensemble of this region."**

**3 Detailed changes based on reviewers' comments**

Here we specify, point-by-point, all changes that were done according to the reviewers' comments. Note that previous authors' answers are given in **bold** and revised points in **blue**. Line numbers in black are from the old version of the manuscript, while in blue are the updated line numbers.

**3.1 Reviewer #1**

1) Line 45-50: Not clear why primary producer assemblages are mentioned here for NS but no equivalent description for BS. I suggest to introduce the physical aspects first, then biogeochemical and finally the plankton assemblages.

**We indeed only mention cyanobacteria in the Baltic Sea, as these are the most problematic species there. However, we will restructure the paragraph and add more details on primary producers in the Baltic Sea. The second paragraph of the introduction has been restructured and more information regarding primary producers has been added in lines 58-68.**

2) Line 66: Replace "recycling of benthic phosphorus minerals" with "recycling of phosphorus from sediments"

**We will change the wording as suggested. Done, now in line 82.**

3) Line 70-71: Please update this setting of the context with a citation of the following publication and potentially references therein: "The Baltic and North Sea Strategic Research and Innovation Agenda BANOS SRIA 2021 The final BANOS SRIA draft of the proposed, new, joint Baltic and North Sea Research and Innovation Programme -BANOS BANOS CSA Deliverable 1.5".

**We will adjust the context to include this relevant work (Koho et al., 2021). Reference to this work has been added in lines 86, 89 and 648.**

4) Caption of Fig. 1: Capitalize SHARK

**We will capitalize this. The caption of Figure 1 has been changed to add 3 stations in the North Sea and 'shark' has been removed from the caption.**

5) Table 1: Modify mg CHL m-3" to mg Chl-a m-3"

**We will change this. 'mg CHL m-3'has been changed to 'mg Chla m-3' in Table 1.**

6) Figure 2 legend: Typo in atmosphere

**We will correct this. This has been corrected.**

7) Line 183: Typo 100 kton N/yr

**We will correct this. This has been corrected, now in line 203.**

8) Line 195: Does this mean "reduced to 0.3 x and 0.75 x the original value, respectively"? Please clarify. Also clarify (as implied in Fig. 2) that the resulting nutrient flux to the sea is entirely in the dissolved fractions and comment briefly on simplifications with respect to e.g. fluxes of TP, that in reality are largely particulate.

**Yes, the reduction factors are: 0.3 x detritus P, and 0.75 x detritus N. This means that we have reduced the riverine input of particulate organic phosphorus and nitrogen (in our model assumed to be detritus of nitrogen and phosphorus) by 75% and 30%, respectively, to account for coastal retention and bioavailability. This has been clarified in the text, lines 219-220.This is based on values at the Swedish coast (Edman and Anderson, 2014; Eilola et al., 2009). These factors are largely unknown for most of other coastal areas in the model domain and therefore the same reduction factors are used throughout the domain. We will ensure that this is clear in the text. Information on this has been added in the text, lines 221-223.Regarding the resulting nutrient flux, figure 2 shows the total phosphorus and nitrogen (so the sum of dissolved inorganic and detritus) that enters the model domain. We will ensure this becomes clear in the text. This has been clarified in the caption of figures 3 and 4. Please also see our reply to comment 30.**

9) Line 286: Modify to e.g. "depending on their proximity to one another"

**This will be changed as suggested. Done, now in line 317.**

10) Line 301: Typo "to analyze"

**This will be corrected. Done, now in line 332.**

11) Line 315: Replace 80s with 1980s

**This will be changed as suggested. Done, now in line 347.**

12) Line 317: Typo observations are too low

**This will be corrected. The sentence has been corrected, line 350.**

13) Line 326: Replace dont with do not

**This will be corrected. Done, now in line 359.**

14) Line 346: Typo statistically

**This will be corrected. Done, now in line 381.**

15) Line 347: Remove "not studied here", it confusing to state this

**This will be removed as suggested. Done.**

16) Line 349: Rephrase to e.g. "There is a lack of observational data for bottom water oxygen during the period 1975-1995".

**There is indeed a lack of observations before the 1995 at ANHOLT and at most stations in the Skagerrak-Kattegat region. However, it is not limited to oxygen. While nitrate observations are totally lacking before $\sim$ 1995, phosphate and oxygen observations are scarcer (with fewer data points per year) before the year around 1995. Note that the year where observations become more abundant is different at each station in the Skagerrak-Kattegat, but fall between 1992 to 2000. However, we agree that the sentence is confusing. This will be rephrased as follows:**

**"In the Skagerrak-Kattegat region (e.g., at ANHOLT, fig. 6) observational data is largely lacking before the 1990s, including that for oxygen in bottom waters and therefore model trends may be more representative of the system for historic values." This has been rephrased in lines 383-385.**

17) Line 408: Typo less than

**This will be corrected. Done, now in line 442.**

18) Line 421: Typo extent

**This will be corrected. Done, now in line 456.**

19) Line 422: Modify to The main inference of...

**This will be corrected. Done, now in line 457.**

20) Line 425: Modify to Despite such specificities of..."

**This will be changed. This has been rephrased in line 460.**

21) Line 457: Modify to its (no apostrophe)

**This will be corrected. Done, now in line 493.**

22) Line 471: Modify to "bias is" or "biases are"

**This will be corrected. Done, now in line 507.**

23) Line 498: Modify to most models

**This will be corrected. Done, now in line 533.**

24) Line 572: How could low oxygen theoretically inhibit primary production? Not easy to understand what is meant here.

**"low oxygen" will be removed from the sentence. Done.**

25) Line 574: The model vs. data discrepancy in the timing of the late winter/spring bloom in the Kattegat is one of the key question marks raised in the study, and indeed it is highlighted in Fig. 7. I think it deserves more elucidation at this point in the discussion, for example an assessment of the degree to which light or temperature might be inhibiting the early onset of the bloom in the model. The authors return to this in Section 3.6 Future work and data gaps but in the current version the reader is left hanging for an explanation after it is established that there is no nutrient limitation at the time in question.

26) **We will add, most likely in the appendix, a plot of the sensitivity of modeled phytoplankton growth to light and temperature. We will then add more information on its relevance in the text here. A sensitivity figure of growth rate to temperature and secchi depth has been added to Appendix A, Fig. A1. Related text has also been added in lines 612-617. Please note that we attribute the bloom delay occurring in the Kattegat-Skagerrak mostly to the fact that the model does not capture the seasonal variations of the light attenuation depth in this area. In addition, the model temperature is underestimated during spring at for example ANHOLT (by approx. two degrees, see Figure 1 below). This indicates that the stratification in the Skagerrak-Kattegat is not sufficient in the model during spring and may inhibit phytoplankton growth at the exact time of the blooming period. We realize that we have not included the latter in the text and will be added as a discussion point. We have looked at this in more detail and, actually, the temperature (stratification) is unlikely to be the problem regarding the delay in phytoplankton bloom at ANHOLT. This is because the model bias for temperature is only found for March/April, which is later than the phytoplankton maximum seen in the observations (February). In February, the observed temperatures and model temperatures are very similar (2.8 vs 2.5 $^o$C), as well as for observed and modeled salinity. Therefore, we attribute the phytoplankton bloom delay to a too low sensitivity of phytoplankton growth to light. This is now highlighted in the main text, lines 617-622. Please also see our reply to reviewer #3, on pages 4 and 5, comment 0.8.**

27) Line 590: Again please check the BANOS SRIA for up-to-date statements about the need for integrated modeling of BS-NS system

**Yes, we will. The reference to BANOS (Koho et al., 2021) has been added and the text modified accordingly, line 649.**

28) Line 595: Citations in brackets

**The missing brackets will be added. Done, now in line 654.**

29) Line 604: Modify to "in Ford et al. (2017)..."

**Brackets for this citation will be removed. Done, now in line 667.**

30) Line 616-623: This is a very important section of the discussion: the model assumes (if I understand correctly) that a certain bioavailable fraction of nutrients enters the sea, directly into the dissolved

phase state variables PO4, DSi, NO3, NH4 (Fig. 2). This is of course a large simplification of reality, where there is transfer between bioavailable and non-bioavailable fractions within the coastal filter, as well as heterogeneous removal of nutrients. Different types of coastlines may behave quite differently in this regard, see eg. Asmala et al., L&O 62 (2017). Are there any of the observed model vs. data discrepancies e.g. in near shore nutrient or Chl-a concentrations, that could be affected by this simplification? If so, it would deserve some mention higher up in the discussion for those specific areas.

**The bioavailability factors are only used on the particulate organic matter that enters the sea with the rivers. The factors are to one part accounting for the different quality of the organic matter from land, of which some part is refractory and will not decompose fast enough to be a part of the marine nutrient circulation within the model system. In the ocean model, this factor also accounts, to some extent, for the filtering effect of the coastal zone. In SCOBI, the bioavailable detritus (particulate organic material that has been decreased from the rivers) enters the sea into the detritus pools for the specific nutrient and then go through the transport, grazing, sinking and decomposition processes. Thus, the bioavailable fraction of the organic pool does not directly enter the dissolved nutrient (see Figure 1 in the main text). We will better clarify this in the text. This has been clarified in line 220-221.**

**Regarding the filtering effect, the reviewer is correct that different types of coastlines have different response to nutrient removal. The bioavailibilty factors used here are typical values for the Swedish coast and the Baltic Sea Eilola et al. (2011) , which are also not too far from those in Asmala et al. (2017) (16% of nitrogen and 53% of phosphorus removal from land versus 30% and 75% assumed here). These factors are, however, poorly quantified for the North Sea. As a first approach we assumed a homogeneous value for our entire domain following the approach of Eilola et al. (2011). This has been added in line 220-223. In addition, we did perform a scenario with the exact same settings as that presented here, but without any retention factor. When comparing results from the latter to the hindcast scenario here, the spatial effect of the retention factor was minor, especially for nitrogen which decreased by one or two mmol/m3 almost homogeneously in our domain. The chlorophyll-a was not much affected by this and therefore we dedicated only minor discussions to this in lines 620-624. We think that the major effect of this is confined to specific coastal areas, probably those limited by P. However, we agree with the reviewer that this is relevant and deserves more explanation in the text. This also links to our reply to reviewer #3, comments 12 and 13 where we say that we will look further to better detect which coastal regions are most affected by the rivers. Additional information on this has been added in lines 700-703. We will also add the reference of Asmala et al., 2017 in line 621. The latter reference has also been added in line 696.**

31) Line 666: typo therefore

**This will be corrected. Done, now in line 736.**

32) Appendix Line 737: Important: should this read "BOPREM-BIP decreases with increasing salinity and decreasing bottom oxygen concentrations..." ? That is the implication of the equations and following text, and the logical relationship.

**Yes, that is correct and 'decreasing' will be added before 'oxygen concentrations'. Done, now in line 826.**

33) Appendix Line 771: Typo from

**This will be corrected. Done, now in line 862.**

34) Appendix Table A2: Comment on the validity of using a single porosity value for the entire NS-BS system.

**We are fully aware that porosity is not homogeneous in the Baltic Sea and the North Sea - see for example a model study of benthic phosphorus cycling in the Baltic Sea by co-authors of this study, where porosity was region-specific (Almroth-Rosell et al., 2011, 2015). The results presented here are, however, from the first version of NEMO-SCOBI which indeed is simplified and has currently a hard-coded porosity. It is definitely an important point in our list of future developments for NEMO-SCOBI, where it will be made regional-specific. A comment on this will be added to the text. A comment on this has been added in appendix A2, lines 848-850.**

**3.2  Reviewer #2**

1. While I was reading the manuscript, although it is clear that having a coupled model for the North Sea-Baltic Sea represents a clear advancement for the scientific community, it was not clear to me if its performances are better or worst that other 3D modeling tools available for the region that are cited by Authors at line 80 for example. Are the performances of NEMO-SCOBI better or worse than other modeling tools? Does NEMO-SCOBI improve the biases of other modeling tools or not? Im asking that since a potential user should have all this information to decide to use your modeling tool/simulated data than others provided by another modeling tool.

**As mentioned in lines 450-455 and specified in line 585, our results show that NEMO-SCOBI is neither better nor worse than previous model results depending on the variable and area. This means that in some areas our model performs better than other models for a specific variable and in others not (details are in sections 3.2 and 3.3). Importantly, the aim of this paper is to evaluate NEMO-SCOBI, not to make a model intercomparison, which requires a total different methodology and extensive additional work (see for example Gröger et al., 2022). Because no model is perfect, the best way is to combine results from several models and use ensembles, when possible. Currently, there are only three biogeochemical-ocean models (including NEMO-SCOBI) that account for both the North Sea and the Baltic Sea, and therefore our work represents an important step to create representative ensembles from independent models that cover both the North Sea and the Baltic Sea. We have also discussed in detail (in section 3.4) the strong and weak points of NEMO-SCOBI model, as well as its possible application as a stand alone model. These points are also summarized in section 3.5. As concluded in line 663, NEMO-SCOBI can fully compete with previous models that account for the North Sea and the Baltic Sea. However, we will read through the text to see where we can make this more clear and extend section 3.5. More text in section 3.5 has been added, lines 638-645. One of the major benefits in a model covering both North Sea and Baltic Sea is that the open boundary is moved out from the transition zone between the seas (as mentioned in lines 580-581). This is a very important step to better study the Baltic Sea and the Skagerrak-Kattegat area as it allows to study these areas without interactions with the open boundary. We will highlight this more in the text. This has been highlighted in lines 625-635.**

2. Strictly correlated with that: please provide more quantitative information about these performances. Good, Comparable, Acceptable and so on are not informative from my point of view. Please quantify in the manuscript the biases and the values of the trends and their statistical significance.

**As written in section 2.2.3, and mentioned in lines 265-270, the words "good" and "acceptable" have a statistical value attached to them that is also used in other studies - see for example Edman and Anderson (2014); van Leeuwen et al. (2023). They are specified according to the Pearson correlation coefficient (r) as 1-r and the cost function (CF), which are statistical methods broadly used for model validation against observations. In summary, the model performance is considered to be "good" when both 1-r and CF of an assessed parameter fall within an inner quarter circle (see figs. 10 and 11). If the CF evaluation is equal to or lower than 1, the model values are very similar to the observed values. A value of 1-r that is equal to or lower than 0.33 means that the variability in the model for the evaluated parameter matches well that the observations. When combining both 1-r and CF, if values fall withing the inner circle, it means that both the absolute values and the variability of the model are in good agreement. The model performance is considered to be "acceptable" if the combined value of 1-r and CF fall within the inner and outer quarter circle (i.e. when 1-r and CF values are in between 0.33,1 and 0.66,2). If 1-r and CF values are larger than 1 and 2, respectively, the model data is far from observations, meaning that there are significant differences in both absolute values and variability between model and observations. We will add clarifications where relevant in the results section. Note that the value of the absolute bias is also given where relevant in the text (e.g., line 308 and line 357).**

**Regarding the trends, the regression analysis and the corresponding p-values (which give the statistical significance) are calculated and mentioned where relevant in the results (e.g, lines 319, 321, 326, 346 and 364). We do not see the need for more statistics in this case, as the goal of this paper is not to detect specific temporal trends, as mentioned in line 234.**

1) Line 11: What do you mean with differences? please explain

**We mean that the model values and the values given by measurements for chlorophyll-a and nitrate differ considerably in the mentioned areas both in time and space. We will rephrase this sentence as follows:**

**"However, there are important differences for chlorophyll-a and nitrate between model results and observations in coastal areas of the southeastern ...". This has been rephrased in lines 10-12.**

2) Line 26: Modelled or Observational studies? Please explain

**Previous biogeochemical studies in the Kattegat-Skagerrak; both modelling and observations. This will be added to the text. This has been added in the text, line 26.**

3) Line 31-40: What about the intermediate layers of the basin?

**This paragraph will be adjusted (also in response to reviewer #1, comment 1. More detailed information on the water column structure will be added. This paragraph has been rephrased and information about intermediate waters in the Baltic Sea has been added, lines 34-36.**

4) Line 54: What do you mean with point sources. Please explain

**A point source is any single identifiable source of pollution from which pollutants/nutrients are discharged, such as sewage. This clarification will be added in the text. This has been added in lines 70-71.**

5) Line 70-76: Please provide more information about these differences since it is the starting point for showing that this model is a step forward for the scientific community

**We do not fully understand what differences the reviewer is referring to. If the reviewer refers to the difference between previous studies and our study, this is mentioned in lines 74-84 - i.e., our model covers both seas and only 2 other 3D models have this similar domain. In models that cover either the Baltic Sea or the North Sea, the Skagerrak or the Kattegat represent the boundaries of the models, which implies having simplified dynamics in these areas (this is written in lines 579-585 and also mentioned in our conclusions, line 659). Having several models covering the same area is a big advantage, as each model comes with their respective pros and cons. We have a dedicated section where we discuss the advantages that NEMO-SCOBI provides. Therefore, we will not add information on this in the introduction, but we will better highlight this in section 3.5. Please also see our reply 0.1 to reviewer #3. This has been better highlighted in section 3.5, lines 637-644.**

6) Fig.1: It would be great to have super imposed here the bathymetry of the domain (eventually using contours)

**Adding the bathymetry superimposed to this map it will be too messy and confusing. However, we will consider to add the bathymetry next to this figure or in appendix. Because the manuscript already has too many figures and because the bathymetry of NEMO-Nordic has already been published, we have instead added a sentence in the caption of Figure 1 pointing to the reference where the figure of the bathymetry of NEMO-Nordic can be found.**

7) Line 107: Did you assess the drifts in the tracers to assess whether 14 years are (lets say) enough for the spin up or not? In many ocean/biogeochemical models even 30 years are not enough to stabilize the numerical solutions.

**Yes, the years that are actually considered for validation correspond to those with less drift for biogeochemical parameters. The exact year is difficult to pick due to lack of observations before the 1980s. Therefore, we also performed several other runs also starting from 1961 and compared the drift. The initial drift generally decreased around the year 1975 in all runs. In addition, the biogeochemistry was not initialized from scratch - i.e. with homogeneous 3D tracer fields. The spinup was started from earlier SCOBI runs that represent already the physical-biogeochemical conditions as imprint in the initialization followed by the actual spinup. We will better clarify this in the text, line 168. A line on this was added, now in lines 190-192.**

**For the physical initial conditions we used restart files for the year 1973 from the simulation in Hordoir et al. (2019) that were the closest to the observations for physical properties at the start of the simulation (adding another 12 years of spin up).**

8) Line 113: what kind of grid are you using? Regular, structured/unstructured? Please explain

**We use a regular grid. This information will be added. This was added in line 127.**

9) Line 119: Is Iron not important in your domain of study? (since I remember that iron is important in the global ocean).

**In case of dissolved iron in the water column as a limiting substance for the phytoplankton growth, the answer is no. Iron is not important in the domain of our study. However, for the phosphorus cycling iron may be an important factor for the ability of the sediment to adsorb dissolved phosphate. One of the greatest challenges in implementing iron in this domain is the lack of observational data of iron in both the deep water and in the sediment as well as in the supply from atmosphere and land. Until enough observations are available, iron is assumed not to be a limiting factor.**

10) Section 2.1.2 I would include in the paragraph the treatment of light and PAR (that I see located in the SM). How is primary production parametrized? Did you use Q10 formulation? Please explain here.

**The full equations for primary production in NEMO-SCOBI are detailed in appendix A1 (equations A1 to A14) as written in lines 152-153. The specific equations where PAR is used is shown in equations A1 to A5. These equations shows that the modelled growth of phytoplankton is depending on temperature and can be limited by nutrients or light (i.e. we do not use Q10).**

11) Line 155-170: How tick is your sponge layer in the ocean domain? Why do not you use ORAS5 that is more recent than ORAS4? Did you use bulk formulas for latent and sensible heat fluxes? What formulation of albedo did you use in your ocean model (since it influences the quantity of shortwave radiation reflected by the surface)

**Because NEMO was originally intended for global setups with periodic boundary conditions, it has developed differently than other ocean model. Except for the nesting option called AGRIF, there is no sponge layer option in NEMO. Here, we have used the same settings as in Hordoir et al. (2019) for open boundary conditions (key_bdy). This means that we do not consider a sponge layer per say, instead we use a default thickness of one grid cell along the open boundaries. However, to make the model stable near the boundary, the viscosity/diffusivity coefficient has been significantly increased (by a factor of $\sim$ 10) above the halocline near the boundary, as explained in detail by Hordoir et al. (2019).**

**The reason we use ORAS4 instead of ORAS5 is simply that we have not yet prepared ORAS5 OBCs, but we may do it in the future. We used the CORE bulk formulation as mentioned in Hordoir et al. (2019).**

**Regarding the albedo, we used the original formulation and constants for the ocean, ice and snow as given in LIM3, which is the ice model to which NEMO is coupled with (Vancoppenolle et al., 2008). The LIM3 has been broadly used for climate simulations and operational oceanography. Its code is open access and therefore detailed formulation can be seen here.**
**In summary, the feedback of the albedo of ice promotes high absorption of shortwave radiation.**

**The constants used could of course be tested and better tuned for the Baltic Sea and the North Sea, but this has not yet been done.**

12) Fig3-4 I would put the runoff as first panel in both figures.

We prefer having the top panels with nutrients (TP and TN) as these are the most relevant for this work.

13) Line 289: What do you mean with similar? please explain

**we will add more detail as follows:**

**"This is because the model response at these two stations is similar, both in magnitude and trend, to that at stations within their corresponding region ..."** This has been rephrased in now lines 320-321.

14) Figure 6 and after: please move the small panel Model Obs .. outside the first panel since it covers partially the lines beneath.

**We will improve this figure.** Figures 6, 7, 8 and 9 have been improved and labels moved so that they do not cover the results.

15) Line 388 What do you mean with good period

**This will be rephrased as follows:**

**"All these stations show good model skill for both the period and the seasons when evaluated for temperature and salinity (Fig. 10a and Fig. 10b)."** Done, now in lines 424-425.

16) Paragraph 3.4. I think that this paragraph should go before the comparison between models and observations at the single stations. This could provide a general overview of the model performances better that the comparison with single point.

**We have taken this feed back into consideration, but came to the conclusion that we do not agree with the reviewer suggestion. The comparisons per station are given with time (as times series and interannual averages). This gives a better overview of when the model has large bias or not. In addition, figures for only 4 stations are shown, but the analysis is done for 27 stations (as written in line 210), which are regrouped by regions and therefore, also give a spatial overview. The analysis is then summarized with the model evaluation plots (Figs 10 and 11) in sections 3.2 and 3.3.**

17) Line 520-541, Line 551-560: These parts should go in the introduction since they provide an interesting description of the Baltic-North Sea system.

**We will add information on this in the introduction.** Information on this has been added in the introduction: lines 42-46 and 53-57.

18) Figure 13-14: Put in first row the observations.

**This will be changed according to reviewer's suggestions.** Figure 13 has been changed. Note, however, that Figure 14 shows only model results (no observations) and therefore kept as it was.

**3.3 Reviewer #3**

**Review overview**

The manuscript is in general well written (some grammatical errors remain) and includes an exhaustive validation exercise. The authors focus a lot of attention on the fact that this set up includes both the wider North Sea and the Baltic Sea: most model set-ups do either one or the other due to the different governing mechanisms. For this set up the biogeochemical model was extended, from its natural domain the Baltic, to cover the North Sea and Channel areas. Therefore is seems strange that the validation is mainly focused on the Baltic and the Kattegat/Skagerrak area (for which SCOBI was designed) and hardly on the new areas it now has to represent. The authors truthfully cite a lack of observational evidence in the newly covered areas, but some station validation is surely possible. The biogeochemical model tends to capture the phosphorous and oxygen dynamics pretty well, but this is what I would expect from a model designed for and tuned to the Baltic Sea. The North Sea has very different dynamics, and although hypoxia and anoxia can occur there as well they are not a defining feature of the modern system. The authors themselves state that the model performs better in P-limited areas than N-limited areas, and the North Sea is mainly the latter. Modelled phytoplankton consists of diatoms, flagellates and cyanobacteria, but the latter hardly play a role in the salty North Sea and Channel: what groups could be added for a better representation of phytoplankton in the North Sea? Would addition of Phaeocystis spp. be an option (the North Sea nuisance species), and how good is the model at representing primary production at pycnocline depth? Figures 7 and 9 seem to indicate an overestimation of the top mixed layer depth, which should be discussed more. Overall, the model misses the seasonal dynamics of both systems (e.g. timing of spring bloom, autumn bloom), which could be due to the light climate, silica dynamics, phytoplankton parametrization or the nutrient inputs (temperature is usually easy to get right). Without additional analysis it is hard to say what is the main cause, particularly as this might differ per region. But some light attenuation validation could be added, as could a comparison with continuous Chla observations (few locations, usually short temporal coverage) or comparison with Chla satellite observations to get a better grip on this issue. I miss an in-depth analysis and discussions on these topics in the current manuscript. But the manuscript itself is worth publishing, as the model represents a valuable addition to both North Sea and Baltic modelling efforts.

**We thank reviewer #3 for the interest and the thorough review of the manuscript. The reviewer pin points already here important questions that we dissect and address one by one below:**

"Therefore is seems strange that the validation is mainly focused on the Baltic and the Kattegat/Skagerrak area (for which SCOBI was designed) and hardly on the new areas it now has to represent. The authors truthfully cite a lack of observational evidence in the newly covered areas, but some station validation is surely possible."

**0.1) The reviewer has a good point. However, it should be noted that monitoring programs are handled very differently in the North Sea than in the Baltic Sea. In the North Sea, single stations with long time series are greatly lacking or not publicly available, mainly confined to coastal areas and have less observations per year than in the Baltic Sea. The latter means that monthly or even seasonal profiles cannot always be obtained, and therefore, plots such as 7 or 9 are not that useful for the North Sea and statistically not representative. Many observations in**

the North Sea come from cruises limited in time that do not always cover the same exact transect from year to year. In addition, from our point of view, we showed that the evaluation per area is also very informative for detecting regional model biases. Following earlier studies (e.g. Pätsch et al., 2017) in the North Sea, we have prioritized the spatially compiled observations instead of isolated station data. Also note that both the Kattegat and the Skagerrak are new areas with respect to models that cover either the Baltic or the North Sea. In such models, both these areas constitute the boundaries of the models, which implies having simplified dynamics in these areas (this is written in lines 579-585 and also mentioned in our conclusions, line 659). We will add more discussion on this in section 3.5, to make it more clear that both Kattegat and Skagerrak are better represented in models covering both the North Sea and the Baltic Sea. **This has been added in lines 626-634.** In this respect, the North Sea, even though it can now be studied with NEMO-SCOBI, is mainly used here to better capture the complex dynamics of the Skagerrak-Kattegat transition zone. As a Swedish governmental institution, covering the entire Swedish coast including both the Skagerrak and Kattegat areas is important. Expanding the model to cover the North Sea allows for better dynamics in both these complex areas. This will be better emphasized in the text. **This has been added in lines 636-637.** Importantly, we have analyzed results from several stations in the Skagerrak and Kattegat areas (see figure 1 in the main text), therefore covering also both of these new areas. Please also see our reply to comment 17. **Please note that we have selected three stations in the southern North Sea and plotted time series for nitrate and phosphate. The figures are now added to the appendix and cited in the main text. For more detail, please see our reply to comment 17..**

The biogeochemical model tends to capture the phosphorous and oxygen dynamics pretty well, but this is what I would expect from a model designed for and tuned to the Baltic Sea. The North Sea has very different dynamics, and although hypoxia and anoxia can occur there as well they are not a defining feature of the modern system. The authors themselves state that the model performs better in P-limited areas than N-limited areas, and the North Sea is mainly the latter.

**0.2) Our analysis indicates that chlorophyll-a is best captured in areas where the same nutrient is limiting in both model and observations (lines 498-503). The reviewer could have been confused by line 417 (in section 3.2): "Because phosphate is better captured in the model, chlorophyll-a is also best captured at sites where phytoplankton is limited by phosphate ...". This phrase refers to stations in the Baltic proper (and not the North Sea). However, we agree that this is confusing and misleading. Line 417 will be rephrased as follows:**

**"In the Baltic proper, good model skill for chlorophyll-a is found at six stations (i.e. BY4, BSCIII-10, HANÖBUKTEN, BY10, BY15 and BY20), which also show good or acceptable model skill for both nitrate and phosphate". Now, added in lines 453-454**

**0.3) It is actually the other way around. It is surprising that by expanding the model to a such different dynamic region, and with very little additional tuning, most processes in the North Sea are still well captured. This point will be better highlighted in the text. This is now highlighted in lines 643-644. Note that the entire paragraph, lines 637-644 has been rephrased according to further reviewer's comments.. It is only specific coastal areas (mentioned in line 457) which are not well represented in NEMO-SCOBI and in other models as well. Such areas are affected by multiple uncertain factors (e.g., riverine input, non-Redfield ratios for phytoplankton growth, light penetration depth, etc). The fact that phosphorus is better captured in the model**

in both the North Sea and the Baltic Sea could be indeed linked to a better model representation/parametrization of the phosphorus cycle than that for the nitrogen cycle. It can also be related to a better phosphorus forcing than that for nitrogen. It is true that the oxygen and phosphorus cycle in the Baltic Sea are key study points to understand this system, but this knowledge also applies to the North Sea.

Modelled phytoplankton consists of diatoms, flagellates and cyanobacteria, but the latter hardly play a role in the salty North Sea and Channel: what groups could be added for a better representation of phytoplankton in the North Sea? Would addition of Phaeocystis spp. be an option (the North Sea nuisance species)

**0.4) Yes, the cyanobacteria species included in SCOBI do not grow in the North Sea, but they do grow in other areas of the domain. As in other biochemical models covering a similar model domain (e.g., ERGOM Maar et al., 2011), we consider diatoms and flagellates to be the dominant species in the North Sea and do not plan to add additional species for the North Sea in this version of the model. Note that the category PHY2 in SCOBI accounts for "flagellates and others" (line 126) and therefore implicitly accounts for other species such as Phaeocystis spp. However, explicitly including a phytoplankton group that reacts on changes in the N to P ratios, as seen in Phaeocystis spp in Dutch coastal waters (Riegman et al., 1992), could indeed improve results in the southern coast of the North Sea. The results presented here are results from the first version of NEMO-SCOBI, which is under constant development. The reviewer suggestion will be certainly considered for future improvements.**

how good is the model at representing primary production at pycnocline depth? Figures 7 and 9 seem to indicate an overestimation of the top mixed layer depth, which should be discussed more.

**0.5) The chlorophyll-a values at the pycnocline are in very good agreement with observations (both seen in the Hovmöller diagrams and the seasonal and period averaged profiles of Figures 7, 9, B1, B2 to panels). The overestimation above the pycnocline seen in the summer months in the Hovmöller diagrams comes from the delay in the timing of the phytoplankton bloom. The maximum chlorophyll-a concentrations is generally well captured by the model at all stations, but comes too late in the year at several station in the Skagerrak-Kattegat and the Baltic proper. This is mentioned in section 3, more specifically in line 335 for the Skagerrak-Kattegat (ANHOLT) and in line 516 for the Baltic proper (BY15). Station-specific differences in chlorophyll-a are also explained in detail for ANHOLT and BY15 and discussed in lines 367-377. However, we could add a line summarizing the fact that the overestimation above the pycnocline for the summer months comes from the bloom delay in lines 571-572. This has been added in lines 373-374 instead, as it fits better the context in these lines..**

Overall, the model misses the seasonal dynamics of both systems (e.g. timing of spring bloom, autumn bloom), which could be due to the light climate, silica dynamics, phytoplankton parametrization or the nutrient inputs (temperature is usually easy to get right). Without additional analysis it is hard to say what is the main cause, particularly as this might differ per region. But some light attenuation validation could be added, as could a comparison with continuous Chla observations (few locations, usually short temporal coverage) or comparison with Chla satellite observations to get a better grip on this issue. I miss an in-depth analysis and discussions on these topics in the current manuscript. But

the manuscript itself is worth publishing, as the model represents a valuable addition to both North Sea and Baltic modelling efforts.

**0.6) To capture all aspects of the seasonal cycle is challenging. We have, however, shown that the seasonal cycle for all assessed parameters are well captured at least in several regions of the model (e.g. all dots falling within the inner and outer quarter circle in Figure 10 and written in lines 418-420). In the text, we do highlight the specific parameters that are not well captured for specific months and areas in order to understand what to improve in the model. We also acknowledge that the reviewer consider the manuscript worth publishing.**

**0.7) Regarding the phytoplankton bloom delay, it is mainly occurring at some stations in the Skagerrak-Kattegat transition zone and the Baltic proper, as highlighted in lines 421-424. To our knowledge, no current biogeochemical model of the Baltic Sea and/or the North Sea have good representation of chlorophyll-a everywhere in their domain. Much of the modelling community work is focusing in improving this, but it is not trivial work. NEMO-SCOBI is not an exception and therefore we are currently working, in a parallel study, in improving chlorophyll-a concentrations, where we indeed aim to include high temporal resolution data and satellite data sets. For the present study, this is not feasible, as it requires much more additional information that would make the paper indigestible. For example, satellite data should come with additional analysis on its own, especially for chlorophyll-a, and additional evaluation periods as well as additional stations would be required. Importantly, in this paper we focus on long-temporal trends. In addition, the model has been partially validated against chlorophyll-a satellite data in the North Sea in a recent study (van Leeuwen et al., 2023). We will add the latter information. This has been added in lines 640-643.**

**0.8) Regarding the light attenuation validation, the reviewer is correct and this is indeed an important parameter for phytoplankton growth. In fact, as mentioned in line 562, we have analyzed the light attenuation in the model and it is overestimated in the Baltic proper and the Skagerrak-Kattegat transition zone. The figure for observations and model results averaged over 1975 to 2017 for winter and spring are shown below, as an example. We will specifically add a line summarizing this information and specify that the model overestimation of the light attenuation in the Skagerrak-Kattegat is likely linked to the delay of phytoplankton bloom in lines 574-577. This has been added in lines 613-622 as well as 2 new figures: Fig. A1 shows the sensitivity of the light attenuation in the model to temperature and chlorophyll-a concentrations when considering all other variables constant and Fig. B8 shows the seasonal observed Secchi depths averaged from 2001 to 2017 and the corresponding differences between observations and model results. Figure A1 is now described and shown in Appendix A. It is cited in the main text in line 616, line 622 and line 680. Figure B8 is now described in Appendix B, lines 910-912 and cited in the text in line 601 and line 672. Please see also our reply to reviewer #1, comment 26.**

REVIEWER'S RECOMMENDATION

Moderate revision

I like the manuscript but would like to see further validation results added to it for 1. the North Sea area and/or Channel area (main) and 2. the riverine forcing used (appendix). This would require no new simulations but new post-processing. I would also like to see figures 6 and 8 reorganized and figure B3

moved from the appendix to the main article. If necessary, figure 5 could be banned to the appendix to make way for figure B3.

**0.9) In our reply to the reviewer's first comment (reply 0.1), we have detailed the reasons why we have chosen not to validate model results for the North Sea at single stations. Please note, however, that we are aware of higher temporal resolution observational data for recent years and will be included in future work. We would like to pin point again that validation for stations in both the Kattegat and Skagerrak (new areas for this model) have been done and shown (Figures 1,6,7,10 and B1).**

**0.10) Regarding figure B3, we are not sure that moving figure B3 to the main text will be better, as it shows similar (almost repeating) information as figure 11, due to the fact that observations are mainly confined to surface waters. However, we will consider this within the author team (please also see our more detail answer to comment 23, page 11).** **After discussion within the author team, we would prefer to keep figure B3 in appendix.** **Regarding figures 6 and 8, please see our reply to comment 19.**

**0.11) We would prefer to not ban figure 5, as it clearly shows the lack of observations in the North Sea with respect to the Baltic Sea. This means that extra care must be taken when analysing trends in the North Sea, that some analysis cannot be performed and that analysis for the North Sea will not have the same statistical significance as those for the Baltic Sea. This is one of the main reasons we did not show observations for the North Sea at single stations. Regarding the river forcing, please see our more detailed answer below (comments 12 and 13).**

**Detailed Comments**

1) Line 14-16: The validation is in agreement with assessment areas ... ? Do you mean that you are using assessment areas for the validation (i.e. method), or that validation within these assessment areas confirms with reported values in those same areas (i.e. validation result)?

**This has been rephrased as follows (lines 14-16): "Our model results are validated for individual areas that are in agreement with policy management assessment areas ..."**

2) Line 19: the references are not in alphabetical or chronological order.

**This will be corrected to be in chronological order.** **This has been done (line 19).**

3) Line 44-45: too many commas and bad grammar. Not sure what is meant here: such areas refers to the deeper parts of the North Sea (previous line), but those are not coastal.

**This will be rephrased.** **This has been rephrased in lines 34-36.**

4) Line 50: referred to as cyanobacteria and I miss a reference for the statement that cyanobacteria do not grow in the North Sea.

**This will be rephrased (also in response to reviewer #1, comment 1), but the overall message is that the filamentous (and sometimes toxic) cyanobacteria that we parameterize in the model in the Baltic Sea do not grow in the North Sea, because the latter is too salty for this species. We will add the following reference:**

Olofsson, M., Suikkanen, S., Kobos, J., Wasmund, N., Karlson, B., 2020. Basin-specific changes in filamentous cyanobacteria community composition across four decades in the Baltic Sea. Harmful Algae 91, 101685. https://doi.org/10.1016/j.hal.2019.101685

**The entire paragraph has been rephrased, see lines 58-68. The reference above has been added together with additional relevant references.**

5) Line 78: bad grammar, I suggest , which is particularly true in"

**This will be rephrased according to reviewer's suggestion. This has been rephrased in line 94.**

6) Line 80: but contain biases for

**This will be corrected. It is now corrected in line 97.**

7) Line 87-90: Not sure why this text is here, not relevant to the subject of this manuscript.

**We agree with the reviewer that these lines are not relevant in this section. We will therefore move the sentence to the section "relevance of this study" (section 3.5). This has now been removed from the introduction to section 3.5 in lines 658-661. Note that this entire paragraph has been rephrased.**

8) Fig 1: I would say observational SHARK stations. The acronym is later used with capitals, without those it is rather confusing here. Indeed, SHARK is only explained in line 205, so some explanation is due here.

**Yes, this is a typo and shark should be capitalized (also in response to reviewer 1). The word 'shark' has been removed from the caption as the caption has been rephrased to include the new North Sea stations, which are not part of the SHARK database.**

9) Line 131: please refer to the figure before the textual explanation, to make it easier on the reader.

**We agree with the reviewer that it would be better and will therefore move line 131 to line 121. The line has been moved, now line 137, and rephrased to better fit the text..**

10) Line 148: do you mean that the phytoplankton parameters were tuned to represent both the Baltic and North Sea areas? That is to say, the parametrization the model had previously was tuned to the Baltic and these parameters did not fit with North Sea simulations and so needed adjustment? If so, can you speculate why this was necessary? What processes/groups/functionality difference is there between these areas that make this adjustment necessary?

**The explanation for these adjustments are given in appendix A1 (lines 674-678):**

**"In the SCOBI version coupled to NEMO, rates and dependencies for phytoplankton growth were modified with respect to previous SCOBI versions in order to account for silica limitation of diatoms (not included in earlier versions), to improve the occurrence of dominant groups in both the North Sea and the Baltic Sea and to limit cyanobacteria growth in the Skagerrak-Kattegat transition zone and in the North Sea."**

**For more clarity, this information will be moved to the main text, instead.**

**The former lines were moved from the appendix to lines 162-167 in the main text (with some rephrasing to better fit the flow of the text).**

11) Fig 2: this is a spaghetti diagram, hard to read for the many arrows. I think it is a bad idea to include so much detail that the model visual abstract (which is what this is) becomes visually unattractive. I would leave out the coloured arrows explanation, readers can see for them selves if a flux stays in the pelagic or not. Or use different line styles. Maybe group it a bit more, with all pelagic nutrients together in a circle and 1 arrow going in and out if all nutrients are needed? And why are all the fluxes named in the caption rather than in a separate table?

**This is a matter of taste, but we agree that to meet the color blind test, the usage of colors for the different arrows is not the best choice. We will try different line styles instead. Figure 2 has been improved according to our reply. Additionally, the figure was cleaned from unnecessary arrows. The typo 'athmospere' was also corrected to 'atmosphere' in the legend.Gathering nutrients in one circle would simplify too much the figure and hide relevant information as all nutrients do not have the exact same pathway or dependency. Regarding the fluxes, we could add a table with all fluxes but we do not think it will be relevant for the main text as it would be too much detail. It can then be added in the appendix, but the fluxes still would need to be mentioned in the main text in relation with the figure. If the reviewer really wants the table, we can add it, but we think our choice to mention them in the caption, even if we agree that it is not perfect, it is still the best choice in this case.**

12) Line 167: please provide a reference for the applied reduction factor, assuming this is a generally available dataset. If it is not I dont quite see why the authors would use this particular product.

**The runoff data used here (as mentioned in line 165) comes from a dedicated run performed with the EHYPE model, thus especially done for this study. The EHYPE model was forced with an atmospheric database called Hydro-GFD (Berg et al., 2018), which overestimated precipitation by about 10 percent, according to a validation done by the meteorological group at our institute. Hence the river discharge correction factor of 0.9. This will be better clarified in the text. Therefore, there is no published reference for this particular EHYPE run, but it has been tested against standard checkups within the EHYPE team before delivering it for this work. It also accounts for more realistic number of river outlets than those in observed runoff data sets., which will also be emphasized in the text. The latter has been added to the main text, now lines 208-209. In addition, the EHYPE model (Donnelly et al., 2009), is a hydrological model that has been well validated and broadly used in many different settings (e.g. Donnelly et al., 2016; Hundecha et al., 2016; Nijzink et al., 2018; Macian-Sorribes et al., 2020). In fact, together with LISFLOOD and VIC, EHYPE is one of the most popular models applied for large-regional scale (Piniewski et al., 2022). We have written documentation describing the applied river forcing that will be made available upon acceptance of the manuscript, together with the riverine data set used in this study. This will be published in the ZENODO webpage with reference Ruvalcaba Baroni et al. (XXXX) and doi 10.5281/zenodo.10185658. Note that the year of publication for this dataset will be updated upon publication. All the above information on runoff is detailed in this document (Ruvalcaba Baroni et al., XXXX). We refer to it in the main text in line 185.**

13) I am getting a bit confused about the riverine forcing applied. Am I correct in thinking that you used

- discharge values calculated by a hydrological model, which were adjusted evenly across the domain for a known, uneven model error?

**Yes. A constant reduction factor was applied over the entire model domain, as mentioned in our reply to comment 12.**

- nutrient values based on the same hydrological model, but adjusted for each year and basin to observational values based on two different observational data sets?

**Yes, one data set is used for the Baltic Sea (Gustafsson et al., 2012) and the other for the North Sea (Lenhart et al., 2010). Note that the ICG-EMO data, even if it contains data for the Baltic Sea, it neglects important river outlets in this sea.**

If so, then I think it unlikely that any modeller could replicate your efforts, as they cannot replicate this forcing set. And it makes me wonder why the observational sets were not used directly. This mix up of 3 different sources complicates interpretation of results, which are reported in eutrophication-relevant variables. Please provide more detail on this forcing set (in an appendix), as well as a comparison for a few selected rivers (e.g. some of the larger dots in figure 3) of the applied discharge and nutrient loads compared to the observations that you state you also have. Can some of your mismatches in coastal zones be related to this forcing data?

**This is indeed a large part of the work we did. It remains, however, a side part for this manuscript. There are many possible ways to compile river forcing and we selected the method that we think best fits our purpose.**

**The EHYPE data set covers all relevant rivers and models physically consistent and well calibrated changes in river input according to meteorological forcing. Therefore, as written in lines 188-190, captures well the interannual variability (both for runoff and nutrients). However, it does not yet account for the anthropogenic effect on nutrients (detailed in lines 189-191). The latter is extremely relevant for our study or else our model results for nutrients will be meaningless. Importantly, the ICG-EMO and the Baltic Sea riverine data sets are not directly applied in this study because the EHYPE data includes many more river outlets than in both of these data sets combined (636 *vs* 208 outlet points, respectively). Another relevant point is that the ICG-EMO and the Baltic Sea riverine data sets are not pure observations per say. They have been derived from observations, meaning that several assumptions had to be applied to those data sets. Real observations for nutrients (for example those reported per country) lack not only spatial resolution but also temporal resolution (mostly available per year). Also, the Baltic Sea data (Gustafsson et al., 2012) provides monthly loads per basin, which need to be redistributed in the different rivers of the corresponding basin in the Baltic Sea. Thus, both the ICG-EMO and the Baltic Sea data sets come with large uncertainties. We will add information on this in the main text. Information on this has been added in lines 699-700 and detailed in Ruvalcaba Baroni et al. (XXXX).**

**Regarding the effect of the river forcing on our model results, the nutrient trends in the model closely follow those seen in the riverine data set in many regions of the model domain, especially**

[Figure]

(a) Historical loads for total nitrogen in the Elbe plume area according to the OSPAR assessment area division.

[Figure]

(b) Historical loads for total phosphorus in the Elbe plume area according to the OSPAR assessment area division.

[Figure]

(c) Historical loads for total nitrogen in the Rhine plume area according to the OSPAR assessment area division.

[Figure]

(d) Historical loads for total phosphorus in the Rhine plume area according to the OSPAR assessment area division.

in the Skagerrak-Kattegat area and coastal areas. Thus, improving the riverine data set will certainly lead to improved model results. In fact, improving the riverine data sets in both the Baltic Sea and the North Sea is an active field of research. For example, the ICG-EMO data set is constantly being updated and discussed within a large group of experts from which we are part of (see acknowledgements). More information on the importance of the river forcing will be added in the main text. This has been done in lines 700-703. Note, however, that we will not necessarily show results for one or a few specific rivers, as this implies validating riverine data sets which is out of the scope of this manuscript. Because a detailed validation of the nutrient loads is out of the scope of the manuscript, the comparison between our river foricng and observations is done instead in the dataset documentation. Below, we show some figures included in Ruvalcaba Baroni et al. (XXXX) of total nitrogen and phosphorus loads. These figures show the comparison between our new nutrient forcing (in the figure, referred to as E-HYPE DecVar) and the ICG-EMO and the original E-HYPE data sets for the Elbe plume area (a and b) and for the Rhine plume area (c and d). The E-HYPE DecVar is in good agreement with the ICG-EMO data set, especially from the 1980s and onward.

Currently, the riverine data applied in this study is available on demand but we will look for a place to make it freely downloadable, transparent and citable. Note that, due to its high spatiotemporal resolution, our riverine data set is quite heavy, even if provided in netCDF files and it may not be possible to store it in an open platform. If this is the case, we will add that the dataset is available on demand and provided with a full description. As mentioned above, the created riverine dataset used in this study will be made available upon acceptance of the manuscript on the ZENODO webpage, with DOI 10.5281/zenodo.10185658. A reference to the nutrient loads of this dataset is now added in the main text in line 217-218. Note that the year of publication of this dataset will be updated once the data has been made public.

14) Line 233: the reference year was chosen because of high nutrient values. Where those simulated values or observational values?

**The period was chosen when nutrients where high in both the model and the observations. This will be clarified in the text. This has been specified by adding 'and observations' in line 260.**

15) Line 245: explanation of the applied seasonal delineation (meteorological? astronomical?) is only given in the caption of figure 10. Please provide this here.

**We will add this information in line 245 as well. This has been added but in lines 265-267, in relation to the first mention of the seasonal calculations.**

16) Line 280: this has been stated before.

**We understand that the phrasing is similar, however, line 255 refers to the fact that we do not show areas with less than 100 observations in figure 5. In line 280, we refer to the method for the evaluation of the model, where we do not evaluate such areas. Thus, the information, even if similar, do not refer to the exact same thing.**

17) Section 3.1: the authors should include a North Sea station here, maybe one on the Danish or Dutch transects or an individual research station from the UK. It may not have everything the authors want but an extension of the SCOBI model into the North Sea and Channel areas should be validated in detail there. Stations like the Oystergrounds (NL), West Gabbard (UK) or L4 (UK) spring to mind, though the latter is I think just outside of the domain. These may have standard surface monitoring and limited at depth monitoring, but it is better than nothing. They also have high resolution observational data for a few years, generally. In the very least a North Sea station comparison will provide more detail on the local Chl-a seasonal signal and bloom timing capacity of the model there (difficult to derive from figure 12).

**We thank the reviewer for providing recommendations for further stations. In our study, we used the ICES data, which should include all publicly available data for the North Sea. Figure 12 together with figure B4 already shows that the general spatial pattern of the phytoplankton bloom in the North Sea is actually well captured, but that the model mainly underestimates it in the southern coast of the North Sea in winter, spring and autumn, while in summer it overestimates it. They also give information on the timing of the bloom and show that chlorophyll-a values in the model do increase when observation values also increase in most areas in the North Sea, except at very coastal areas during winter and autumn. This indicates that the timing is generally well captured in the North Sea, except for these very coastal areas (so a very small part of the domain) where the blooming lasts longer in observations than in the model. We therefore argue that single stations in the North Sea will not provide much more new information in terms of long-term trends and general patterns. However, we will further look into specific stations, keeping in mind that this is an ocean model and not a coastal model, and add text if new relevant information is found.**

**We have now added time series of nitrate and phosphate at three stations in the southern North Sea which showed the largest number of observations for the period 2001 to 2017 in the ICES database (e.g., Terschelling 100 km). Most stations with high number of observations are extremely coastal and NEMO-SCOBI is not meant to resolve such shallow areas. However, we**

**also show observations and model results at Walcheren, which is located in an estuary along the Dutch coast. Even when considering the stations with the largest number of observations, it is not enough to perform the same statistical analysis as that done for the other stations and therefore no further statistics have been applied to these North Sea stations. As expected, not much new relevant information could be extracted from such plots, except that they do show a general good agreement with model results in both magnitude and seasonality. The period of the year with high nitrate concentrations seems to last longer in the model than in the observations, resulting in an overall overestimation of nitrate. This bias is larger in Walcheren than in the offshore stations. This information is complementary to what can already be seen in the maps shown in the main text (now Fig. 12 and 13) and appendix (Fig. B4). Therefore, we have added these new figures (now B3 and B4) only in the appendix (Appendix B), where both figures are described in lines 893-898. The figures are referred to in the main text in lines 557 and 582. These North Sea stations were also added to the map in Fig. 1 and the caption of Figure 1 was modified accordingly.**

**Regarding the high resolution chlorophyll-a, we are aware that these exist for recent years (this is also the case for the Baltic Sea), but have chosen not to include them in this particular study as they do not cover the complete studied period (2001 to 2017). Again, the focus of this work is to validate the model performance on long-temporal trends and general biogeochemical patterns, which we have explained in detail in the text. We agree with the reviewer that high resolution data is extremely valuable and relevant for model improvements and we are currently including such observations (as well as satellite data) for further analysis. However, we think it is too much information to add in this study, as such observations should come with additional detail description. Please also see our reply above (page 4, reply 0.7).**

18) Line 315-317: can you provide an overview of the trends in table form in the appendix? Now it is hard to see and compare trends.

**We will summarize the trends shown in figures 6 and 8 in a readable table and add it where relevant. A new table has been added in section 3.1 (Table 2., page 18).**

19) Figures 6 and 8: I would suggest restructuring these. A label over results in a graph is a no-go, in any case. Suggestion: make a two column graph (which these are already). Top left: the legend for surface values. Rest of left: surface graphs for T, S, NO3, PO4, Chl-a. Top right: legend for bottom values. Rest of right: bottom graphs for T, S, NO3, PO4, O2. The legend for O2 can be removed and explained in the caption. This would make the graph more accessible as surface or bottom processes can be viewed at a glance (vertically) while top and bottom values can still be compared easily (horizontally).

**The figures will be improved (also in reply to reviewer # 2, comment 14). Figures 6 and 8 have been improved by moving the legend where so it is not on top of the results.**

20) Line 330-333: no guarantees that the measurements did not fail to, the double negative here makes this sentence hard to read. I presume your point is that observational evidence is discrete in time and so can easily miss the peak of the spring bloom. This is a very valid and important point to make, which merits unambiguous text.

**"no guarantees that the measurements actually captured the chlorophyll peaks..."**

**Line 365: 'did not fail to' has been changed to 'actually'.**

21) Line 350-354: the model correctly predicts inflow of North Sea waters into the Baltic proper, though bottom temperature and salinity values are too low compared to observations. But this is a feature of the existing hydrodynamics model, NEMO-Nordic, which was already used in the presented domain before, and calibrated and validated there. I would not expect the extension of the SCOBI model to influence these dynamics.

**The reviewer is correct: the code for the dynamics in NEMO-Nordic has, indeed not been changed and the SCOBI model does not affect the hydrology of NEMO-Nordic. However, the applied physical forcing has changed, especially that for runoff. As mentioned in lines 378, the new river forcing significantly improved the surface salinity results in the Baltic Sea when compared to results in (Hordoir et al., 2019), but increased the existing negative bias in intermediate and deep waters of the Baltic proper. The Baltic Sea in NEMO-Nordic is, fairly sensitive to changes in fresh water input and, while we are currently working on improving this bias, here we have chosen to compromise the known bias in deep waters in the Baltic proper for improved surface results.**

22) Figures 7 and 9: the little cyan plusses (not crosses as it says in the caption, that would be x) are very hard to see. Can this be done by shading instead? I do love the surface values on top of the depth graphs, very nicely done!

**Thanks! We have already tried the shading and it is not great as it hides the standard deviation for the model (or make it less visible). However, we will further look on how to improve the averaged profile figures. We have improved figures 7 and 9 by replacing the cross markers by error bars and moved the legend so it is not on top of results.**

23) Line 435: figure B3 is mentioned here, I would prefer to see figure B3 in the main text rather than in the appendix. If there is a limited number of figures allowed, I would suggest moving figure 5 to the appendix instead, as it does not show simulated results. Within B3 the markers are very hard to see, can you make then larger? Some of the colours are quite light, resulting in a number without a visible marker in my printed version: enlargement might help with this too.

**We will indeed enlarge the markers of Figure B3. Done. Regarding its position in the manuscript, we had the same discussion with the co-authors before submission and decided that it was too detailed information for the main text. If adding it to the main text, a detailed discussion on it should be added as well, describing the main differences between areas, parameters and position in the water column. This will significantly complicate the flow and readability of the manuscript. The main relevance of this figure is to show whether the model performs well or not. For example, if wanting to use the model results for a special variable in a particular area, one can look at this figure and have an idea if such results can be trusted or not. It also helps the developers of NEMO-SCOBI to see if changes in the code or forcing improve both these model performance figures (namely Figure 10 and B3). Importantly, the figure results for the North Sea below surface waters are uncertain due to the great lack of observations (as clearly seen in Figure 5). Moreover, we have summarized the relevant patterns in section 3.4 and discussed the main processes that could have an effect on the seen model biases in section 3.6, which is the main focus of this manuscript. It is also somewhat repetitive information as many of the area points show similar**

**positions than those of Figure 10, especially when looking at surface values. As mentioned before, we prefer to leave figure B3 in appendix.**

24) Line 463: in the Baltic Sea, four HELCOM-OSPAR assessment areas. Surely these are HELCOM assessment areas?

**We agree that this is confusing, this will be changed. This has been changed. The change is now in lines 498-499.**

25) Line 477-4780: surely you can see in your simulation results if accumulation happens or not?

**Yes. We do see an accumulation over time in the time series at stations in the Gulf of Bothnia when compared to observations. However, there are only a few data points for nitrate at for example F3 and F9 in the entire time series and even if they are clearly below model results in recent years, it is difficult to say much more than this.**

26) Line 483-485: this is an important message for the monitoring organisations, please make it stronger.

**It is indeed a relevant point and we will emphasize it more in the text. This is now highlighted in the conclusions, lines 745-747. Note that we now also emphasize the lack of observations in the Northern North Sea.**

27) Line 492-493: grammatically incorrect sentence and it doesnt make much sense.

**We agree with the reviewer that this line is confusing. "The Northern North Sea" is the official name for the central area in the OSPAR assessment areas. We refer to "the HELCOM-OSPAR assessment areas" to the areas adapted to our model domain, which means that they are not exactly the same, notably for OSPAR. The OSPAR assessment areas include also the North Atlantic and therefore some of the areas around the boundaries are cut in our domain. What we mean in this sentence is that the central part of the North Sea is the largest assessment area in figure 1 but also the least measured, which makes the model skill evaluation unreliable. We will rephrase this sentence.**

**The sentence has been rephrased as follows (lines 526-528):**

**"In addition, the Northern North Sea (area 25 in Fig. 1) is the largest assessment area of the North Sea and the least spatially measured regarding phosphate and nitrate ..."**

28) Line 494-495: surely this is not about which model is better? Grammatically also incorrect, I assume the model by Daewel et al captures the southern coast of the North Sea just fine. Maybe not in biogeochemical terms, but the coastline itself is in the model so it captures it.

**Correct, it is certainly not a competition between models. We are only highlighting the areas where most models fail to capture the nutrient dynamics. This will be rephrased as follows:**

**"Nonetheless, our biogeochemical results are comparable to those of Daewel and Schrum (2013), which is in overall good agreement with observations in the North Sea but with biases in the southern coast of the North Sea." This has been rephrased in lines 529-531.**

29) Line 505: sentence is too long and loses it grammatical structure by the end. Please rephrase.

**We will split the sentence in two. This has been done in lines 540-542.**

30) Line 513-516: please speculate on what the missing process for phytoplankton growth could be. And add riverine nutrient validation to the appendix (e.g. figure 3 but with an applied forcing-observational evidence focus), to better quantify the nutrient input issue. How well does your input capture suspended matter from fluvial sources?

**The missing processes we refer to in this sentence are later on explained in section 3.5. For clarity, we will add at the end of the sentence "(further discussed in section 3.5)". This has been added to line 552. Note that section 3.5 should be section 3.6, as now correct in the new text.. Regarding suspended matter from riverine input, again, here we do not validate the riverine data set. In addition, we have modified EHYPE nutrient inputs to match those of observations and therefore we do not really understand the question. This said, we do mention (in lines 614) that there are large uncertainties regarding organic matter (detritus) in both seas. This also concerns all riverine data sets, as suspended and particulate organic matter is not necessarily automatically measured in all countries.**

31) Line 524: have you considered the following works?

Capuzzo, E., Stephens, D., Silva, T., Barry, J., & Forster, R. M. (2015). Decrease in water clarity of the southern and central North Sea during the 20th century. Global change biology, 21(6), 2206-2214.

Capuzzo, E., Painting, S. J., Forster, R. M., Greenwood, N., Stephens, D. T., & Mikkelsen, O. A. (2013). Variability in the sub-surface light climate at ecohydrodynamically distinct sites in the North Sea. Biogeochemistry, 113, 85-103.

And how does this work relate to your findings?

**We thank the reviewer for providing additional interesting references. We will look into both references and add findings if relevant for this study. These references were indeed very useful and a comparison to their findings was added in lines 465 and 668-677.**

32) Line 526: you mean the Rhine, arguably the largest river to exit into the North Sea, has no influence here? Surely not.

**Thank you for catching this. It is a typo and the Rhine will be added. The Rhine was added in this sentence, line 562.**

33) Line 532-535: figure 12 shows no observational support for this. How do you know your model is not simply overestimating the local light climate?

**Lines 532-535 do not refer to our results (no reference to Fig. 12 there), but discuss findings in the literature. The sentence will be rephrased as follows:**

**"The elevated chlorophyll-a concentrations along the eastern UK coast has been shown to be likely related ..." This has been rephrased, now in line 569.**

**Please note that we do mention in line 562 that our model does overestimate the light attenuation depth. See also our reply 0.8 on light limitation and attached figures.**

34) Line 542-544: maybe, but a comparison with satellite observations could verify this point better spatially.

**Please sea our reapply in page 4, comment 0.7.**

35) Line 547: in figure B4 the matching points are hard to see as they are white, overemphasizing the discrepancies. Can you use a blue-yellow-red colourbar here to emphasize where model and observational evidence do agree, and where there is simply an observational dessert? The same applied to figure 13, where observational points with a N:P ratio of (near) Redfield values are invisible.

**We will see how to improve the visibility of matching points. The matching points are now in gray (not white), so we agree that they are difficult to distinguish not from missing data but between an island/land point from an actual observation point. We have changed to a blue-yellow-red color scale for Figure B4, so that matching values are more visible. This was also done for Figure 14 (the N to P ratio maps), but it did not improve the visibility of observations. While values around 16 become more visible the relevant large patterns become less clear due to too light blues and reds when close to the value 16. Therefore the former red-gray-blue color scale is kept for Figure 14.**

36) Line 570-575: spring bloom timing is mainly driven by temperature and light conditions in the North Sea, so a discussion on the simulated light climate is due here. Diatoms have evolved to be more light receptive than most other phytoplankton species, so they lead the spring bloom. Does the biogeochemical model allow for a proper succession of species? Figure B5 indicates it does, but a general seasonal succession graph (daily resolution, maybe for the different basins) would be better to display the models inner workings. A difference of 3 months in spring bloom timing is a lot, even for a large scale biogeochemical model.

**Yes, the growing succession of species have been plotted and analyzed as these were tuned for the North Sea. It is not the main issue causing the bloom delay in the Skagerrak-Kattegat. In the text, we have highlighted the main factors that are likely the cause: the light limitation and the fact that its seasonality is not well captured in the Skagerrak-Kattegat area. We will make this point more clear in the text (in lines 562). A more detailed explanation is now added on this in lines 675-682.**

37) Figure 12: I love this graph but at the current size results are hard to compare to observational evidence. Can these graphs be enlarged? The colourbars are also hard to read.

**Thanks! The figure is full page (A4), so enlarging it may be difficult. This may depend on how the journal handles it in the final version. However, we can enlarge the fonts. We have enlarged the font size and separated winter and spring from summer and autumn figures, so the figures are bigger now. Figure 12 is now Figure 12 and Figure 13.**

38) Line 580: allows for a study of the North Sea

**This will be changed according to reviewer's suggestion. Done, now in line 629.**

39) Line 584: , rather than prescribed boundary conditions

**This will be changed according to reviewer's suggestion. Done, now in line 634.**

40) Line 585: again, this should not be a model contest on who performs best.

**We fully agree, but still comparison is required to understand model discrepancies. We leave the sentence as it is also in response to reviewer #2, who is keen on knowing the performance of this model in comparison with others.**

41) Line 595-597: you have shown that your model is capable of simulations from which you can derive relevant indicators for HELCOM and OSPAR, taking into account model performance and bias. Certainly for Chl-a there would be caveats, but most models have these. But you have not shown that the model can be used for climate projections with specific relevant improvements, as you have not made these improvements yet. And there is no detailed information in the manuscript on what these improvements would be: several ideas have been floated but there was no priority list of things to implement in the model. I would remove the latter part of this statement. For example, line 650 list improving the seasonal cycle of benthic denitrification, but contains no statement to how important this is with regard to other suggested improvement (e.g. cyanobacteria life cycle inclusion), or how this will be achieved.

**A priority list is difficult to provide, because all mentioned future changes have a relevant role and can lead to significant improvements. However, solving the timing of the chlorophyll-a bloom is the first thing we will consider for further improvements. Therefore, it is the first point mentioned in the discussions of the section 'Future plan and knowledge gaps'. For this to be clear, we will rephrase the first sentence of this section as follows:**

**"Solving the phytoplankton bloom timing in the southeastern coastal North Sea and the Skagerrak-Kattegat transition zone in NEMO-SCOBI is a priority as it would significantly improve the model results, ..." This is now rephrased in line 664.**

**Note that this work necessarily implies improving the light penetration depth (as mentioned in line 612). In addition, we have shown that the main spatio-temporal patterns are well captured and with the evaluation made here, we have a very good idea of where results can be trusted and where not. Of course, we do want to improve our model results and further understand them. However, this is not a major problem for the future projections, because that work will not focus on seasonal patterns, but on yearly averages (or even period averages). We have shown that these are in good agreement with observations and therefore, the model can be used as it is. For climate projection the aim is to keep the model as simple as possible, but ensuring that the main processes are captured. This is to be able to perform long-term runs at a reasonable computational cost.**

42) Line 596-597: this is why I want to see a validation of the applied riverine forcing. The atmospheric deposition bias was discussed, but the reader lacks information on the riverine input bias.

**Again, the focus of the paper is not on river validation. However, we will add more discussion on the river's effect. Please see our detailed reply to comment 12 and 13.**

43) Line 613: how about suspended sediment?

**The model does not take into account suspended inorganic sediments. However, it accounts for resuspended material from the sediments and it is taken into account for the light penetration depth as they go back to the corresponding nutrient pool in the water column (see for example N1 and P5 fluxes in figure 1).**

44) Line 633: please provide references for the claim that the model compares well with previously published estimates (assuming you mean other publications than Dalsgaard et al, 2013).

**In line 633, we indeed compare the model values to the values mentioned in Dalsgaard et al, 2013, which are mentioned just above (line 631). We agree with the reviewer that this is confusing and will rephrase these lines as follows:**

**line 630 "The total nitrogen removal from water column denitrification in the Baltic proper with persistent large hypoxic areas has been estimated to be..." This is now rephrased in line 710.**

**and**

**line 633 "This compares well with previous published estimates" will be removed from the text. Done.**

**4   Minor additional changes**

1) We have made some rephrasing to improve the readability of the conclusions as follows:

Lines 736-737: "It is therefore ready to be used to produce climate projection **such as those in the SMHI** Climate Scenario Service."

Line 739: "... NEMO-SCOBI avoids many problems associated with a lateral boundary in the area of Kattegat and Skagerrak. **This is of** fundamental importance for the salt and oxygen inventory of the Baltic Sea as it controls the North Sea-Baltic Sea mass exchange."

Lines 745-747: "**It should be noted that robust statistical evaluations for long-time trends are particularly difficult to obtain in the Northern North Sea, the Gulf of Riga, the Gulf of Finland, the Åland Sea, the Quark and the Bothnian Bay due to great lack of observations in these areas**"

2) We have rephrased the caption of Figure 10, as it was misleading:

**"Shown biases are for the whole period (black) and for each season (colors) for water-column"**...

3) We have moved the description of the acronyms HELCOM, COMP4 and OSPAR at their first appearance in the text (Introduction, lines 105-107) instead of in section 2.2.2. We have also removed the references HELCOM (2013) and OSPAR (2022) from line 23 to avoid confusion.

6) For clarity, we have rephrased the following:

Lines 353-355 "The model trends in the Skagerrak-Kattegat transition zone are strongly linked to the applied river forcing in this region **which shows an increase in** nutrients from the start of the period followed by a decrease after the 1990s ..."

Line 748: "allows **for better identification** of regions with similar biogeochemical behavior that are not limited to one or the other sea,

7) We have corrected typos as follows:

line 73: we have added **the** after (late-)summer cyanobacteria bloom

line 78: replaced 'on' by **in**

line 101: a comma was added after 'study'

line 104: replaced 'observational based' by **observational-based**

line 161: replaced 'are' by **is**

line 192: replaced 'were' by **was**

line 263: replaced 'null' by **the null**

line 264: replaced 'is' by **are**

line 274: replaced 'over' by **from**

line 275: we added **its corresponding**

line 284: replace 'are' by **is**

line 307: removed **both** and changed line 302 to **is also**

line 318: replaced 'show' by **shows**

line 322-323: corrected station names **SLÄGGÖ** and **HANÖBUKTEN**

line 330: replaced 'set' by **sets**

line 359: replaced 'from 1996 to 2017' by **from 1975 to 1996 and towards 2017 (Fig. 6e and Table 2)**

line 371: 'that' was removed

line 379: replaced '(with regression slopes near zero)' by **(with small regression slopes)**

line 380: added **both model and**

In the last sentence of the caption of figure 6, 'are' is replaced by **is**

line 381: '1990's, however' has been changed to **1990s, however,**

In the caption of Figure 7, 'display' was replaced by **displayed**

Line 410: 'is' was replaced by **are**

Line 427: 'have' was replaced by **has**

Line 437: 'however' was replaced by **however,**

Line 443 was rephrased as **the model and the observations differ**

Line 498: 'to capture' was replaced by **from capturing**

Line 499: 'include' was replaced by **includes**

Line 504: 'bias' was replaced by **biases**

Line 513: 'In such case,' was replaced by **In this case,**

Line 528: 'that' was replaced by **since**

Line 532: 'heavily' was replaced by **is heavily**

Line 554: 'in' was replaced by **of**

In the caption of Figure 11, 'waters' was added after 'surface'

Line 596: 'the' was added before 'open and coastal ocean

Line 599: the comma was removed after 'cyanobacteria bloom starts in the summer.

Line 647: a comma was added after 'To our knowledge'

Line 657: 'bias' was replaced by **biases**

Line 658: 'of' was replaced by **on**

Line 681: 'better correlation' was replaced by **a better correlation**

Line 688: 'on' was replaced by **in** and **also** was added

Line 691: 'to' was replaced by **from**

Line 708: 'over' was replaced by **from**

Line 723: 'is' and 'has' were replaced by **are** and **have**

Line 728: 'vary' and 'have' were replaced by **varies**

Line 923: 'Jonathan' was replaced by **Nathan**, as he has now changed his officially name to Nathan Grivault

---

## Author Response (AR2)

Dear Editor,

We have addressed the final corrections, except the suggestion for Figure A1. This is because observations cannot be plotted together with model output in the same axes. They fall outside the graph. That is why we have added them in the caption, as reference.

In addition, we have corrected a sentence in line 180 as it was misleading and it is now better described in the accompanying document about the river forcing:

old text:  "by recommendation of the E-HYPE developers, due to an overestimation in the precipitation in this E-HYPE run, especially over the Baltic Sea"

new text: "based on the best results for salinity in surface waters from a sensitivity analysis performed with NEMO-SCOBI."

We have also updated the reference for the accompanying document, which will be public as soon as we have the new DOI and final references of the manuscrupt.

Thank you and best regards,

Itzel Ruvalcaba Baroni